# Ammonium-derived nitrous oxide is a global source in streams

Shanyun Wang[1,2], Bangrui Lan[1,2], Longbin Yu[1,2], Manyi Xiao[1], Liping Jiang[1,2], Yu Qin[1,2], Yucheng Jin[1], Yuting Zhou[1], Gawhar Armanbek[1,2], Jingchen Ma[1], Manting Wang[1], Mike S. M. Jetten[3], Hanqin Tian ◉[4,5], Guibing Zhu ◉[1,2] ✉ & Yong-Guan Zhu ◉[1,2]

Global riverine nitrous oxide ($N_2O$) emissions have increased more than 4-fold in the last century. It has been estimated that the hyporheic zones in small streams alone may contribute approximately 85% of these $N_2O$ emissions. However, the mechanisms and pathways controlling hyporheic $N_2O$ production in stream ecosystems remain unknown. Here, we report that ammonia-derived pathways, rather than the nitrate-derived pathways, are the dominant hyporheic $N_2O$ sources (69.6 ± 2.1%) in agricultural streams around the world. The $N_2O$ fluxes are mainly in positive correlation with ammonia. The potential $N_2O$ metabolic pathways of metagenome-assembled genomes (MAGs) provides evidence that nitrifying bacteria contain greater abundances of $N_2O$ production-related genes than denitrifying bacteria. Taken together, this study highlights the importance of mitigating agriculturally derived ammonium in low-order agricultural streams in controlling $N_2O$ emissions. Global models of riverine ecosystems need to better represent ammonia-derived pathways for accurately estimating and predicting riverine $N_2O$ emissions.

Nitrous oxide ($N_2O$) is a potent greenhouse gas and dominant ozone-depleting substance, and its atmospheric mole fraction has increased by 23% since 1750[1]. Riverine $N_2O$ emissions have increased fourfold since 1900 and are important components of the global $N_2O$ budget[2]. A 2020 global modeling study[3] showed that small streams (lower-order streams rather than high-order streams) contribute up to 85% of the global riverine nitrous oxide ($N_2O$) emissions[3], most of which are produced in hyporheic zones (beneath stream beds where stream waters exchange with adjacent sediments). However, in current process-based $N_2O$ models, the $N_2O/N_2$ ratio is used to represent the $N_2O$ production rate during denitrification[3]. Model parameters associated with microbially mediated hyporheic $N_2O$ production are poorly represented[3,4]. Although the IPCC Guidelines for national greenhouse gas inventories include $N_2O$ from nitrification[4], previous studies have

been focused on quantifying denitrification and $N_2O$ emissions resulting from nitrate ($NO_3^-$)[5,6]. Furthermore, it is difficult to distinguish between nitrifier denitrification, nitrification-coupled denitrification, and heterotrophic denitrification $N_2O$ production pathways, as these processes occur via homologous genes under similar conditions, such as low $O_2$ conditions[7]. Hyporheic $N_2O$ production has been shown to be very heterogeneous[8–10], but this heterogeneity is extremely difficult to account for in models and, therefore, is often excluded[5]. Currently, the microbial mechanisms underlying $N_2O$ production in hyporheic exchange zones are largely unknown. These processes may significantly influence global $N_2O$ budgets and be potentially severely underestimated.

Accordingly, the aim of this study was to investigate the microbial sources and mechanisms that are responsible for hyporheic $N_2O$ pro-

[1]Research Center for Eco-Environmental Sciences, Chinese Academy of Sciences, Beijing 100085, China. [2]University of Chinese Academy of Sciences, Beijing 100049, China. [3]Department of Microbiology, Radboud University Nijmegen, Nijmegen, AJ 6525, the Netherlands. [4]Center for Earth System Science and Global Sustainability, Schiller Institute for Integrated Science and Society, Boston College, Chestnut Hill, MA 02467, USA. [5]Department of Earth and Environmental Sciences, Boston College, Chestnut Hill, MA 02467, USA. ✉e-mail: gbzhu@rcees.ac.cn

duction in riverbed and riparian zone sediments. We first investigated the spatiotemporal characteristics of N$_2$O production along transects of the Baiyangdian riverine network, the largest riverine network in the North China Plains, using isotopic $^{15}$N-$^{18}$O and $^{15}$N tracing, quantitative reverse-transcription PCR (RT–qPCR), and metagenome analysis at the site and regional scales. The North China Plains cover an area of 300,000 km$^2$ and account for 23% of the Chinese cropland area. They account for more than 25% of global fertilizer N use, making them a global N$_2$O emission hotspot[2,11,12]. However, N$_2$O emissions from agricultural streams in China[13,14] or in the North China Plains specifically[15,16] have only been reported in a few studies. We further compared these results to those measured globally in temperate and tropical streams to obtain comprehensive conclusions.

## Results and discussion

### Site-scale investigation of heterogeneous N$_2$O emission fluxes along a transect of a riverine hyporheic zone

Over a period of 4 years, the N$_2$O emission fluxes in the riparian zone and riverbed sediments along a transect of a riverine hyporheic zone were continuously measured using a closed-chamber method (Fig. 1a and Supplementary Data S1). The emission fluxes were significantly greater in summer than in winter ($t$-test, $p < 0.01$), but there were no significant differences between the two zones. Ammonium (NH$_4^+$) showed the most positive correlation with N$_2$O flux, irrespective of the sampling time and zone (Supplementary Data S2 and Table S1), as reported in many previous studies on rivers, agricultural catchments, and estuaries (Supplementary Table S2). In contrast, nitrate (NO$_3^-$) was negatively correlated with N$_2$O flux, in contrast with previous findings[17,18].

### Site-scale investigation of $^{15}$N tracing of the semi-in situ sediment core revealed NH$_4^+$-derived and NO$_3^-$-derived sources of hyporheic N$_2$O production

Generally, N$_2$O production is a biogeochemical process driven by microorganisms via two main microbial processes with four pathways: the NH$_4^+$-derived process (NH$_4^+$ as a substrate; includes nitrifier nitrification (NN), nitrifier denitrification (ND), and nitrification-coupled denitrification (NCD) pathways) and the NO$_3^-$-derived process (NO$_3^-$ as a substrate; heterotrophic denitrification (HD) pathway)[6,19–21]. Our results confirmed that the contribution of biotic N$_2$O production in our samples was more than 92–95%, and abiotic processes accounted for less than 5–8% of the total N$_2$O production (Supplementary Data S1, S2 and Fig. S1). A $^{15}$N tracing semi-in situ sediment-core incubation was performed to investigate the semi-in situ hyporheic N$_2$O production rate and to clarify NH$_4^+$-derived and NO$_3^-$-derived sources without distinguishing the NH$_4^+$-derived process along the transect of the Xiaoqinghe River hyporheic zone (Fig. 1b and Supplementary Data S3). Interestingly, the NH$_4^+$-derived process was the dominant hyporheic N$_2$O source in both the riparian zone (87.3 ± 3.9%) and the riverbed sediments (92.6 ± 5.6%), and the remaining N$_2$O production could be attributed to the NO$_3^-$-derived process.

Furthermore, we used the 0.01% C$_2$H$_2$-inhibitor method to confirm the above results (Fig. 1c and Supplementary Data S1, S2). First, we found that the potential rate of biotic N$_2$O production in the riparian zone (0.61 ± 0.03 µg N$_2$O-N kg$^{-1}$ h$^{-1}$) was lower than that in the riverbed zone (0.73 ± 0.09 µg N$_2$O-N kg$^{-1}$ h$^{-1}$), in agreement with the $^{15}$N semi-in situ sediment-core incubation results. However, this was not in agreement with the previous, which reported a higher N$_2$O production rate in the riparian zone in different river systems[18,22,23]. In the riparian and riverbed zones, NH$_4^+$-derived N$_2$O accounted for 89.2 ± 2.9% and 76.1 ± 2.6% of the total N$_2$O produced, respectively. The NO$_3^-$-derived process resulted in the production of the remaining N$_2$O. These findings differ from the prevailing opinion that NO$_3^-$-derived processes are the main contributors to N$_2$O in riverine hyporheic zones[6,24].

### Site-scale investigation of quantitative reverse-transcription PCR (RT–qPCR) analysis of N$_2$O-related gene mRNA

To further investigate the microbial mechanism and activity related to N$_2$O production, the transcript abundances of microbial N$_2$O-related genes were quantified via reverse-transcription PCR (RT–qPCR) (Fig. 1d). The transcript abundances of all N$_2$O-related (production and reduction) genes showed low heterogeneity between the riparian and riverbed zones. N$_2$O production-related genes (amoA, norB, nirS, and nirK) had greater transcriptional abundances than the reduction gene (nosZ) in both the riparian and riverbed zones, providing evidence that the hyporheic zone had greater potential for N$_2$O production. RT–qPCR cannot be used to distinguish both the nirK and norB genes in the NH$_4^+$-derived and NO$_3^-$-derived N$_2$O production pathways. However, we found that the transcript abundances of genes in the NH$_4^+$-derived N$_2$O production pathway, including the amoA gene encoded the NN pathway and the norB gene encoded the ND and NCD pathways, were greater than those of the norB gene encoded the HD pathway, irrespective of sediment location (Fig. 1d).

### Regional-scale investigation of microbial pathways and key parameters influencing hyporheic N$_2$O production

To support the above site-scale results, samples were also collected from 50 riparian zones and 50 riverbed sediments (0–20 cm depth) in 25 hyporheic zone transects at equal distances along five streams in the high- and low-water-level seasons (Fig. 2a, Supplementary Data S1, S2, and Fig. S2). The 0.01% C$_2$H$_2$ inhibitor results showed that N$_2$O production in hyporheic sediments was dominated by NH$_4^+$-derived processes, the contribution of which to N$_2$O production (72 ± 3%, $n = 100$) was significantly greater than that of NO$_3^-$-derived processes (28 ± 3%, $n = 100$) (paired $t$-test, $p < 0.0001$; Fig. 2b and Supplementary Data S2). There was less heterogeneity in hyporheic N$_2$O emissions among the five different streams or in all streams at both the site and regional scales, but there was large temporal heterogeneity at the seasonal scale, with lower heterogeneity in winter (0.63 ± 0.22 mg N kg$^{-1}$ soil d$^{-1}$) than in summer (1.26 ± 0.34 mg N kg$^{-1}$ soil d$^{-1}$) ($p < 0.05$), and the values mostly depended on the NH$_4^+$ content ($r = 0.851$, $p < 0.0001$, $n = 100$) (Fig. 2b, c; Supplementary Fig. S2; Data S2; and Table S3). This result indicates that in hyporheic riverine sediments, microbial N$_2$O generation is mainly driven by NH$_4^+$-derived processes rather than by NO$_3^-$-derived processes.

Furthermore, we used improved $^{15}$N-$^{18}$O dual-isotope tracing to verify these results in one of the five streams, the Tang River. Based on our findings, NH$_4^+$-derived N$_2$O production (73 ± 5%, $n = 10$) was significantly greater than NO$_3^-$-derived N$_2$O production (27 ± 5%, $n = 10$) (paired $t$-test, $p < 0.0001$; Fig. 2d; Supplementary Fig. S2; and Data S2, S4), which was consistent with the above 0.01 % C$_2$H$_2$-inhibitor results. Among the NH$_4^+$-derived pathways, the ND pathway was dominant upstream (38 ± 4%, $n = 4$), downstream (44 ± 3%, $n = 6$), in riparian zone sediments (45 ± 3%, $n = 5$), and in riverbed sediments (38 ± 4%, $n = 5$), with little heterogeneity at both the site and regional scales. The remaining NH$_4^+$-derived N$_2$O production was attributed to the NN and NCD pathways, with a range of 8–23%. These results showed that ND was an important pathway for N$_2$O generation, which has long been overlooked but has recently been widely reported[25–29]. However, the biogeochemical mechanism underlying the role and key factors influencing this pathway are still unknown.

### Regional-scale investigation of potential metabolic N$_2$O production mechanism

As previously mentioned, the nir and nor genes are involved in the ND, NCD, and HD pathways; as a result, these processes cannot be distinguished based on the presence of these genes. To resolve this issue, species annotation was combined with functional gene annotation to study the genes related to nitrogen cycling in nitrifying and denitrifying bacteria and distinguish between the nir and

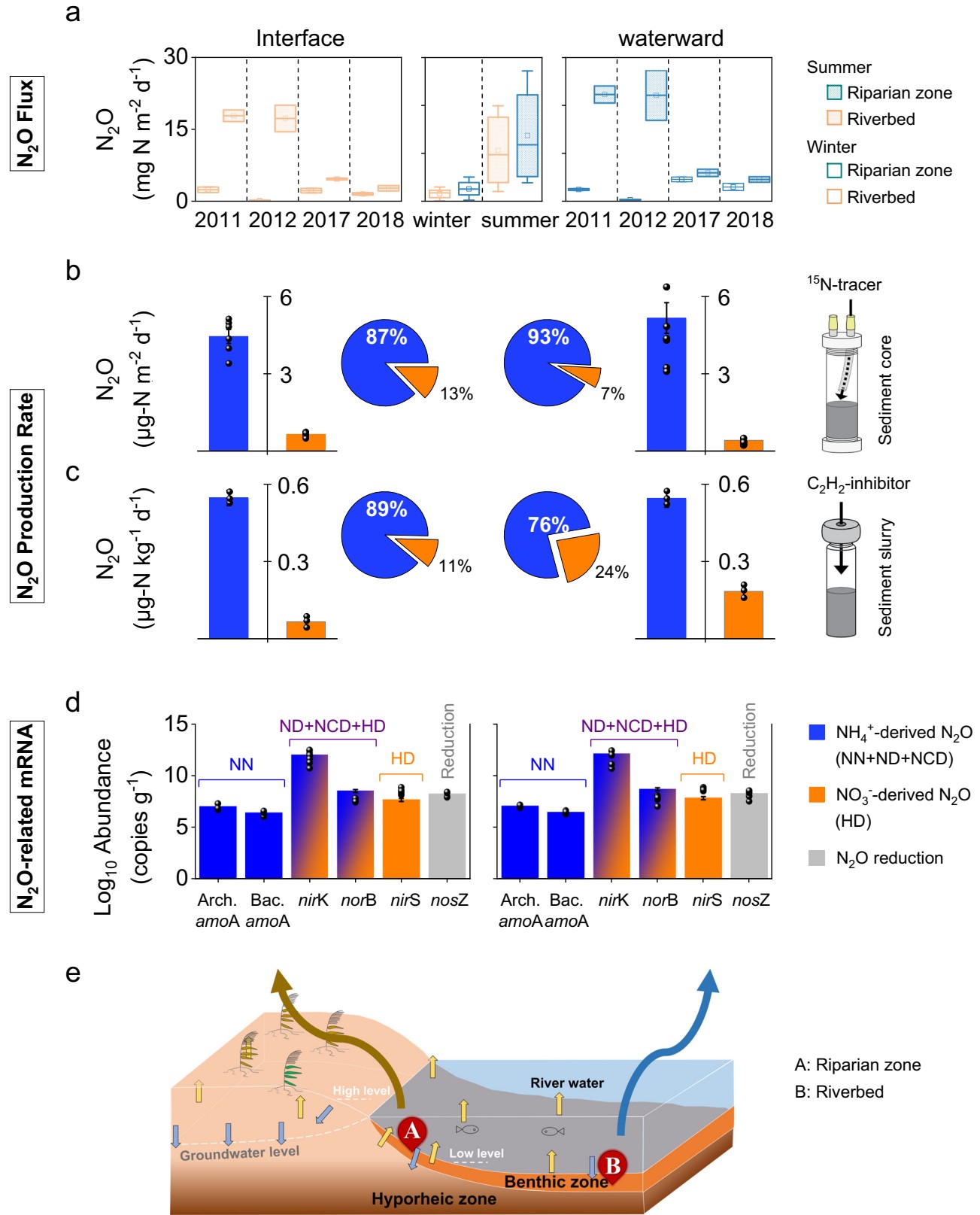

*nor* genes in ND, NCD, and HD. The assembly and binning processes generated 198 high-quality (completeness >75%, contamination <15%) metagenome-assembled genomes (MAGs), 111 of which encoded genes involved in $N_2O$ production and reduction (Supplementary Data S5, S6). Through species and functional gene annotation, we obtained eight MAGs containing *amoABC*, *hao*, or *nxrAB* genes, which have been identified as nitrifying bacteria, and seven MAGs that contained the *nirK/S* gene, identified as denitrifying bacteria[30,31]. The annotation results of the species and functional genes were further verified via phylogenetic analysis (Fig. 3a). The relative abundances of nitrifying and denitrifying bacteria were 9524 and 8310 TPM, respectively (Fig. 3b).

**Fig. 1 | Spatial and temporal N₂O emission fluxes and microbial production sources in the riparian zone and riverbed sediments along a transect of a riverine hyporheic zone at the site scale. a** N₂O emission flux over 4 years of monitoring ($n = 48$ independent experiments). For each box chart, the horizontal line indicates the median, the box represents the 25th and 75th percentiles, and the whisker shows the range from the 5th to the 95th percentile. **b, c** The microbial rate and contribution to N₂O production via $NH_4^+$-derived and $NO_3^-$-derived processes based on semi-in situ sediment-core incubation via the $^{15}N$ tracing method (**b**) and slurry incubation involving the $C_2H_2$-inhibitor method (**c**) ($n = 6$ and $n = 3$ biologically independent samples for core and slurry incubation, respectively). Data were

presented as mean values ± SEM; **d** Transcript abundance of N₂O-production related (*amo*A, *nor*B, *nir*S, and *nir*K) and N₂O-reduction related (*nos*Z) genes ($n = 3$ biologically independent samples). Data were presented as mean values ± SEM. Here, *amo*A, ammonia monooxygenase gene encoded the NN pathway ($NH_4^+ \rightarrow NH_2OH$); *nir*SK, nitrite reductase genes ($NO_2^- \rightarrow NO$) encoded the ND (*nir*K only), NCD and HD pathways; *nor*B, nitric oxide reductase gene ($NO \rightarrow N_2O$) encoded the ND, NCD and HD pathways; and *nos*Z, nitrous oxide reductase gene ($N_2O \rightarrow N_2$) encoded the NCD and HD pathways. **e** Schematic representation of the sampling sites along a transect of a riverine hyporheic zone.

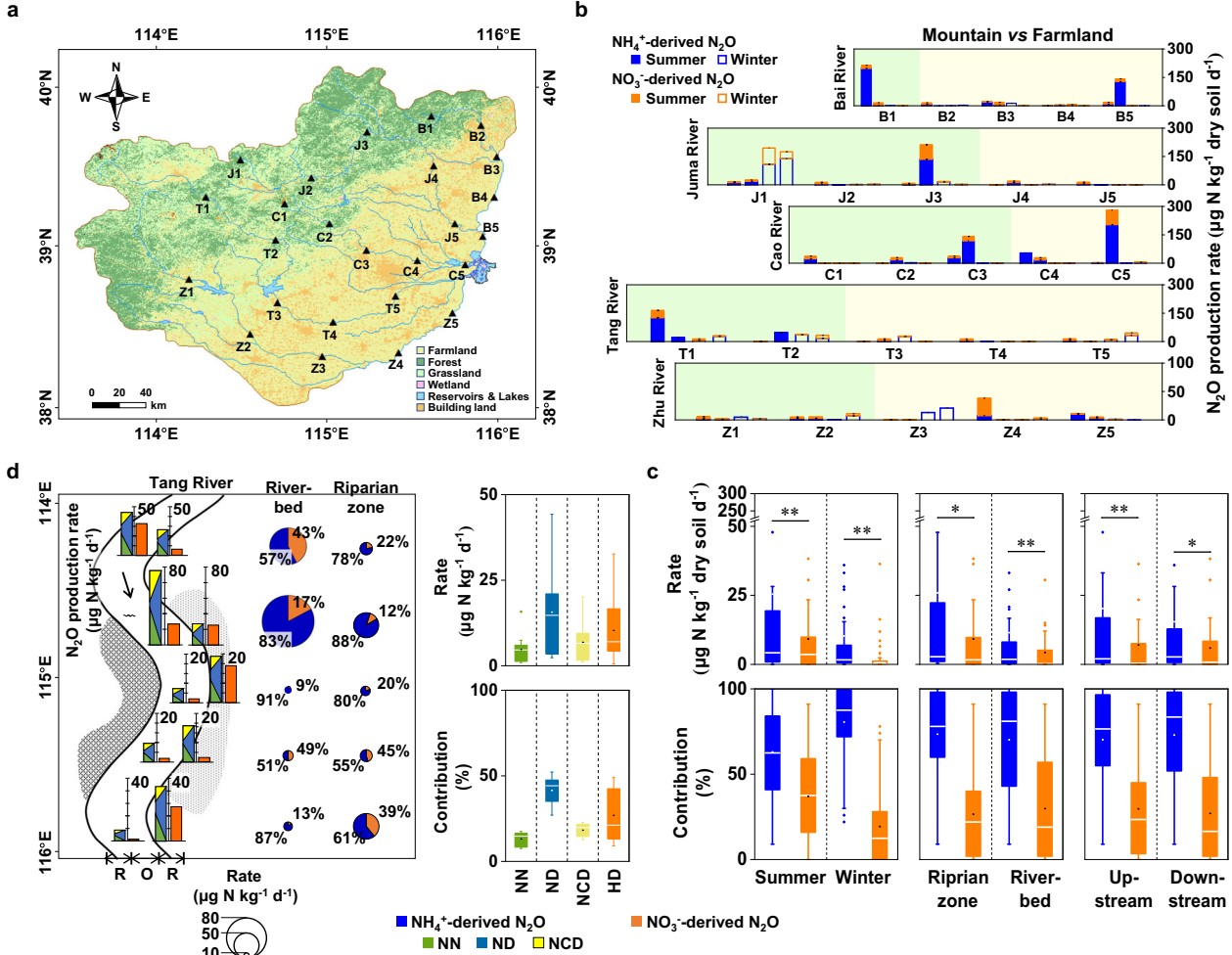

**Fig. 2 | Biogeographical distribution of microbial N₂O production sources in the riparian zone and riverbed sediments along transects of riverine hyporheic zones at the regional scale. a** Overview of the Baiyangdian riverine network and the sampling sites; **b** Spatiotemporal distribution of the potential rate of $NH_4^+$-derived and $NO_3^-$-derived N₂O production in the five streams (46–273 km length) of the Baiyangdian riverine network ($n = 3$ biologically independent samples). Data were presented as mean values ± SEM; **c** Spatiotemporal heterogeneity analyses of the potential rate and contribution of $NH_4^+$-derived and $NO_3^-$-derived pathways at the regional scale. *P* values were calculated with the two-tailed independent *t*-test

(*$p < 0.05$, **$p < 0.01$ two-tailed; Except $n = 44$ and 56 independent experiments for upstream and downstream, respectively, $n = 50$ for other groups); **d** Potential rates and contributions of nitrifier nitrification (NN), nitrifier denitrification (ND), nitrification-coupled denitrification (NCD), and heterotrophic denitrification (HD) pathways in the sediments of the riverine hyporheic zones in the Tang River ($n = 20$ independent experiments). For each box chart, the horizontal line indicates the median, the box represents the 25th and 75th percentiles, and the whisker shows the range from the 5th to the 95th percentile. Bai River, B; Juma River, J; Cao River, C; Tang River, T; Zhulong River, Z; and Riparian zone, R; riverbed zone, O.

Subsequently, the abundances of the N₂O-producing genes *amo*ABC, *hao*, *nir*SK, and *nor* in nitrifying and denitrifying bacteria were determined (Fig. 3b and Supplementary Data S6). Traditionally, $NH_4^+$ oxidation is the rate-limiting step in nitrification[32–34] and in the N cycle[35]. However, both the total and individual abundances of N₂O-producing genes in nitrifying bacteria (26, 8, 48, and 6 TPM, respectively) were significantly greater than those in denitrifying bacteria (0, 0, 6, and

4 TPM, respectively). In contrast, the abundance of the N₂O-reducing gene *nos*Z ($N_2O \rightarrow N_2$) was 4 TPM in denitrifying bacteria, but it was not detected in nitrifying bacteria (Fig. 3b). These results further revealed the mechanism of microbial N₂O production and highlighted the dominant role of nitrifying bacteria ($NH_4^+$-derived pathways) in N₂O production.

The microbial mechanism of $NH_4^+$-derived and $NO_3^-$-derived N₂O production was further revealed from the perspective of energy

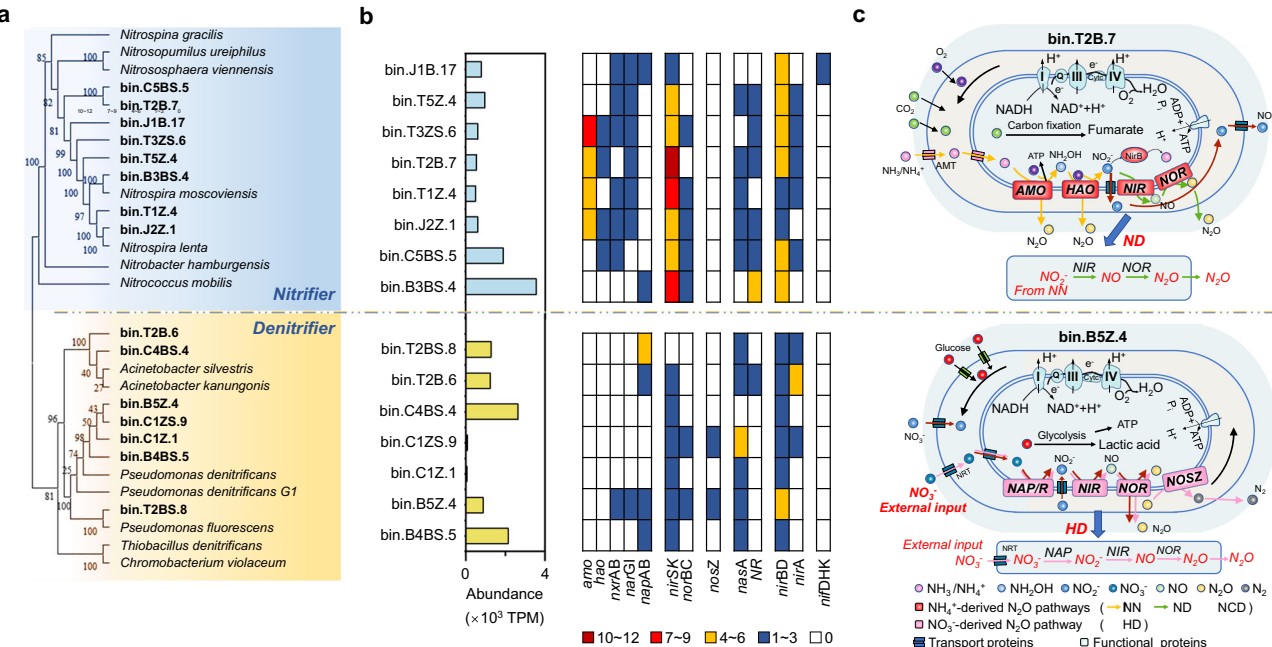

**Fig. 3 | Metabolic N$_2$O production mechanism. a** Phylogenetic diagram of the genome associated with metagenomic recombinant binning. Based on the 16S rRNA sequence, phylogenetic relationships were calculated using the maximum likelihood method. The sequences associated with the N cycle based on the NCBI database and metagenome-assembled genomes (MAGs) identified in this study are shown in black and red, respectively. Each MAG was labeled as bin."MAG source"."MAG number" and included River (Bai River, B; Juma River, J; Cao River, C; Tang River, T and Zhulong River, Z), Site (1, 2, 3, 4, and 5), Zone (Riparian zone, R; Riverbed zone, O), and Season (Summer, S; Winter, W). The species, abundance, and metabolic path diagrams of nitrifiers and denitrifiers are shown above and below the orange dotted line. **b** Relative abundance of 15 MAGs and gene recombination and characterization of N$_2$O production- and metabolism-related microbial processes. The cyan bar graph represents the relative abundance (TPM) of 15 MAGs. Different colors in the grid represent different gene abundances (TPMs). **c** Metabolic relationship of substances related to N$_2$O production. The bin.T2B.7 genome and the bin.B5Z.4 genome represent nitrifiers and denitrifiers, respectively. The black arrows in the figure indicate material transfer, transformation, or electron transport. The yellow, green, brown, and pink arrows represent the NN, ND, NCD, and HD pathways, respectively, in the nitrogen cycle. Filled blocks of different shapes or colors represent different enzymes in cells.

metabolism by metagenome binning analysis and is shown as a single-cell draft (Fig. 3c). The transmembrane transport of nitrite and nitrate, which are necessary substrates for ND and HD, respectively, is an energy-consuming process[36,37]. The NO$_3^-$ and NO$_2^-$ produced by nitrifying bacteria can be directly used in the ND pathway as substrates[38], whereas denitrifying bacteria in the HD process need to consume adenosine triphosphate (ATP) to absorb NO$_3^-$ from the environment[36]. Additionally, the *nir*KS gene is significantly more abundant in nitrifying microorganisms than in denitrifying microorganisms ($p = 0.003$); hence, in the consumption of NO$_2^-$ for detoxification[39–41], the ND pathway has greater potential to alleviate the toxic effects of nitrite on microorganisms[39–41] than does the HD pathway. Furthermore, nitrifiers have the potential to produce large amounts of N$_2$O because, based on reports to date, nitrifying bacteria do not contain N$_2$O reductase[42]. Denitrifying bacteria have the *nos*Z gene encoding N$_2$O reductase; hence, the N$_2$O produced during NCD and HD can be further reduced to N$_2$[43]. In contrast, there is no *nos*Z gene in nitrifying bacteria, and the N$_2$O produced by NN and ND is not consumed within the same cells[7,27,44] and can be released. Overall, there is greater potential for N$_2$O production from the NH$_4^+$-derived pathways than from the NO$_3^-$-derived pathway because of energy savings, NO$_2^-$ detoxification, and the absence of N$_2$O reductase.

## Global-scale investigation of hyporheic N$_2$O production in streams

The above findings indicate that NH$_4^+$-derived pathways, rather than NO$_3^-$-derived pathways, largely influence hyporheic N$_2$O production in the Yangtze, Yellow, Pearl, Yarlung Zangbo, Huai, Liao, Songhua-jiang, and Heilongjiang river basins. To further expand our survey, we carried out a global spatiotemporal investigation across a wide range of streams around the world using the 0.01% C$_2$H$_2$ inhibitor method (Supplementary Data S1). Analysis conducted on a spatial scale showed that the rates of NH$_4^+$-derived N$_2$O production were significantly greater than those of the NO$_3^-$-derived pathway at both the regional ($4.56 \pm 1.20$ vs. $3.60 \pm 1.68 \, \mu$g N kg$^{-1}$ d$^{-1}$, $n = 27$) and global scales ($6.00 \pm 1.20$ vs. $2.40 \pm 0.72 \, \mu$g N kg$^{-1}$ d$^{-1}$, $n = 11$, $p = 0.003$) (Fig. 4a, b and Supplementary Table S1). Seasonal scale analysis in the Buerhatong River (15–22 °C) and Xiaoqing River (14–23 °C) also revealed that the NH$_4^+$-derived N$_2$O production rates ($8.40 \pm 2.40 \, \mu$g N kg$^{-1}$ d$^{-1}$) were substantially greater than the NO$_3^-$-derived rates ($3.60 \pm 1.68 \, \mu$g N kg$^{-1}$ d$^{-1}$; $n = 4$, $p = 0.001$) and were the main contributors to N$_2$O production ($67.5 \pm 6.7\%$) (Fig. 4c). Statistical analysis at the spatial and temporal scales revealed a similar trend, namely, that NH$_4^+$-derived pathways contributed most significantly to N$_2$O production ($64.7 \pm 2.9\%$) (Fig. 4d). In summary, NH$_4^+$-derived pathways are the dominant hyporheic N$_2$O sources in low-order agricultural streams around the world.

Based on our results, the NH$_4^+$-derived process, rather than the NO$_3^-$-derived process, is the dominant hyporheic N$_2$O source in lower-order agricultural streams. Together, these findings provide insights into better estimation of N$_2$O emissions in global models of riverine ecosystems, which has been under debate for many years because of either under- or overestimation of riverine N$_2$O budgets in IPCC assessments[2,5,39]. Furthermore, the results emphasize the importance of managing ammonium[11]. This is particularly important, as China has long been the largest ammonia fertilizer consumer in the world[12]. Approximately 60% of the fertilizer applied is not utilized by crops but leaches into stream systems, stimulating global climate warming; that is, the climate benefits of increased CO$_2$ uptake via crop ammonia application are outweighed by stimulated greenhouse gas production

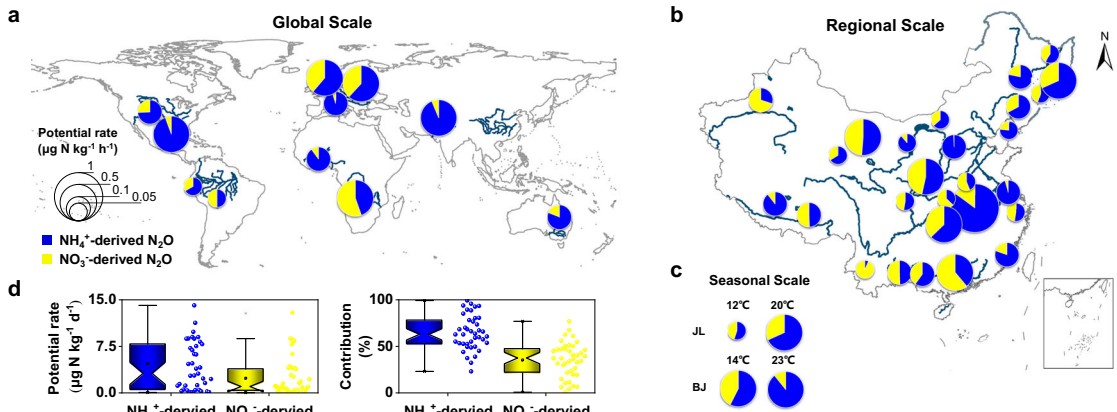

**Fig. 4 | Significance and contribution of NH₄⁺-derived pathways to riverine hyporheic N₂O production at the global (a), regional (b), and temporal (c) scales and statistical analysis of rates and contributions (d). a–c** The locations of the pie charts show the sampling sites; the sites for temporal-scale analysis are marked with solid orange circles. Detailed information on the sampling sites can be found in Supplementary Data S1. For each sampling site, the pie chart shows the total N₂O production rate. The pie chart colors represent the relative contributions of the NH₄⁺-derived (blue) and NO₃⁻-derived (yellow) pathways to total N₂O production. **d** Box charts (the horizontal line indicates the median, the box indicates the 25th and 75th percentiles, the whisker shows the range from the 5th to 95th percentiles, and the colors show the N₂O production pathways) indicating the potential rate of N₂O production via NH₄⁺-derived and NO₃⁻-derived pathways and their related contributions in riverine hyporheic zones worldwide ($n = 41$ independent experiments).

from ammonia pollution[12]. Therefore, it is necessary to optimize fertilizer and ammonia management.

## Methods

### Study site and sampling

The Baiyangdian riverine network (113°40′–116°20′E, 38°10′–40°10′N) in the Haihe River Basin, with a total area of $3.12 \times 10^4 \text{ km}^2$, is the largest riverine network in the North China Plains. The river network density is 0.50–0.99 km/km², representing the upper level in China, with significant temporal distribution differences. Its upper and middle regions receive 30–40 first- or second-order tributaries or small streams with a total length of ~3000 km. These small streams drain various land areas in agricultural, urban, and mountain regions and are affected by various human activities and agricultural fertilization practices. Their catchment areas exceeded 5.5% of the agricultural area. First, a site-scale study was conducted on one of these small streams, the Xiaoqinghe River (Supplementary Data S1). Nine parallel in situ sediment cores (0–20 cm depth) were collected separately from the riparian (2 m away from the water–soil transition zone) and riverbed (the middle of the river) zones. Six sediment cores were subjected to semi-in situ ¹⁵N-tracer assays, and the other three were subjected to 0.01% $C_2H_2$ inhibitor assays and molecular analysis. Furthermore, a regional-scale study was conducted on five rivers in the Baiyangdian riverine network (Fig. 2a and Supplementary Data S1). A total of 100 sediment samples were collected in the riparian zone and from riverbed sediments in the five rivers at five sampling sites along each river during the dry (Jan to Mar 2021) and rainy (Aug to Oct 2021) seasons. Then, we chose steams located adjacent to farmlands as the study sites, characterized by their land use type being cropland land[45]. Twenty-eight low-order agricultural streams were sampled in major river basins worldwide, including the Mississippi River, Colorado River (North America), Amazon River, Ucayali River (South America), Elbe River, Weser River, Po River (Europe), Niger River, Zambezi River (Africa), Murray River (Oceania), Indus River, Yangtze River, Yellow River, Pearl River, Yarlung Zangbo River, Huai River, Liao River, Songhuajiang River, and Heilongjiang River (Asia) (Supplementary Data S1).

All the sediment cores were collected in triplicate using an auger (Beijing New Landmark Soil Equipment, Beijing, China) with a plexiglass tube (5.5 cm diameter, 20 cm height). The site-scale sediment cores were stored in individual plexiglass tubes, with both nozzles of each plexiglass tube sealed with caps. The regional-scale and global-scale sediment cores were placed in individual sterile plastic bags. The collected sediment cores were immediately transported to a laboratory at 4 °C for subsequent analyses (Supplementary Data S1).

### ¹⁵N-tracer assay for semi-in situ sediment-core incubation

The potential rates of NH₄⁺-derived and NO₃⁻-derived N₂O production were measured via the ¹⁵N-tracer semi-in situ incubation method for the sediment core. Once the cores were at the laboratory, the gas in the headspace of the plexiglass tube was evacuated and then adjusted to standard atmospheric pressure using high-purity Ar (99.99%; Beijing Huayuan Gas, Beijing, China). The oxygen content was immediately adjusted by the injection of a known volume of high-purity $O_2$ (99.99%; Beijing Huayuan Gas) to the site oxygen content after the same volume of Ar was withdrawn from the headspace to balance the pressure. Two treatments for ¹⁵N enrichment were applied in triplicate: (i) ¹⁵NO₃⁻ (¹⁵N at 99.19%) + ¹⁴NH₄⁺ and (ii) ¹⁵NH₄⁺ (¹⁵N at 99.16%) + ¹⁴NO₃⁻. In each treatment, final enrichments of ¹⁵N-NH₄⁺/NO₃⁻ were added to 8.0 atom% ¹⁵N excess. The final N concentrations of both NH₄⁺ and NO₃⁻ remained close to the in situ concentrations in the sediments. The ¹⁵N-enriched cores were incubated under site temperature and oxygen conditions in a constant-temperature incubator. At defined intervals (0, 18, 36, 72, and 144 h), headspace gas was collected by using a 25-mL gas-tight syringe (Agilent, USA) and transferred into a 12-mL vacuum exetainer (Labco, UK). The ¹⁵NO₃⁻ in the sediments was first converted by sponge cadmium (1.0 g) into ¹⁵NO₂⁻ and then to ²⁹N₂. The ¹⁵NH₄⁺ in the sediments was converted by hypobromite into ³⁰N₂.

### ¹⁵N-¹⁸O tracer assay for N₂O production

A slightly modified version of a previously reported improved ¹⁵N-¹⁸O dual-isotope tracing method[7,19–21] was used to carry out incubation experiments under site oxygen and temperature conditions. The improved ¹⁵N-¹⁸O dual tracing method was applied to 20 sediment samples from the Tanghe River, one of the five rivers in the Baiyangdian River. After visible roots and plant residues were removed, 5 g of homogeneous fresh sediment with 25 ml of overlying water close to the sediment (5:1, v/w) was placed into a 60-mL glass serum vial (Ochs Laborbedarf, Germany). Four treatments enriched in ¹⁸O and ¹⁵N were applied in triplicate: (i) $H_2{}^{18}O$ (¹⁸O at 97.2%) + NH₄⁺ + NO₃⁻, (ii) N¹⁸O₃⁻ (¹⁸O at 96.3%) + NH₄⁺ + NO₃⁻, (iii) ¹⁵NO₃⁻ (¹⁵N at 99.19%) + NH₄⁺ + NO₃⁻, and (iv) ¹⁵NH₄⁺ (¹⁵N at 99.16%) + NH₄⁺ + NO₃⁻. In all treatments, final enrichments

of $^{18}O$-$H_2O$/$NO_3^-$ and $^{15}N$-$NH_4^+$/$NO_3^-$ were added to 1.0 atom% $^{18}O$ and 10 atom% $^{15}N$ excess, respectively. The final N concentrations of both $NH_4^+$ and $NO_3^-$ were the site N concentrations in the fresh sediments. All vials were sealed with plugs (Ochs Laborbedarf, Germany) and aluminum crimp cap (Agilent, USA) septum-equipped lids during the incubation period. Subsequently, the gas in the vials was evacuated and then adjusted to standard atmospheric pressure using high-purity Ar (99.99%). Subsequently, the $O_2$ concentration was immediately adjusted by the injection of a known volume of high-purity $O_2$ (99.99%) to the site $O_2$ concentration after the same volume of Ar was evacuated from the headspace to balance the pressure. The treatments were incubated at 60 rpm for 36 h to guarantee complete exposure of the sediments to all substrates under site temperatures. At the end of the incubation, the gas samples were transferred to 12-mL vacuum exetainers (Labco, UK) to quantify $N_2O$ concentrations.

### 0.01% $C_2H_2$ inhibitor assay

Three treatments were applied to the sediments at each scale in triplicate: (i) no treatment (control), (ii) treatment with 0.01% $C_2H_2$ (v/v; ammonia oxidation inhibitor), and (iii) treatment with $ZnCl_2$ (600 μl, 7 M; biotic process inhibitor). The incubation conditions were the same as those in the $^{15}N$-$^{18}O$ experiment. The headspace gas was sampled with a locked syringe equipped with a Luer lock valve (25.0 ml; Agilent) and injected into a 12.0 ml vacuumed glass serum vial after 0, 3, 12, 24, and 36 h.

### $N_2O$ concentration, production rate, and contribution of $^{15}N$ and $^{18}O$ tracer assays

The $N_2O$ concentration was determined via a gas chromatograph (7890 A, Agilent, USA) with an autosampler (precisions ± 2.8%), while linear regression was carried out for the potential $N_2O$ production rate with incubation time, with coefficients of determination ($R^2$) greater than 0.80.

The $^{15}N$ and $^{18}O$ signatures of $N_2O$ in gas samples were measured using an isotope-ratio mass spectrometer (IRMS and Precon, Delta V Advantage, Thermo Fisher Scientific, Bremen, Germany; precisions <0.04 ‰ $δ^{15}N$ and <0.07 ‰ $δ^{18}O$, respectively). The relative contributions of microbial $N_2O$ production pathways ($NH_4^+$-derived pathways, NN, ND, and NCD; $NO_3^-$-derived pathway, HD) were calculated according to the methods of refs. [20,21].

### RNA extraction and quantitative reverse-transcription (RT–qPCR) assays

In the site-scale study, RNA was extracted from three parallel sediment cores from both the riparian zone and riverbed sediments using an RNeasy Power Microbiome RNA Isolation Kit (QIAGEN, Hilden, Germany) according to the manufacturer's protocol. RNA quality and concentration were estimated using a NanoDrop 2000 Spectrophotometer (NanoDrop Technologies, Wilmington, DE, USA).

Reverse transcription was performed with a PrimeScript™ RT Reagent Kit with gDNA Eraser (Perfect Real Time) (TaKaRa, Dalian, China). All RT-qPCR analyses were performed on a sequence detection system (ABI 7500; Applied Biosystems, Foster City, CA, USA) with SYBR-Green fluorescent dye (TaKaRa). The copy numbers of $N_2O$-related genes were quantified by using the specific primers for the archaeal *amo*A gene with Arch-amoAF and Arch-amoAR, the bacterial *amo*A gene with amoA-1F and amoA-2R, the denitrifier and ammonia oxidizer *nir*K gene with nirK-876F and nirK-1040R, the *nir*S gene with nirS-F and nirS-R, the denitrifier and ammonia oxidizer *nor*B gene with norB-F and norB-R, and the *nosZ* gene with nosZ-F and nosZ-R. All tests were conducted in triplicate with amplification efficiencies between 90% and 110% and correlation coefficients ($R^2$) above 0.98. More details on the primers and thermal profiles are listed in Supplementary Table S4.

### DNA extraction, metagenomic library sequencing, $N_2O$-related genome binning, taxonomic classification, functional annotation, and phylogenetic analyses

DNA was extracted from 100 regional-scale sediment cores by using the FastDNA Spin Kit for Soil (MP Biomedicals, Solon, OH, USA) according to the manufacturer's protocol. DNA quality and concentration were estimated using a NanoDrop 2000 Spectrophotometer (NanoDrop Technologies). Approximately 1.5 μg of extracted DNA (per sample) was used for metagenomic library preparation and subsequent sequencing on the Illumina PE150 platform (150-bp paired-end) with a sequencing depth of 10 G. Clean data were generated from raw metagenomic reads after low-quality nucleotides and reads with any ambiguous base calls were filtered out using Kneaddata (github.com/biobakery/kneaddata), and the data were subsequently quality-checked using FastQC (Babraham Bioinformatics, Babraham Institute, Cambridge, UK) (Supplementary Data S5). The quality-controlled clean data were used to obtain contigs using Megahit[46], while Bowtie2[47] and Samtools[48] were used for comparison and format conversion, respectively.

Metagenome binning was performed with contigs above 1500 bp using MetaWRAP (v 1.2.1)[49]. The obtained metagenome-assembled genomes (MAGs) were purified with RefineM[50] to remove contaminating contigs (Supplementary Data S6). The MAGs were quality-checked by using CheckM[51] and were dereplicated with dRep software[52], and 198 MAGs with a degree of completion greater than 75% and a degree of contamination less than 15% were selected for subsequent analysis[52]. The Quant_bins module in MetaWRAP (salmon algorithm)[53] was used to calculate the average relative abundance. The taxonomic affiliation of the MAGs was determined by GTDB-Tk v2.3.0[54]. Functional gene and protein annotation for the 198 MAGs was performed against the KEGG, NCyc[55], COG, and GO databases at an e value <1e$^{-5}$. According to taxonomy affiliation and functional gene and protein annotation, the MAGs containing *amo*, *hao*, or *nxr*AB genes and those containing the *nir*KS[56,57] gene were identified to belong to nitrifying and denitrifying bacteria, respectively. A phylogenetic tree of high-quality MAGs was generated by the maximum likelihood statistical method with the nearest-neighbor-interchange (NNI) ML heuristic method in MEGA 11 software[58].

### Analysis of environmental variables

The sediment $NH_4^+$, $NO_3^-$, $NO_2^-$, TOM, TN, and TP concentrations were determined[59,60]. The oxygen concentration and temperature in sediments were measured in situ by using a Pocket Oxygen Meter (FireStingGO2, PyroScience GmbH, Germany). The details are shown in the Supplementary Information.

### Statistical analysis

The mean values, *t*-tests, Spearman's correlations, and linear regression analyses were performed by using Statistical Product and Service Solutions 18.0 software (SPSS Inc., USA). The significance level was to α = 0.05 (*p* value ≤0.05).

### Reporting summary

Further information on research design is available in the Nature Portfolio Reporting Summary linked to this article.

## Data availability

The data generated in this study are provided within the article, Supplementary Information, and Supplementary Data files. Metagenomic sequencing data and metagenome-assembled genomes are available in the NCBI Sequence Read Archive (SRA) under the accession codes PRJNA943572 and PRJNA1031250, respectively.

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

## Acknowledgements

This research is financially supported by the Strategic Priority Research Program of the Chinese Academy of Sciences (Grant No. XDB0750400), the National Natural Science Foundation of China (42177063 and 92251304), and the Special Project on eco-environmental technology for peak carbon dioxide emissions and carbon neutrality (RCEES-TDZ-2021-20). The authors, Shanyun Wang and Guibing Zhu, gratefully acknowledge the Program of the Youth Innovation Promotion Association of the Chinese Academy of Sciences. The authors would like to thank all the persons for sampling.

## Author contributions

The project was conceived and led by G.Z. and S.W. S.W., B.L., L.Y., M.X., L.J., Y.Q., Y.J., Y.Z., G.A., J.M., M.W., and Y.Z. contributed to the sample chemical, molecular, and isotopic analysis. G.Z. and S.W. wrote the manuscript, and M.S.M.J., H.T., and Y.-G.Z. substantially contributed by commenting upon and revising it. All authors discussed and interpreted the results and contributed to the manuscript.

## Competing interests

The authors declare no competing interests.
