## [Peer Review File · Nature Communications]

Ammonium-derived nitrous oxide is a global source in streamsEditorial Note: Parts of this Peer Review File have been redacted as indicated to remove third-party material where no permission to publish could be obtained.

REVIEWER COMMENTS

Reviewer #1 (Remarks to the Author):

Dear authors:

Overall I found this an interesting paper but it seems too long for Nature Comm and could use a lot of clarification and reworking to make the story clear.

For example, many refs seem a bit irrelevant and many I thought were very helpful for your story were missing. (It's a huge field!) I think you may overstate how often anything relating to nitrifiers is ignored in the literature - I have provided quite a few examples.

I think you should restructure your paper so the crux of the problem is presented clearly in the intro (it's really hard to tell some of these pathways apart because they use the same genes under the same conditions) and then the most compelling evidence is provided first. You may also wish to cut some less compelling data or save for supplementary material. I didn't find the sed incubations as useful as the dual isotope and gene annotation data because the sed incubations couldn't differentiate the various pathways that ultimately go back to NH_4^+ oxidation. You need to be careful to differentiate NCD from NH_4^+ oxidation. Both use labelled NH_4^+ but the second is done by denitrifiers. The best term I could come up with was " NH_4^+ -limited N_2O production" but you may have a better one.

I have answered the prompt questions below and then given specific comments by line number.

- What are the noteworthy results?

I think the authors overstate the significance of their results as several studies have shown that NH_4^+ quickly makes its way into the the N cycle and can be denitrified. I don't get the general sense in the literature that NH_4^+ ultimately contributing to N_2O production is ignored. However, their dual isotope data from the Tang R is interesting. I'm not a microbiologist so having a hard time evaluating the gene annotation data (it strikes me as very odd that some denitrifying bacteria have no genes for N_2O reductase) but those results are also compelling.

- Will the work be of significance to the field and related fields? How does it compare to the established literature? If the work is not original, please provide relevant references.

I think this paper brings a lot of analyses (perhaps too many) to one problem but the individual pieces are not necessarily new.

Dual isotope methods for N_2O production in rivers are not new. This paper does it on sediment only in the lab, which is fine but may not reflect whole river production.

E.g. of dual isotope N₂O work in rivers (in situ):

<https://www.sciencedirect.com/science/article/pii/S0016703721002994>

The authors don't compare their results much to previous studies which is unfortunate. It would be interesting to compare when NH₄⁺-limited N₂O production dominates and hypothesize why.

- Does the work support the conclusions and claims, or is additional evidence needed?

The correlations between NH₄⁺ and N₂O production should also include dissolved O₂. Several studies have shown strong negative correlations between O₂ and N₂O concs in rivers.

The conclusion that most N₂O is from ammonia oxidation is misleading as some likely is from denitrifiers, coupled to ammonia oxidizers. Additionally, some seems to be from ammonia oxidizers running denitrifier proteins under similar conditions that denitrifiers thrive in.

- Are there any flaws in the data analysis, interpretation and conclusions? - Do these prohibit publication or require revision?

I didn't really understand if nos was included in two datasets for the statistical tests and if this is considered OK.

- Is the methodology sound? Does the work meet the expected standards in your field?

Methods for isotopic analysis, concs etc look fine. Dissolved O₂ should be measured.

- Is there enough detail provided in the methods for the work to be reproduced?

C₂H₂ method (to inhibit production of N₂ from N₂O) is not explained in methods. I didn't follow what the authors mean by "in-situ O₂" concentrations in the sed incubations.

Line-by-line comments:

Line 44: Do all 3 refs say up to 98%? Can you put in the range reported instead? I would think this could be very low, depending on substrate permeability. Also does "overlying surface waters" include the benthic zone sediment surface layer and/or biofilm on rocks etc? I would expect a lot of N₂O from anoxic/hypoxic areas which could include all of these, and not very much production in the water column itself (unless the water goes hypoxic).

Line 45: I think ref 8 likely does not have N₂O:N₂ models (it's about plant nutrition) and ref 12 might be

included in this list. I'm actually not sure what you mean to cite for Ref 8 - can you clarify? (I checked the index and there's nothing about denitrification, N₂O, etc.) Please check refs over carefully.

Line 46 - this sentence needs a reference (even if it's the same ref as next sentence). Clarify if this is true for all/most models? Just one specific paper?

Another mechanistic model that I don't see cited but will probably help you is here: Maavara et al. 2018 (<https://doi.org/10.1111/gcb.14504>). It does include nitrification.

47: "water surface area" - I assume this means the upper surface area, not the benthic surface area, but unclear how this relates to the hyporrheic zone and why it's so hard to measure water surface area globally? (The second part may not matter, I'm not just sure how this relates to the previous sentence.)

48: suggest removing ref 14 because it's about soils. I don't think confounding soil regimes and streams/rivers will help make your case.

48: "hardly considered" seems a bit harsh! The IPCC method assumes all NH₄⁺ is nitrified, producing N₂O, and half is then denitrified, producing more N₂O. Some papers such as Beaulieu et al. (2011) were not designed to examine nitrification, so suggest wording something like "though the IPCC methods include N₂O from nitrification (ref), some larger studies have focused on quantifying denitrification and N₂O emissions resulting from adding NO₃⁻ (e.g. Beaulieu et al., 2011)." Beaulieu et al. specifically state that there are few N₂O:NO₃⁻ yields for nitrification but use the IPCC method to say nitrification could be ~56% of N₂O production in streams.

50: You could mention that it is difficult to distinguish denitrifier and nitrifier-denitrification N₂O without analysing the microbes' genetic material as they both increase in low-O₂ settings. This is probably a major reason some studies don't tease them out (also, NH₄⁺ is often quite low!)

51: clarify who is not taking this into account as refs 16-18 are all about how hot spots are important. The problem (as I understand it!) is that heterogeneity is very difficult to model. Could reword as something like, "Hyporrheic N₂O production has been shown to be very heterogeneous (refs) but this is very difficult to account for in models and thus is often left out (refs)".

61: add refs to papers that do look at N₂O from Chinese streams

62: awkwardly worded, suggest something like "we compared these results to those measured globally" (or maybe list the areas covered). Suggest specifying temperate and tropical as I'm not seeing boreal polar areas represented in Fig 4. Also suggest clarifying if you looked at other sites in less detail (if not, why focus on the paper on the Chinese rivers?)

72: Did you measure dissolved oxygen in the water column and if so, did it show a negative trend with N₂O concentration, as in Rosamond et al. (2012) (<https://www.nature.com/articles/ngeo1556>)? This is

also seen in a few other sites cited in their paper.

82: remove "further"

86: You get into this later, but I think you need to clarify here what the labelling can and can't do, ie can't distinguish NCD and denitrification.

Could you also interpret these results as "nitrification occurs very rapidly, and then that NO₃⁻ (or NO₂⁻) is used in denitrification"? If more NH₄⁺ is added to the system, does that increase the rate of nitrification or is it limited by something else?

101: these papers do not appear to be about the same sites as yours. Please clarify "in different river systems" as I don't expect all riparian zones to be the same in terms of permeability, organic carbon availability, reactive N, microbial community etc.

101: wording it this way is potentially misleading as it's not necessarily the NH₄⁺ oxidation itself that produces the N₂O unless it's coming from the NH₄⁺ → NO₃⁻ suite of enzymes. (I see new research indicating that N₂O can come from hydroxylamine oxidation in AOA, here: <https://www.pnas.org/doi/10.1073/pnas.2220697120>. Previously, my understanding is that all N₂O from nitrifiers comes from them running nitrogen reductases (as a way to remove toxic NO₂⁻). Also it's hard to know what type of microbe is producing the N₂O with the data you've shown so far. Suggest something like, "N₂O was produced from NO₂⁻ or NO₃⁻ that originally came from labelled NH₄⁺"

104: I've always been of the opinion that (the way I do research), I can't differentiate NCD, ND or regular denitrification so they are lumped as "denitrification" in the environment, to me. I think you may see that reflected in the literature. I don't think too many people deny that NCD or ND are unimportant but they are hard to tease out.

I suggest rewording the paragraphs above as "it was clear the N₂O was produced from NH₄⁺ that rapidly cycled in the system" and then get into the genetics to further refine what was happening.

107: what are the nitrate and ammonia concentrations like in the riparian zone? If they are low (or dissolved O₂ is high), may not be much opportunity for denitrification (of any kind).

109: the second part of this sentence was unclear to me. I think you are using "open water sediment" and "hyporheic zone" interchangeably but it keeps making me think you have 3 types of sites instead of 2.

"higher potential of N₂O production" - does this mean that more copies of NOR genes were transcribed? Please clarify.

113: add references. I would assume this is true in pristine systems but perhaps not ones with a lot of NH_4^+ pollution. Certainly NH_4^+ is often much lower than NO_3^- in many oxic systems, suggest it's getting cycled quickly.

115: specify which denitrification gene(s) as there are several as it's a stepwise reaction. If you're looking at NIR, both NIRk and NIRs?

116: again, add ref that this is "traditionally" believed. It's important to clarify here that AOB could be using denitrification enzymes to produce N_2O . Probably needs the context that (at least in marine AOA) we now know that N_2O could also come from hydroxylamine oxidation. I see you mentioned this on line 120 - I suggest moving it earlier as it makes your results a lot easier to understand.

117: Is this the ratio of transcriptions? It's a bit confusingly worded but I take this to mean if there are more N_2O reductase transcriptions than NO reductase transcriptions, the microbes must be reducing N_2O from elsewhere? I'm not an expert in what controls the transcription of N_2O reductase. is it N_2O conc in the cell? low dissolved oxygen? Something else? I know the word limit is tight here but I think you need to explain this more.

123-126 this really clearly explains some of the things I thought were confusing/misleading about your work above. Suggest putting this in the intro so the reader can follow.

126: Are you counting the same nor genes in both pools (nitrifier pathway and lumped denitrifier pathways)? There's no way to tell the nor genes apart, right? Does a paired t test make sense if you're using the same data in both populations?

I think what's missing from this analysis is an estimate of how much N_2O you'd expect to get from HAO vs NOR for every molecule that goes through. Does anyone know this? What controls it? (So if you have really high HAO but only 1/10 000 molecules going through becomes N_2O , is that a significant pathway of N_2O production?)

135: suggest same edits as above to clarify you can't really tell which pathway is being used, just that you added $^{15}\text{N-NH}_4^+$ and it's ending up in N_2O . NCD is a type of "heterotrophic denitrification".

128: "at seasonal scale" sounds wrong to me. How about "on the seasonal scale"?

140: Again, do you have dissolved O_2 measurements? I would expect these to be highly negatively correlated to NH_4^+ .

142: I keep thinking about how to word this and maybe the best way is " N_2O production is NH_4^+ -limited" (??) It would be easier to think about what's going on if we saw NH_4^+ and NO_3^- (and dissolved O_2) in the samples.

154: change to "dual-isotope tracing". I have to look up to see if you added labelled NH₄ or just assumed some isotopic fractionation factors...

This paragraph is interesting! It seems strange to do it on a different river than the river with the more intensive sampling. I assume there was a logistical reason. But it makes me wonder if it's not better to do more intensive microbial genetics on samples from Tang R. Is the Tang R included in the regional results above? Might be worth bring it all the data from that river together so it's easier to follow.

160: why "should" be and how do you know they're not significantly different? My understanding is that isotopes will not distinguish NN and NCD but please clarify.

163: add refs for this idea - what recent papers? Should also but put in intro. I see you don't ref anything from Bradley Eyre - he has a lot of dual isotope N₂O papers (I think mostly in estuaries). E.g. <https://www.sciencedirect.com/science/article/pii/S0016703721002994>

165: I'm not a microbiologist but this seems more convincing to me than the gene transcriptions. I would suggest restructuring the paper so the data that are most conclusive (this and dual isotopes) are first and highlighted. The rest can always be put in supplementary material.

196: is it possible to have a denitrifying bacteria with no NO reductase? Does it just stop at NO and that's how much energy it makes? I have never heard of this. I hope the other reviewer is a microbiologist who knows how to evaluate this claim but it strikes me as so odd.

204: is this also true for microbes (your ref is about plants)?

208: change consuming to consumption. My understanding was that AOB don't really do nitrite reduction for energy but rather to remove toxic NO₂⁻. I'm happy to be proven wrong but please provide some references.

212: My understanding was that AOB are generally chemolithoautotrophs. You cite a paper for one unusual marine species that seems to also be able to do aerobic respiration. Reword or clarify if this is common for AOB and provide appropriate references.

214: You have not convinced me that AOB produce N₂O for energy. If they did, why not run denitrification to completion and make N₂? but they don't seem to have an N₂O reductase gene.

215: "In addition, the genetic diversity of nitrifying and215 denitrifying bacteria is different." Very unclear, just remove this sentence and say, "additionally, nitrifiers have the potential to produce high amounts of N₂O because they do not have N₂O reductase" or whatever. [Again, is it true that NO nitrifiers have N₂O reductase? Most? a few?]

227: is temporal scale, seasonal? Does the seaonality matter here? Unclear which methods you're using

on the global scale - this controls how strongly you can state your results!

246: I am not convinced this is the current opinion! Suggest rewording to indicate the NCD and ND use the same genes as denitrification (or are denitrification). There is some evidence HAO can produce N₂O in nitrifiers but unclear how much. You might need to coin a term here like "nitrifier-supplied denitrification" or something to combine NCD and ND if you can't tell them apart.

Suggest comparing these results to IPCC estimates, e.g. all reactive N is nitrified, half is denitrified. Using their EF5s how does this compare to your results? Nothing wrong with the Nevison paper but it is 23 years old and has been revisited, e.g. by Maavara et al. (2018).

315: is in-situ O₂ atmospheric or what was measured in the field when the sediment was collected? O₂ will have a big role to play in whether nitrification or denitrification dominates.

317: Not sure what "completely designed at random vials" means. Please clarify.

Reviewer #2 (Remarks to the Author):

Noteworthy results: 1) evidence that "nitrifying bacteria contain higher abundances of N₂O production-related genes than denitrifying bacteria"; 2) "ammonia oxidation was the most important hyporheic N₂O source"; 3) "These findings differ from the prevailing opinion that heterotrophic denitrification is the main contributor to N₂O in the riverine hyporheic zone."

Significance: high, paradigm shifting.

Evidence: the evidence provided supports the claims and conclusions well.

Methods/Analysis: sound and repeatable.

Comments:

There are a few, minor typos throughout. Please revise with the copy editor.

L117: 2.2:3.1 is a unique way to display a ratio. Why not 0.7:1?

Conclusion: overall, very good work. Worthy of publication.

Reviewer #3 (Remarks to the Author):

Summary: This study aims to investigate the microbial sources and mechanisms responsible for N₂O production in riparian sediments. Specifically, investigators used various methods including omics, tracer experiments, and geochemistry to parse N metabolism at site-level (Xiaoqinghe River), regional-level (Baiyangdian river network), and global scales. Key findings highlight ammonia as a more important predictor of N₂O production over nitrate, in contrast to a widely held modeling assumption.

Major comments:

1. Data deposition- The manuscript does not contain a data availability statement and should not be considered for publication until this is included. Specifically, this manuscript should not be accepted for publication until raw metatranscriptomic and metagenomic reads, along with the derived genome bins are deposited into a repository such as NCBI or equivalent. Additionally, raw numbers for tracer or physio-chemical characteristics should be reported as an additional data file rather than as averages in flat pdf tables.

2. Methodological concerns

a. Details on numbers of samples seem to be left out throughout the entire manuscript and should be added at least in the methods. Specific areas of improvement include:

- Line 263- how many sediment cores?
- Line 266 five sites along each river for the regional study- how many rivers? Were these also collected in triplicate?
- In general, each study (site, regional, and global) should have a sentence that states exactly how samples were collected (e.g., in triplicate, where in the river they were collected from, sites per river), the total number of samples generated, and the number of rivers sampled. It may also be nice to have a table of this information per site, regional, and global studies along with the analyses (e.g., was metagenomic performed at every site?) performed on each scale.
- Which analyses were performed on which samples- the N-tracer analyses state it is just the site-specific samples, the N-O tracer studies do not state which samples these were done on, and for the global studies Supplementary table has cited N₂O fluxes. Were the global samples not analyzed but taken from literature values?
- Line 326- how many samples were RNA extracted from?
- How many metagenomes are included in this study? What is the depth of sequencing?

b. For the sampling and analysis of global rivers, more information should be provided on the methods. Were all these global rivers sampled by the same team? Where were these samples analyzed? On line 272 it states these samples were immediately transferred back to the lab- which lab, were they on ice? Were these shipped to a specific lab where all analyses were done? What time of year were the global river sites sampled? Were any analyses done on the global sediments, as no data or methods specific to global rivers is reported in the supplementary tables- it appears to just be a citation table? It is very unclear what was done in this study for the global aspect.

c. How were samples collected- the manuscript just states that sediment cores (line 263) were taken? E.g., what type of corer? Scratch this- I see this information is on lines 278. Consider moving up and stating this information for all studies (site, regional, global), as this coring information seems to only be

stated for the site-specific study.

d. No details were provided on DNA extraction or metagenomic sequencing, the manuscript states “filtering was carried out on the original metagenomic data...” with no citation. What study was the original metagenomic data from? Why are analyses (e.g., binning, assembly) being redone here? Either cite the original data publication with DNA extraction, library prep, and sequencing methods or include details in this manuscript.

e. Annotation of key nitrogen genes- Functional nitrogen genes have high similarity to functional genes of other processes, thus without stringency in annotation, these genes could easily be misclassified if additional analyses were not performed. For example, ammonia monooxygenase (amo) is a homolog with methane monooxygenase (pmo), while nxr and nar are homologs. Please provide additional annotation information on how these annotations were done. What bitscore/e-value cutoffs were used (line 354)? Were additional phylogenetic analyses/trees done on key functional genes? Was taxonomy of bin taken into consideration? Which subunits or other genes were considered?

f. Figure 3a has a genome phylogenetic tree. Please include methods for how this tree was generated, and the accession numbers for the isolate genomes.

g. Line 351 states there were 198 dereplicated genome bins used for subsequent analysis, however the MAG table includes less than 30 MAGs (Tables S5-S6). Where is the information for the rest of the MAGs? Are the ones in the tables only N cycling? I suggest including a supplementary data file for all MAGs (n=198) with all the relevant reporting information for MIMAGs outlined in Bowers, et al. For example, information on rRNAs and tRNAs should also be included to be in compliance with genome reporting standards.

h. How were the 28 global streams determined to be agricultural? What metrics needed to be met to be included in this study as a global agricultural stream?

3. In general, the supplementary tables seem to be incorrectly referenced throughout the manuscript, but especially in the methods, making the manuscript hard to review in its current form. Please revise. Specific examples are listed below but these table references should be revised throughout the manuscript.

a. Line 272, supplementary table 4 should be about the global study but is instead spearman correlation table at the regional scale.

b. Line 267- should be something on the regional scale study but is table on site scale investigation.

c. Line 238- states information on sampling sites can be found in Supplementary Table S5, but S5 is a table of bins.

d. Line 172- Supplementary Tables S5-6 only have information for less than 30 MAGs, not 111 or 198 as stated.

4. Global claims throughout manuscript are not warranted based on data and should be toned down to fit the results presented in manuscript. See specific examples below, but statements should be revised throughout the manuscript.

a. Line 233- “Ammonia oxidation pathways are the dominant hyporheic N₂O source in streams” – the streams surveyed in the global study are all agricultural, low-order streams which should be clarified in this sentence.

b. Lines 35-36- while ammonia loading might be more important for low order agricultural streams, nitrate may be important in others.

Minor comments:

- Remove grey highlight text on lines 258-259- "Its upper and mid regions"
- Line 259- one should be first if talking about stream order
- Line 342- citation for Kneaddata should be included
- Line 342- what is citation 39 citing? It appears to be for a tool called PRINSEQ for quality control of sequencing data- was that used or Kneaddata?
- Line 343- add the word "while" before Bowtie2
- Line 346- "15,00" should be "1,500"
- Line 348- "Checkm" should be "CheckM"
- Line 349- "de-redundant" should "dereplicated"
- Line 353- what database version was used for GTDB?

Responses to Reviewer Comments

We thank all three reviewers very much for their professional and insightful comments, which will help in the dissemination of our results to a wide scientific community. Meanwhile, we want to provide special acknowledgement to the editor and reviewers for your careful thinking and kind patient on this long-time modification of our manuscript, as the first author underwent a major surgery. It is our sincere hope that the improvement of our manuscript has been satisfying and that our contribution is now ready for discussion in the public domain.

Words labeled *in italic* are the reviewers' comments, and labeled with blue color is the response.

Reviewer #1

Overall I found this an interesting paper but it seems too long for Nature Comm and could use a lot of clarification and reworking to make the story clear.

For example, many refs seem a bit irrelevant and many I thought were very helpful for your story were missing. (It's a huge field!) I think you may overstate how often anything relating to nitrifiers is ignored in the literature - I have provided quite a few examples.

Response:

We are deeply appreciative of your endorsement of our manuscript and incredibly grateful for your constructive comments.

As you suggested, we have meticulously rejudged the structure of our manuscript, diligently scrutinized the citations enveloping the whole manuscript, excised unrelated references, and cited the most pertinent and pivotal references within the expanse of the revised manuscript. Meanwhile, we have modified the imprecise statements, i.e. "nitrifiers is ignored in the literature". These alterations have consequently augmented the accuracy and readability of our manuscript, creating a more polished and refined piece. We cordially invite you to delve into the revised manuscript and the detailed responses provided below.

I think you should restructure your paper so the crux of the problem is presented clearly in the intro (it's really hard to tell some of these pathways apart because they use the same genes under the same conditions) and then the most compelling evidence is provided first. You may also wish to cut some less compelling data or save for supplementary material. I didn't find the sed incubations as useful as the dual isotope and gene annotation data because the sed incubations couldn't differentiate the various pathways that ultimately go back to NH₄⁺ oxidation. You need to be careful to differentiate NCD from NH₄⁺ oxidation. Both use labelled NH₄⁺ but the second is done by denitrifiers. The best term I could come up with was "NH₄⁺-limited N₂O production" but you may have a better one.

Response:

First of all, we would like to express our sincere gratitude for your professional comment. Based on other representative literatures (Zhu et al., Proc. Natl. Acad. Sci. U.S.A. 2013; Kool et al., Eur. J. Soil Sci. 2010; Kool et al., Soil Biol. Biochem. 2011), we have clarified the ammonia oxidation and heterotrophic denitrification processes to NH₄⁺-derived and NO₃⁻-derived N₂O production processes, respectively. With ambient NH₄⁺ as substrate, the NH₄⁺-derived N₂O production process includes nitrifier nitrification (NN), nitrifier denitrification (ND), and nitrification-coupled denitrification (NCD) pathways. With ambient NO₃⁻ as substrate, the NO₃⁻-

derived process represents the lumped heterotrophic denitrification (HD) pathway. **We have used the new terms, NH₄⁺-derived N₂O production and NO₃⁻-derived N₂O production, in responses to reviewer comments and in new versions of the manuscript.** We thought the new terms were more precise.

Secondly, thank you for pointing out some unclear scientific issues and providing guidance on how to clarify research highlights. In fact, due to the increase in riverine N₂O emission, we have studied the microbial sources and mechanisms of hyporheic N₂O production in stream ecosystems, especially in low-order agricultural streams. This is largely unknown and poorly modelled. Based on isotope ¹⁵N-¹⁸O/¹⁵N, RT-qPCR, C₂H₂-inhibitor, and metagenomic technologies, we found that NH₄⁺-derived N₂O production (NN, ND, and NCD) is the dominant hyporheic N₂O sources (69.6 ± 2.1%) in global streams, rather than NO₃⁻-derived N₂O production (HD). ND, as an important pathway of N₂O production, has long been ignored but has been widely reported recently (Zhu et al., Proc. Natl. Acad. Sci. U.S.A. 2013; Kool et al., Eur. J. Soil Sci. 2010; Kool et al., Soil Biol. Biochem. 2011). Thus, global models of riverine ecosystems need to better represent NH₄⁺-derived N₂O production in order to accurately estimate and predict riverine N₂O emissions. Taken together, these factors highlight the importance of N management, especially NH₄⁺ fertilizer, in agricultural field. We have introduced the main research highlights more clearly **in the introduction of the revised manuscript (Lines 42-57).**

Thirdly, we are honored to receive your approval of the ¹⁵N-¹⁸O dual isotope and gene annotation data. As for your comment that “¹⁵N tracing incubation of *semi-in-situ* sediment core couldn't differentiate NCD from NH₄⁺ oxidation”, this is a misunderstanding caused by unclear writing. In fact, the ¹⁵N tracing incubation of *semi-in-situ* sediment core was designed by the first author Dr. Wang for this study. The ¹⁵N-NH₄⁺ and ¹⁵N-NO₃⁻ were added to trace NH₄⁺-derived and NO₃⁻-derived N₂O production, respectively. The sediment core was incubated under the intact sediment structure and the *in-situ* oxygen and temperature conditions, closer to actual production conditions than slurry incubation (Trimmer et al., Mar. Ecol. Prog. Ser. 2006; Zhu et al., Nature Geo. 2013; Wang et al., ISME J. 2020). Although the ¹⁵N tracing incubation of *semi-in-situ* sediment core cannot distinguish the NN, ND and NCD pathways of NH₄⁺-derived N₂O without the addition of ¹⁸O tracers, it can accurately determine the rates and contributions of NH₄⁺-derived and NO₃⁻-derived N₂O, which is similar to the C₂H₂ inhibitor method. Due to the limitations in *in-situ* field of widely spatiotemporal scales, we selected different but suitable research methods at different scales. The ¹⁵N tracing of the *semi-in-situ* sediment core results (site scale) and ¹⁵N-¹⁸O dual-isotope tracing results (regional scale) showed consistency with the corresponding results of C₂H₂ inhibitor results (site and regional scale). Therefore, C₂H₂ inhibitor method was used on all scale studies.

Additionally, we have removed the detailed methods to the **Supplementary Information** to shorten the manuscript length. Thanks again for your positive comments and valuable suggestions.

References:

- Zhu, X., Burger, M., Doane, T. A., & Horwath, W. R. Ammonia oxidation pathways and nitrifier denitrification are significant sources of N₂O and no under low oxygen availability. *Proc. Natl. Acad. Sci. U.S.A.* **110(16)**, 6328-6333 (2013).
- Kool, D. M., Wrage, N., Zechmeister-Boltenstern, S., et al. Nitrifier denitrification can be a source of N₂O from soil: a revised approach to the dual-isotope labelling method. *Eur. J. Soil Sci.* **61(5)**, 759-772 (2010).
- Kool, D. M., Dolfing, J., Wrage, N., & Van Groenigen, J. W. Nitrifier denitrification as a distinct and significant source of nitrous oxide from soil. *Soil Biol. Biochem.* **43(1)**, 174-178 (2011).
- Trimmer, M., Risgaard-Petersen, N., Nicholls, J. C., & Engstroem, P. Direct measurement of anaerobic ammonium oxidation (anammox) and denitrification in intact sediment cores. *Mar. Ecol. Prog. Ser.* **326**, 37-47 (2006).
- Zhu, G. B., Wang S. Y., Wang W., Wang Y., Zhou L., & Jiang B., et al. Hotspots of anaerobic ammonium oxidation at land-freshwater interfaces. *Nature Geo.* **6(2)**, 103-107 (2013).
- Wang, S. Y., Zhu, G., Zhuang, L. et al. Anaerobic ammonium oxidation is a major N-sink in aquifer systems around the world. *ISME J.* **14**, 151-163 (2020).

I have answered the prompt questions below and then given specific comments by line number.

- What are the noteworthy results?

I think the authors overstate the significance of their results as several studies have shown that NH_4^+ quickly makes its way into the the N cycle and can be denitrified. I don't get the general sense in the literature that NH_4^+ ultimately contributing to N_2O production is ignored. However, their dual isotope data from the Tang R is interesting. I'm not a microbiologist so having a hard time evaluating the gene annotation data (it strikes me as very odd that some denitrifying bacteria have no genes for N_2O reductase) but those results are also compelling.

Response:

We would like to thank the reviewer for the recognition of our research on the ^{15}N - ^{18}O double-isotope tracing and metagenomic gene annotation. We came up with many solutions to the problems we encountered during the research process. We feel very relieved and satisfied when the results can be recognized by experts.

Based on your comments and previous literatures, we have revised the introduction to clarify the science and readability of our manuscript:

Regarding global change model, “Current process-based N_2O models simply use the $\text{N}_2\text{O}/\text{N}_2$ ratio to represent the N_2O production rate during denitrification (Yao et al., Nat. Clim. Change. 2020). Model parameters associated with microbially mediated hyporheic N_2O production are poorly represented (Yao et al., Nat. Clim. Change. 2020; Zhu et al., Proc Natl. Acad. Sci. U.S.A. 2013). Though the IPCC methods include N_2O from nitrification (Maavara et al., Global Change Biol. 2019), previous studies have focused on quantifying denitrification and N_2O emissions resulting from adding nitrate (NO_3^-) (Beaulieu et al., Proc Natl. Acad. Sci. U.S.A. 2011). Furthermore, it is difficult to distinguish nitrifier and/coupled denitrifier denitrification N_2O as they occur using the homologous genes under the similar conditions (Zhu et al., Proc Natl. Acad. Sci. U.S.A. 2013). Hyporheic N_2O production has been shown to be very heterogeneous (Biddulph, 2015; Kiel & Cardenas Nat. Geosci. 2014; Boulton, et al., Annu. Rev. Ecol. Syst. 1998), but this is extremely difficult to account for in models and therefore is often left out (Maavara et al., Global Change Biol. 2019). Currently, the microbial mechanisms underlying N_2O production in the hyporheic exchange zone are largely unknown, which may significantly influence and potentially severely underestimate the global N_2O budget calculations. (**Lines 48-57**)” It is in response to this research gap that we have reached a consensus with Prof. Tian, the international expert in the field of climate change and ecosystem (Tian et al., Nature 2016, 2020; Yao et al., Nat. Clim. Change 2020; Lu et al., Nature Food 2022).

Additionally, you pointed out that some denitrifying bacteria do not have genes for N_2O reductase in the metagenome-assembled genomes (MAGs). In fact, denitrifying bacteria without N_2O reductase are possible. Similar results also have been reported in studies on N_2O hot moment in peatlands by our team (Wang et al., ISME J. 2023), on anammox granules in a wastewater treatment system by distinguished microbiologist Professor Mike Jetten and his partners (Speth et al., Nat. Commun. 2016), and on truncated denitrifiers (lacking one or more denitrification genes) and N_2O cycling in tundra soils (Pessi et al., bioRxiv 2020).

Denitrification is a series of enzymatic steps in which NO_3^- is sequentially reduced to NO_2^- , NO , N_2O , and N_2 . Denitrification traits are found in a wide range of archaea, bacteria, and fungi, most of which are facultative anaerobes that convert N oxides as electron acceptors under anoxic conditions (Zumft et al., 1997). To date, only about one-third of the genomes of cultured denitrifiers sequenced encode the full set of enzymes required for complete denitrification (Graf et al., 2014; Pessi et al., bioRxiv 2020). Denitrification has been demonstrated to be a modular process, with some microorganisms able to completely reduce NO_3^- to N_2 , while others can only perform one or a subset of steps (Zhang et al., 2023; Fuchsman et al., 2017; Graf et al., 2014; Wallenstein et al., 2006; Bertagnolli et al., 2020).

Fig. 4: Metabolic reconstruction and features of N₂O production- and reduction-related microorganisms identified in seasonally frozen peatland. (Wang et al., *ISME J.* 2023)

89% bins with no NO reductase, except bins 46 and 50

Figure 4: Schematic overview of N conversions in the Olburgen PNA reactor. (Speth et al., *Nat. Commun.* 2016)

④ CFX1 with no NO reductase

Fig. 2 Metabolic potential for denitrification in metagenome-assembled genomes 152 (MAGs) from tundra soils. (Pessi et al., *bioRxiv.* 2020)

few coexist of NO₂⁻ reductase and NO reductase

References:

- Yao, Y. Z., Tian, H. Q., Shi, H., Pan, S. F., Xu, R. T., Pan, N. Q., & Canadell, J. G. Increased global nitrous oxide emissions from streams and rivers in the Anthropocene. *Nat. Clim. Change*. **10(2)**, 138-139 (2020).
- Maavara, T., Lauerwald, R., Laruelle, G. G., Akbarzadeh, Z., Bouskill, N. J., Van Cappellen, P., & Regnier, P. Nitrous oxide emissions from inland waters: Are IPCC estimates too high? *Global Change Biol.* **25**, 473–488 (2019).
- Beaulieu, J. J., Tank, J. L., Hamilton, S. K., Wollheim, W. M., Hall, R. O., Mulholland, P. J., Thomas, S. M. Nitrous oxide emission from denitrification in stream and river networks. *Proc Natl. Acad. Sci. U.S.A.* **108(1)**, 214-219 (2011).
- Zhu, X., Burger, M., Doane, T. A., & Horwath, W. R. Ammonia oxidation pathways and nitrifier denitrification are significant sources of N₂O and NO under low oxygen availability. *Proc. Natl. Acad. Sci. U.S.A.* **110(16)**, 6328-6333 (2013).
- Biddulph, M. Hyporheic Zone: In Situ Sampling. *Geomorphological Techniques. British Society for Geomorphology* (2015).
- Kiel, B. A., & Cardenas, M. B. Lateral hyporheic exchange throughout the Mississippi River network. *Nat. Geosci.* **7(6)**, 413-417 (2014).
- Boulton, A. J., Findlay, S., Marmonier, P., Stanley, E. H., & Valett, H. M. The functional significance of the hyporheic zone in streams and rivers. *Annu. Rev. Ecol. Syst.* **29**, 59-81 (1998).
- Tian, H., Lu, C., Ciais, P., Michalak, A. M., Canadell, J. G., & Saikawa, E., et al. The terrestrial biosphere as a net source of greenhouse gases to the atmosphere. *Nature* **531(7593)**, 225 (2016).
- Tian, H. Q., Xu, R. T., Canadell, J. G., Thompson, R. L., Winiwarter, W., Suntharalingam, P., Davidson, E. A., Ciais, P., Prather, M. J., Regnier, P., et al. A comprehensive quantification of global nitrous oxide sources and sinks. *Nature* **586(7828)**, 248-256 (2020).
- Lu, C., Yu, Z., Hennessy, D. A., Feng, H., Tian, H., & Hui, D. Emerging weed resistance increases tillage intensity and greenhouse gas emissions in the us corn-soybean cropping system. *Nature Food* (4), 3, 266–274 (2022).

- Wang, X. M., Wang, S. Y., Yang, Y. H., Tian, H. Q., Jetten, M. S. M., Song, C. Q., & Zhu, G. B. Hot moment of N₂O emissions in seasonally frozen peatlands. *ISME J.* **17**, 792–802 (2023).
- Speth, D., in't Zandt, M. H., Guerrero-Cruz, S., Dutilh, B. E., & Jetten, M. S. M. Genome-based microbial ecology of anammox granules in a full-scale wastewater treatment system. *Nat. Commun.* **7**, 11172 (2016).
- Pessi, I. S., Viitamäki, S., Rasimus, E. E., Delmont, T. O., Luoto, M., & Hultman, J. Truncated denitrifiers dominate the denitrification pathway in tundra soil metagenomes. *bioRxiv*. 419267 (2020).
- Zumft, W. G. Cell biology and molecular basis of denitrification. *Microbiol. Mol. Biol. Rev.* **61**, 533–616 (1997).
- Graf, D. R. H., Jones, C. M., & Hallin, S. Intergenomic comparisons highlight modularity of the denitrification pathway and underpin the importance of community structure for N₂O emissions. *PLoS one* **9**, e114118 (2014).
- Zhang, I. H., Sun, X., Jayakumar, A. et al. Partitioning of the denitrification pathway and other nitrite metabolisms within global oxygen deficient zones. *ISME Commun.* **3**, 76 (2023).
- Fuchsmann, C. A., Devol A. H., Saunders J. K., McKay C., Rocap G. Niche partitioning of the N cycling microbial community of an offshore oxygen deficient zone. *Front Microbiol.* **8**: 2384 (2017).
- Wallenstein, M. D., Myrold, D. D., Firestone, M., & Voytek, M. Environmental controls on denitrifying communities and denitrification rates: insights from molecular methods. *Ecol. Appl.* **16**, 2143–2152 (2006).
- Bertagnolli, A. D., Konstantinidis, K. T., & Stewart, F. J. Non-denitrifier nitrous oxide reductases dominate marine biomes. *Environ Microbiol Rep.* **12**: 681–92 (2020).

- Will the work be of significance to the field and related fields? How does it compare to the established literature? If the work is not original, please provide relevant references.

I think this paper brings a lot of analyses (perhaps too many) to one problem but the individual pieces are not necessarily new.

Dual isotope methods for N₂O production in rivers are not new. This paper does it on sediment only in the lab, which is fine but may not reflect whole river production.

E.g. of dual isotope N₂O work in rivers (in situ):

<https://www.sciencedirect.com/science/article/pii/S0016703721002994>

Response:

For investigating the hyporheic N₂O source and rate, this study used complementary methods such as ¹⁵N-¹⁸O dual-isotope tracing, ¹⁵N tracing of the *semi-in-situ* sediment core, C₂H₂ inhibitor, quantitative reverse transcription PCR (RT-qPCR), and metagenomic sequencing and assembly analysis on a global spatiotemporal scale. We obtained the solid conclusion that NH₄⁺-derived N₂O production (NN, ND, and NCD) is the dominant hyporheic N₂O sources (69.6 ± 2.1%) in global streams, rather than NO₃⁻-derived N₂O production (HD).

Due to the limitations in *in-situ* field of widely spatiotemporal scales, we selected different but suitable research methods at different scales. For example, at site-scale study, we extracted transcriptomic RNA and performed ¹⁵N tracing on the *semi-in-situ* sediment core to analysis the transcriptional gene abundance and the *semi-in-situ* N₂O production rate. However, on global-scale study, the implementation of the above methods is inconvenient due to the long-distance transportation of samples. For the regional-scale study, we used the ¹⁵N-¹⁸O dual-isotope tracing. But due to the excessive addition of ¹⁸O-H₂O, it is inappropriate for the site-scale sediment cores. The ¹⁵N tracing of the *semi-in-situ* sediment core results (site scale) and ¹⁵N-¹⁸O dual-isotope tracing results (regional scale) showed consistency with the corresponding results of C₂H₂ inhibitor results (site and regional scale). Therefore, C₂H₂ inhibitor method was used on the global-scale study.

You also pointed out the lack of innovation in methods. We are sorry for not writing clearly. In fact, both isotope tracing methods were designed or developed by our team. See below for details:

- ¹⁵N tracing of the *semi-in-situ* sediment core

The ¹⁵N tracing incubation of *semi-in-situ* sediment core was designed by the first author Dr. Wang. The

$^{15}\text{N-NH}_4^+$ and $^{15}\text{N-NO}_3^-$ were added to trace NH_4^+ -derived and NO_3^- -derived N_2O production, respectively. The sediment core was incubated under the intact sediment structure and the *in-situ* oxygen and temperature conditions, closer to actual production conditions than slurry incubation. In contrast, the slurry incubation completely disrupts the natural gradients of substrates and redox in soils (Trimmer et al., Mar. Ecol. Prog. Ser. 2006), thereby accelerating the conduction of the medium in the pore water. Without the addition of ^{18}O tracers, it is not able to further distinguish the NH_4^+ -derived process into NN, ND and NCD pathways. However, this method could accurately measure the *semi-in-situ* N_2O production rate and distinguish the contributions of NH_4^+ -derived and NO_3^- -derived processes.

- $^{15}\text{N-}^{18}\text{O}$ dual-isotope tracing

The $^{15}\text{N-}^{18}\text{O}$ dual-isotope tracing method was improved by our team compared with the method published by Wrage et al., 2005; Kool et al., 2010 2011; and Zhu et al., 2013, ensuring incubation under *in situ* oxygen and temperature conditions. Therefore, the results detected by improved method were closer to actual production conditions than the original method. The improved $^{15}\text{N-}^{18}\text{O}$ dual-isotope tracing method has recently been reported in the studies of our team (Wang et al., ISME J. 2023; Jiang et al., Global Change Biol. 2023).

Zhu et al., Proc. Natl. Acad. Sci. U.S.A. 2013

Wells & Eyre Geochim. Cosmochim. Acta 2021

As for the paper you mentioned about the dual isotope N_2O in *in situ* rivers, to be honest, this is a great study. Professor Bradley Eyre is an outstanding biogeochemist. He and his team studied the flow of carbon and nitrogen through coastal ecosystems, focusing on global change issues, such as eutrophication, climate change, and greenhouse gases. Wells & Eyre (2021) studied on the stream flow dynamics that regulates the balance between biological NO_3^- and N_2O production using *in-situ* stable isotopes ($\delta^{15}\text{N-NO}_3^-$ and $\delta^{18}\text{O-NO}_3^-$) and Modelling. Conductivity, $\delta^{18}\text{O-H}_2\text{O}$, and ^{222}Rn were used to constrain surface water – groundwater mixing. They did not analyze microbial composition and activity directly, but emphasized the importance of nitrification, rather than denitrification through theoretical modelling, even in strongly heterotrophic stream. We are pleased to see consistent results based on different research ideas, methods and study sites. To its credit, this method is a good way to track water migration in the hyporheic zones of small streams where surface water and groundwater exchange intensively. This *in situ* isotope tracing method has obvious advantages, such as: i) no additional isotopes introduced into nature, ii) Long term tracking, iii) simple experimental procedures, iv) close to true natural value.

Nevertheless, in our study, the isotope labeling method was used for the following reasons: i) Our team has a long-term study on isotope labelling technology, which has been successfully used in various sediments and soils, including lake (Zhu et al., 2013), river (Wang et al., 2012), pond (Zhu et al., 2015), paddy field (Zhu

et al., 2011; Qin et al., 2023), constructed wetland (Zhu et al., 2012), groundwater (Wang et al., 2020), seasonal peatland (Wang et al., 2023) and upland soils (Zhu et al., 2018); ii) Avoiding long-term continuous monitoring and complex environmental models; iii) Directly measuring the activity of functional microorganisms; iv) Suitable for offline measurement at various scales. Therefore, isotope labeling methods are more suitable for multiple-scale investigation, such as this study.

Thanks for pointing this out and leading us to re-examine the highlights of our research. In the revised manuscript, we have cited this excellent work by Professor Eyre's team as well as other relevant publications (**Line 165**). Meanwhile, we have also rewritten both isotope tracing methods to clarify their innovation (**Lines 85-86, 156, 298, 314**).

References:

- Trimmer, M., Risgaard-Petersen, N., Nicholls, J. C., & Engström, P. Direct measurement of anaerobic ammonium oxidation (anammox) and denitrification in intact sediment cores. *Aust. J. Agr. Res.* **326(4)**, 37-47 (2006).
- Wrage, N., Groenigen, J., Oenema, O. & Baggs, E. A novel dual-isotope labelling method for distinguishing between soil sources of N₂O. *Rapid Commun. Mass Spectrom.* **19**, 3298–306 (2005).
- Kool, D. et al. Nitrifier denitrification can be a source of N₂O from soil: a revised approach to the dual-isotope labelling method. *Eur. J. Soil Sci.* **61**, 759–772 (2010).
- Kool, D. M., Dolfing, J., Wrage, N. & van Groenigen, J. W. Nitrifier denitrification as a distinct and significant source of nitrous oxide from soil. *Soil Biol. Biochem.* **43**, 174–178 (2011).
- Zhu, X., Burger, M., Doane, T. A., & Horwath, W. R. Ammonia oxidation pathways and nitrifier denitrification are significant sources of N₂O and no under low oxygen availability. *Proc. Natl. Acad. Sci. U.S.A.* **110(16)**, 6328–6333 (2013).
- Wells, N. S. & Eyre, B. D. Flow regulates biological NO₃⁻ and N₂O production in a turbid sub-tropical stream. *Geochim. Cosmochim. Acta* **306**, 124-142 (2021).
- Wang, S., Zhu, G., Peng, Y., Jetten, M. S. M., & Yin, C. Anammox bacterial abundance, activity, and contribution in riparian sediments of the pearl river estuary. *Environ. Sci. Technol.* **46(16)**, 8834-8842 (2012).
- Wang, S., Wang W., Liu L., Zhuang L., Zhao S., & Su Y., et al. Microbial nitrogen cycle hotspots in the plant-bed/ditch system of a constructed wetland with N₂O mitigation. *Environ. Sci. Technol.* **52**, 6226–6236 (2018).
- Zhu, G., Wang S., Wang Y., Wang C., Risgaard-Petersen N., & Jetten M. S., et al. Anaerobic ammonia oxidation in a fertilized paddy soil. *ISME J.* **5(12)**, 1905-1912 (2011).
- Zhu, G., Wang S., Feng X., Fan G., Jetten M. S. M., & Yin C. Anammox bacterial abundance, biodiversity and activity in a constructed wetland. *Environ. Sci. Technol.* **45(23)**, 9951-8 (2011).
- Zhu, G., Wang S., Wang W., Wang Y., Zhou L., & Jiang B., et al. Hotspots of anaerobic ammonium oxidation at land–freshwater interfaces. *Nature Geo.* **6(2)**, 103-107 (2013).
- Zhu, G, Wang S, Zhou L, Wang Y, Zhao S, Xia C, et al. Ubiquitous anaerobic ammonium oxidation in inland waters of China: an overlooked nitrous oxide mitigation process. *Sci. Rep.* **5**, 17306 (2015).
- Zhu, G., Wang S., Li Y., Zhuang L., Zhao S., Kuypers M. M. M., & Jetten M. S. M., et al. Microbial pathways for nitrogen loss in an upland soil. *Environ. Microbiol.* **20(5)**, 1723–1738 (2018).
- Qin, Y., Wang, S. Y., Wang, X. M., Liu, C. L., & Zhu, G. B. Contribution of ammonium-induced nitrifier denitrification to N₂O in paddy fields. *Environ. Sci. Technol.* **57(7)**, 2970–2980 (2023).
- Wang, X. M., Wang, S. Y., Yang, Y. H., Tian, H. Q., Jetten, M. S. M., Song, C. Q., & Zhu, G. B. Hot moment of N₂O emissions in seasonally frozen peatlands. *ISME J.* **17**, 792–802 (2023).

The authors don't compare their results much to previous studies which is unfortunate. It would be interesting to compare when NH_4^+ -limited N_2O production dominates and hypothesize why.

Response:

In the revised manuscript, “ NH_4^+ -derived N_2O production” and “ NO_3^- -derived N_2O production” are used instead of “Ammonia oxidation” and “Heterotrophic denitrification”, respectively.

We can speculate on the nature of some of the discussions written. In 2013, Zhu Xia's research found that as long as there is oxygen, even if it is very low (0.5%), the NH_4^+ -derived process (named ammonia oxidation process in her study) will contribute a large amount of N_2O , even more than the NO_3^- -derived (named denitrification in her study) process. NO_3^- -derived process produced all N_2O at 0% O_2 . The findings were published in PNAS (Zhu et al., Proc. Natl. Acad. Sci. U.S.A. 2013). Based on this finding, we hypothesized that NH_4^+ -derived N_2O production may be the source of large amounts of hyporheic N_2O in streams, which account for 85% of global riverine N_2O emissions.

Based on site-scale, regional-scale, and global-scale investigations, as well as complementary methods such as isotopic ^{15}N - ^{18}O , ^{15}N tracing of the *semi-in-situ* sediment core, RT-qPCR, and metagenomic assembling and binning analysis for a wide range of sample types and temperature zones globally, we discovered NH_4^+ -derived pathways (NN, ND, and NCD) are the dominant hyporheic N_2O sources ($69.6 \pm 2.1\%$) in global streams, rather than NO_3^- -derived pathway (HD).

There is increasing evidence that NH_4^+ -derived N_2O production contributes more considerably to N_2O production than NO_3^- -derived N_2O production, especially at low O_2 levels (see references below). Therefore, we believe that an important next step is to explain why the NH_4^+ -derived N_2O production, rather than NO_3^- -derived N_2O production, are the dominant N_2O sources in streams.

As you mentioned above, denitrifying genes, *nir* and *nor*, are involved in ND, NCD, and HD pathways, resulting in the inability to distinguish which one is responsible for these pathways. In order to avoid this problem, we combined species annotation with functional gene annotation to study the nitrifying and denitrifying bacteria with N_2O -related genes and then distinguish the *nir* and *nor* genes in ND, NCD, and HD, thereby revealing the metabolic mechanism of N_2O production.

We then discussed that the biological energy and detoxification function required for metabolic processes and basic survival appear to be the two important drivers of N_2O production in addition to substrates and catalytic enzymes. Regarding the perspective of detoxification, we only added it because the questions raised by the reviewers reminded us. But if we only use geochemical research methods instead of molecular biology methods, we may not be able to explain this problem. From another perspective, the explanation of this mechanism and related discussions may also be necessary for all the relevant findings (NH_4^+ -derived N_2O production, rather than NO_3^- -derived N_2O production, are the dominant). In my view, this is more vital to better understand the microbial N_2O production mechanism. **(Lines 75-76, 203-234; Supplementary Table S4)**

References:

- Chen, C., Pan, J. Y., Xiao, S. X., Wang, J. Y., Gong, X. L., Yin, G. Y., Hou, L. J., Liu, M. & Zheng, Y. L. Microplastics alter nitrous oxide production and pathways through affecting microbiome in estuarine sediments. *Water Res.* **221**, 118733 (2022).
- Fang, W. S., Wang, Q. X., Li, Y., Hua, J. L., Jin, X., Yan, D. D. & Cao, A. C. Microbial regulation of nitrous oxide emissions from chloropicrin-fumigated soil amended with biochar. *J. Hazard. Mater.* **429**, 128060 (2022).
- Fang, W. S., D. D. Yan, B. Huang, Z. J. Ren, X. L. Wang, X. M. Liu, Y. Li, et al. Biochemical pathways used by microorganisms to produce nitrous oxide emissions from soils fumigated with dimethyl disulfide or allyl isothiocyanate. *Soil Biol. Biochem.* **132**, 1-13 (2019).
- Wrage, N., van Groenigen, J. W., Oenema, O., & Baggs, E. M. A novel dual-isotope labelling method for distinguishing between soil sources of N_2O . *Rapid Commun. Mass Spectrom.* **19(22)**, 3298-3306 (2005).

- Zhang, Q. Q., Wu, Z., Zhang, X., Duan, P. P., Shen, H. J., Gunina, A., Yan, X. Y., & Xiong, Z. Q. Biochar amendment mitigated N₂O emissions from paddy field during the wheat growing season. *Environ. Pollut.* **281**, 117026 (2021).
- Zhu, X., Burger, M., Doane, T. A., & Horwath, W. R. Ammonia oxidation pathways and nitrifier denitrification are significant sources of N₂O and no under low oxygen availability. *Proc. Natl. Acad. Sci. U.S.A.* **110**(16), 6328-6333 (2013).
- Yao, Y. Z., Tian, H. Q., Shi, H., Pan, S. F., Xu, R. T., Pan, N. Q., & Canadell, J. G. Increased global nitrous oxide emissions from streams and rivers in the Anthropocene. *Nat. Clim. Change.* **10**(2), 138-139 (2020).
- Qin, Y., Wang, S. Y., Wang, X. M., Liu, C. L., & Zhu, G. B. Contribution of ammonium-induced nitrifier denitrification to N₂O in paddy fields. *Environ. Sci. Technol.* **57**(7), 2970–2980 (2023).
- Yuan, D. D., Zheng, L., Liu, Y. X., Cheng, H. G., Ding, A. Z., Wang, X. M., Tan, Q. Y., Wang, X., Xing, Y. Z., Xie, E., Wu, H. M., Wang, S. Y., & Zhu, G. B. Nitrifiers cooperate to produce nitrous oxide in Plateau wetland sediments. *Environ. Sci. Technol.* **57**(1): 810-821 (2023).
- Hefting, M. M., Bobbink, R., & Janssens, M. P. Spatial variation in denitrification and N₂O emission in relation to nitrate removal efficiency in a n-stressed riparian buffer zone. *Ecosystems.* **9**(4), 550-563 (2006).
- Wang, H. J., Wang, W. D., Yin, C. Q., Wang, Y. C., & Lu, J. W. Littoral zones as the "hotspots" of nitrous oxide (N₂O) emission in a hyper-eutrophic lake in China. *Atmos. Environ.* **40**(28), 5522-5527 (2006).
- Wang, H. J., Yang, L. Y., Wang, W. D., Lu, J. W., & Yin, C. Q. Nitrous oxide (N₂O) fluxes and their relationships with water-sediment characteristics in a hyper-eutrophic shallow lake, China. *J. Geophys. Res.: Biogeosci.* **112**(G1), (2007).

- Does the work support the conclusions and claims, or is additional evidence needed?

The conclusion that most N₂O is from ammonia oxidation is misleading as some likely is from denitrifiers, coupled to ammonia oxidizers. Additionally, some seems to be from ammonia oxidizers running denitrifier proteins under similar conditions that denitrifiers thrive in.

Response:

We fully understand and agree with your statement. Your query represents the potential misunderstandings that might be coming from readers in the wider field. In order to make the manuscript clearer and more readable, we used the statements as “NH₄⁺-derived N₂O production” and “NO₃⁻-derived N₂O production” instead of “ammonia oxidation” and “heterotrophic denitrification”, respectively, in the revised manuscript.

With ambient NH₄⁺ as substrate, the NH₄⁺-derived N₂O production process includes nitrifier nitrification (NN) pathway via nitrifier, nitrifier denitrification (ND) pathway via a denitrification function of nitrifiers, and nitrification-coupled denitrification (NCD) pathway via nitrifier coupled lumped denitrifier. With ambient NO₃⁻ as substrate, the NO₃⁻-derived process represents the heterotrophic denitrification (HD) pathway via lumped denitrifier (See the Unpublished Figure below).

N₂O production is a series of complex processes with three grouping methods. i) In terms of strains, NN and ND pathways are performed by nitrifiers, while NCD and HD are performed by denitrifiers. ii) In terms of microbial processes, only N₂O via NN is performed by nitrification, while the last three are performed finally by denitrification. iii) In terms of N source in N₂O, NN, ND and NCD are derived from ambient NH₄⁺ oxidation (namely NH₄⁺-derived N₂O production), while HD is derived from ambient NO₃⁻ reduction (namely NO₃⁻-derived N₂O production). As our findings would be useful for environmental N management, we distinguished the N₂O production pathways according to N source in N₂O. This statement has been used in many studies (Zhu et al., 2013; Kool et al., 2011).

References:

- Kool, D. M., Dolfing, J., Wrage, N., & Van Groenigen, J. W. Nitrifier denitrification as a distinct and significant source of nitrous oxide from soil. *Soil Biol. Biochem.* **43**(1), 174-178 (2011).
- Zhu, X., Burger, M., Doane, T. A., & Horwath, W. R. Ammonia oxidation pathways and nitrifier denitrification are significant sources of N₂O and no under low oxygen availability. *Proc. Natl. Acad. Sci. U.S.A.* **110**(16), 6328-6333 (2013).

The major microbial N_2O production pathways and their associated enzymes (Unpublished)

Microorganisms carry enzymes that perform 8 redox reactions involving 7 key inorganic nitrogen species (see color coded circles for: NO_3^- , NO_2^- , NO , N_2O , NH_2OH , NH_4^+ , N_2). The reactions involve Nitrifier Nitrification (NN, red), Nitrifier Denitrification (ND, purple), Nitrification-Coupled Denitrification (NCD, blue) and Heterotrophic Denitrification (HD, green). Genes encoding enzymes that conduct the important transformations include those for ammonium monooxygenase (Amo), hydroxylamine oxidoreductase (Hao); nitrite oxidoreductase (Nxr); membrane-bound (Nar) and periplasmic (Nap) dissimilatory nitrate reductases; nitrite reductase (Nir); nitric oxide reductase (Nor); nitrous oxide reductase (NosZ).

The correlations between NH_4^+ and N_2O production should also include dissolved O_2 . Several studies have shown strong negative correlations between O_2 and N_2O concs in rivers.

Response:

Unfortunately, we did not fully measure dissolved oxygen in the water column, because all of our samples were sediments, and the primary goal was to study the rate and pathway of N_2O production in stream sediments.

Firstly, based on our limited data on dissolved oxygen of the water column ($7.73 \pm 0.39 \text{ mg L}^{-1}$), we were unable to obtain a reliable relationship between oxygen and N_2O concentration. However, we can confirm that the overlying water in our study area is in an aerobic state. We completely agree with your previous point that the aerobic water column itself cannot produce much N_2O . Therefore, although oxygen in the water column is negatively correlated with N_2O concentration (Rosamond et al., 2012), we personally believe that the N_2O production should also be even more influenced by the oxygen content in sediment.

Secondly, Zhu Xia (Zhu et al., 2013) also found that as long as oxygen was present, even at a very low level (0.5% O_2), the NH_4^+ -derived N_2O production process contributed a large amount of N_2O , even exceeding the process of NO_3^- -derived N_2O production. The NO_3^- -derived process was responsible for all N_2O production at 0% O_2 . In this study, the oxygen content in the sediment we measured was at an available level ($0.34 \pm 0.07 \text{ mg L}^{-1}$), meaning that oxygen may not be the limiting factor. Pearson correlation analysis also showed that the most important physicochemical factor affecting N_2O production was ammonium rather than oxygen content in sediments.

Thirdly, N_2O production is also correlated to many other factors, such as water depth and light. For example, Rosamond et al. (2012) (<https://www.nature.com/articles/ngeo1556>) stated that the sampling site (Grand River Ontario, Canada) is shallow (mean depth $< 1 \text{ m}$ at most sites). In addition, dissolved O_2 gradually decreases with depth along the water column profile until the water/sediment interface changes dramatically. Therefore, it is difficult to determine which layer is more suitable for the correlation calculation.

Therefore, we did not get the relationship between water column oxygen and N_2O .

References:

Rosamond, M., Thuss, S. & Schiff, S. Dependence of riverine nitrous oxide emissions on dissolved oxygen levels. *Nature Geosci.* **5**, 715-718 (2012).

Zhu, X., Burger, M., Doane, T. A., & Horwath, W. R. Ammonia oxidation pathways and nitrifier denitrification are significant sources of N₂O and no under low oxygen availability. *Proc. Natl. Acad. Sci. U.S.A.* **110**(16), 6328-6333 (2013).

- Are there any flaws in the data analysis, interpretation and conclusions? - Do these prohibit publication or require revision?

I didn't really understand if nos was included in two datasets for the statistical tests and if this is considered OK.

Response:

Thanks for your kind remainder.

Gene *nos* was the only N₂O reduction gene, which belong to lumped denitrifiers. The denitrification step of NCD pathways share the same genes, belonging to the same denitrifiers, with HD pathway. In this study, we used the transcriptome abundance of *nos* as the maximum potential for N₂O reduction and compared it to the maximum potential for N₂O production, including both the NH₄⁺-derived and NO₃⁻-derived pathways. However, we did not use *nor* for separate comparisons and statistical tests with NH₄⁺-derived or NO₃⁻-derived N₂O production.

Figure 1

Figure 3

- Is the methodology sound? Does the work meet the expected standards in your field? Methods for isotopic analysis, concs etc look fine. Dissolved O₂ should be measured.

Response:

Thank you for approving the methods of isotopic analysis, concs, etc. in our manuscript.

In fact, we did not fully measure dissolved oxygen in the water column, because all of our samples were sediments, and the primary goal was to study the rate and pathway of N₂O production in stream sediments. We measured the *in-situ* O₂ content in sediments (0.34±0.07 mg L⁻¹) by using a Pocket Oxygen Meter (FireStingGO2, PyroScience GmbH, Germany). See the related method in the section of Analytical Procedures of Environmental Variables (Lines 389-391).

- Is there enough detail provided in the methods for the work to be reproduced?

C₂H₂ method (to inhibit production of N₂ from N₂O) is not explained in methods. I didn't follow what the authors mean by "in-situ O₂" concentrations in the sed incubations.

Response:

First of all, we would like to apologize for this oversight.

In this study, we used C₂H₂ (0.1% v/v) as a selective inhibitor of the ammonia monooxygenase (AMO) enzyme (Taylor et al., 2013; Wang et al., 2021) to distinguish the N₂O source of NH₄⁺-derived and NO₃⁻-derived processes. In the meanwhile, we used ZnCl₂ (600 μl, 7M) as a biological inactivator, taking advantage of its

ability to inactivate proteins (Zhu et al., 2013; Wang et al., 2020). Based on the experimental and groups shown below, the related N_2O production rate could be accurately distinguished and calculated:

a) The production rate of NH_4^+ -derived N_2O was calculated by the different value of the N_2O production rate between treatment i (Control) and ii (C_2H_2 ; 0.1% v/v). Here, the production rate of NH_4^+ -derived N_2O was the total rate of NN, ND, and NCD pathways.

b) The production rate of biotic N_2O was calculated by the different value of the N_2O production rate between treatment i (Control) and iii ($ZnCl_2$; 600 μ l, 7M).

c) The production rate of NO_3^- -derived N_2O was calculated by the different value of biotic and NH_4^+ -derived processes.

In the revised manuscript, we have supplemented the section of C_2H_2 -inhibitor method (Lines 333-338).

Secondly, the *in-situ* O_2 concentration referred to the *in-situ* O_2 content in sediments (0.34 ± 0.07 mg L^{-1}) with a Pocket Oxygen Meter (FireStingGO2, PyroScience GmbH, Germany). Prior to the incubations of ^{15}N - ^{18}O dual-isotope tracing, ^{15}N tracing of the *semi-in-situ* sediment core, and C_2H_2 inhibitor, the gas in the headspace of the plexiglass tube was evacuated and then balanced using high-purity Ar (99.99%; Beijing Huayuan Gas, Beijing, China) to the standard atmospheric pressure. The oxygen content was immediately adjusted by the injection of a known volume of high-purity O_2 (99.99%; Beijing Huayuan Gas, Beijing, China) to the *in situ* oxygen content after evacuating the same volume of Ar from the headspace to balance the pressure. (Lines 299-304, 324-329).

References:

- Taylor, A. E., Vajjala, N., Giguere, A. T., Gitelman, A. I., Arp, D. J., Myrold, D. D., Sayavedra-Soto, L., & Bottomley, P. J. Use of aliphatic n-alkynes to discriminate soil nitrification activities of ammonia-oxidizing *Thaumarchaea* and bacteria. *Appl. Environ. Microbiol.* **79**, 6544-6551 (2013).
- Wang, S. Y., Wang, X. M., Jiang, Y. Y., Han, C., Jetten, M. S. M., Schwark, L., Li, F. B., & Zhu, G. B. Abundance and functional importance of complete ammonia oxidizers and other nitrifiers in a riparian ecosystem. *Environ. Sci. Technol.* **55**(8): 4573-4584 (2021).
- Zhu, G. B., Wang, S. Y., Wang, W. D., Wang, Y., Zhou, L. L., Jiang, B., Op den Camp, H. J. M., Hefting, M. M., Risgaard-Petersen, N., Peng, Y. Z., Schwark, L., Jetten, M. S. M., & Yin, C. Q. Hotspots of anaerobic ammonia oxidation at land/freshwater interfaces. *Nature Geo.* **6**, 103-107 (2013).
- Wang, S. Y., Zhu, G. B., Zhuang, L. J., Li, Y. X., Liu, L., Lavik, G., Berg, M., Liu, S. T., Long, X.-E., Guo, J. H., Jetten, M. S. M., Kuypers, M. M. M., Li, F. B., Schwark, L., & Yin, C. Q. Anaerobic ammonium oxidation is a major N-sink in aquifer systems around the world. *ISME J.* **14**(1): 151-163 (2020).

Line-by-line comments:

Line 44: Do all 3 refs say up to 98%? Can you put in the range reported instead? I would think this could be very low, depending on substrate permeability. Also does "overlying surface waters" include the benthic zone sediment surface layer and/or biofilm on rocks etc? I would expect a lot of N₂O from anoxic/hypoxic areas which could include all of these, and not very much production in the water column itself (unless the water goes hypoxic).

Response:

Nice comments. In fact, the 3 references were originally cited to illustrate the important role of hyporheic zone in rivers and streams. The "overlying surface waters" here didn't include the benthic zone sediment surface layer and/or biofilm on rocks etc, but only referred to the water column itself. To avoid misunderstandings, we have modified this unclear text and the inappropriate citations in the revised manuscript. **(Lines 44-47)**

Line 45: I think ref 8 likely does not have N₂O:N₂ models (it's about plant nutrition) and ref 12 might be included in this list. I'm actually not sure what you mean to cite for Ref 8 - can you clarify? (I checked the index and there's nothing about denitrification, N₂O, etc.) Please check refs over carefully.

Response:

We have replaced Ref 8 with Reference — Yao et al. 2020, Nature Climate Change.

Reference:

Yao, Y. Z., Tian, H. Q., Shi, H., Pan, S. F., Xu, R. T., Pan, N. Q., & Canadell, J. G. Increased global nitrous oxide emissions from streams and rivers in the Anthropocene. *Nat. Clim. Change*. 10(2), 138-139 (2020).

Line 46 - this sentence needs a reference (even if it's the same ref as next sentence). Clarify if this is true for all/most models? Just one specific paper?

Another mechanistic model that I don't see cited but will probably help you is here: Maavara et al. 2018 (<https://doi.org/10.1111/gcb.14504>). It does include nitrification.

Response:

Thank you for the good comment and nice suggestion. Rewrite this statement as follows:

"Model parameters associated with microbially mediated hyporheic N₂O production are poorly represented (Yao et al. 2020; Maavara et al. 2018)." **(Lines 49-50)**

References:

Yao, Y. Z., Tian, H. Q., Shi, H., Pan, S. F., Xu, R. T., Pan, N. Q., & Canadell, J. G. Increased global nitrous oxide emissions from streams and rivers in the Anthropocene. *Nat. Clim. Change*. 10(2), 138-139 (2020).

Maavara, T., Lauerwald, R., Laruelle, G. G., Akbarzadeh, Z., Bouskill, N. J., Van Cappellen, P., & Regnier, P. Nitrous oxide emissions from inland waters: Are IPCC estimates too high? *Global Change Biology* **25**, 473–488 (2019).

47: "water surface area" - I assume this means the upper surface area, not the benthic surface area, but unclear how this relates to the hyporheic zone and why it's so hard to measure water surface area globally? (The second part may not matter, I'm not just sure how this relates to the previous sentence.)

Response:

We deleted this sentence regarding water surface area.

48: suggest removing ref 14 because it's about soils. I don't think confounding soil regimes and streams/rivers will help make your case.

Response:

Deleted.

48: "hardly considered" seems a bit harsh! The IPCC method assumes all NH_4^+ is nitrified, producing N_2O , and half is then denitrified, producing more N_2O . Some papers such as Beaulieu et al. (2011) were not designed to examine nitrification, so suggest wording something like "though the IPCC methods include N_2O from nitrification (ref), some larger studies have focused on quantifying denitrification and N_2O emissions resulting from adding NO_3^- (e.g. Beaulieu et al., 2011)." Beaulieu et al. specifically state that there are few $\text{N}_2\text{O}:\text{NO}_3^-$ yields for nitrification but use the IPCC method to say nitrification could be ~56% of N_2O production in streams.

Response:

Agree. We take your suggested wording into the revised version (**Lines 50-52**). Thanks for your good comment.

50: You could mention that it is difficult to distinguish denitrifier and nitrifier-denitrification N_2O without analysing the microbes' genetic material as they both increase in low- O_2 settings. This is probably a major reason some studies don't tease them out (also, NH_4^+ is often quite low!)

Response:

Agree. We supplement this sentence in the revised manuscript so as to make it more easily understand (**Lines 52-53**). Many thanks.

51: clarify who is not taking this into account as refs 16-18 are all about how hot spots are important. The problem (as I understand it!) is that heterogeneity is very difficult to model. Could reword as something like, "Hyporheic N_2O production has been shown to be very heterogenous (Kiel et al., 2014; Biddulph 2015) but this is very difficult to account for in models and thus is often left out (Maavara et al., 2018)".

Response:

Revised as suggested (**Lines 53-55**) and thanks.

References:

- Biddulph, M. Hyporheic Zone: In Situ Sampling. *Geomorphological Techniques. British Society for Geomorphology* (2015).
- Kiel, B. A. & Cardenas, M. B. Lateral hyporheic exchange throughout the Mississippi River network. *Nat. Geosci.* **7**(6), 413-417 (2014).
- Maavara, T., Lauerwald, R., Laruelle, G. G., Akbarzadeh, Z., Bouskill, N. J., Van Cappellen, P., & Regnier, P. Nitrous oxide emissions from inland waters: Are IPCC estimates too high? *Global Change Biol.* **25**, 473–488 (2019).

61: add refs to papers that do look at N_2O from Chinese streams

Response:

Added.

References:

- Wang, J., Chen, N., Yan, W., Wang, B., & Yang, L. Effect of dissolved oxygen and nitrogen on emission of N_2O from rivers in China. *Atmos. Environ.* **103**(103), 347-356 (2015).
- Kumar, A., Yang, T., & Sharma, M. P. Greenhouse gas measurement from Chinese freshwater bodies: a review. *J. Clean. Prod.* **233**, 368-378 (2019).
- Tang, M.-Y., Hu, X.-K., Wang, H.-W., Wang, Y.-C., Chang, S.-Y., Wang, S.-Q., Zhong, J.-C. Diffusive Fluxes and Controls of N_2O from Coastal Rivers in Tianjin City. *Environ. Sci.* **43**(03): 1481-1491 (2022). (in Chinese)
- Gong, J.-W. Temporal and spatial distribution of CO_2 , CH_4 and N_2O concentration and emission rate in the Chaobai River Basin. *Beijing Jiaotong University* (2022). (in Chinese)

62: awkwardly worded, suggest something like "we compared these results to those measured globally" (or maybe list the areas covered). Suggest specifying temperate and tropical as I'm not seeing boreal polar areas represented in Fig 4.

Response:

Revised and thanks. It would be better to label temperate and tropical regions because we did not sample in the boreal polar areas. (Lines 66-67)

72: Did you measure dissolved oxygen in the water column and if so, did it show a negative trend with N₂O concentration, as in Rosamond et al. (2012) (<https://www.nature.com/articles/ngeo1556>)? This is also seen in a few other sites cited in their paper.

Response:

As mentioned earlier (**Response letter Page 10**), we did not fully measure dissolved oxygen in the water column, because all of our samples were sediments, and the primary goal was to study the rate and pathway of N₂O production in stream sediments.

Firstly, based on our limited data on dissolved oxygen of the water column (7.73 ± 0.39 mg L⁻¹), we were unable to obtain a reliable relationship between oxygen and N₂O concentration. However, we can confirm that the overlying water in our study area is in an aerobic state. We completely agree with your previous point that the aerobic water column itself cannot produce much N₂O. Therefore, although oxygen in the water column is negatively correlated with N₂O concentration (Rosamond et al., 2012), we personally believe that the N₂O production should also be even more influenced by the oxygen content in sediment.

Secondly, Zhu Xia (Zhu et al., 2013) also found that as long as oxygen was present, even at a very low level (0.5% O₂), the NH₄⁺-derived N₂O production process contributed a large amount of N₂O, even exceeding the process of NO₃⁻-derived N₂O production. The NO₃⁻-derived process was responsible for all N₂O production at 0% O₂. In this study, the oxygen content in the sediment we measured was at an available level (0.34 ± 0.07 mg L⁻¹), meaning that oxygen may not be the limiting factor. Pearson correlation analysis also showed that the most important physicochemical factor affecting N₂O production was ammonium rather than oxygen content in sediments.

Thirdly, N₂O production is also correlated to many other factors, such as water depth and light. For example, Rosamond et al. (2012) (<https://www.nature.com/articles/ngeo1556>) stated that the sampling site (Grand River Ontario, Canada) is shallow (mean depth <1 m at most sites). In addition, dissolved O₂ gradually decreases with depth along the water column profile until the water/sediment interface changes dramatically. Therefore, it is difficult to determine which layer is more suitable for the correlation calculation.

Therefore, we did not obtain the relationship between water column oxygen and N₂O reported by Rosamond et al. (2012).

Based on the above reasons, we did not get the relationship between water column oxygen and N₂O.

References:

Rosamond, M., Thuss, S. & Schiff, S. Dependence of riverine nitrous oxide emissions on dissolved oxygen levels. *Nature Geosci.* **5**, 715-718 (2012).

Zhu, X., Burger, M., Doane, T. A., & Horwath, W. R. Ammonia oxidation pathways and nitrifier denitrification are significant sources of N₂O and no under low oxygen availability. *Proc. Natl. Acad. Sci. U.S.A.* **110**(16), 6328-6333 (2013).

82: remove "further"

Response:

Revised.

86: You get into this later, but I think you need to clarify here what the labelling can and can't do, ie can't distinguish NCD and denitrification.

The detailed information has been stated above.

Response:

The ^{15}N tracing incubation of *semi-in-situ* sediment core was designed by the first author Dr. Wang. The $^{15}\text{N-NH}_4^+$ and $^{15}\text{N-NO}_3^-$ were added to trace NH_4^+ -derived and NO_3^- -derived N_2O production, respectively. The sediment core was incubated under the intact sediment structure and the *in-situ* oxygen and temperature conditions, closer to actual production conditions than slurry incubation. Although the ^{15}N tracing incubation of *semi-in-situ* sediment core cannot distinguish the NN, ND and NCD pathways of NH_4^+ -derived N_2O without the addition of ^{18}O tracers, it can accurately determine the rates and contributions of NH_4^+ -derived and NO_3^- -derived N_2O .

In the original version, we expressed what it could and couldn't do in the methods. Now we have added information about what it cannot do in the result part. **(Lines 87-88)**

Could you also interpret these results as "nitrification occurs very rapidly, and then that NO_3^- (or NO_2^-) is used in denitrification"? If more NH_4^+ is added to the system, does that increase the rate of nitrification or is it limited by something else?

Response:

We thought that your question was still due to the unclear terms of N_2O production processes in our original manuscript. Based on your kind suggestion, we have clarified the "Ammonia oxidation" and "Heterotrophic denitrification" processes to " NH_4^+ -derived N_2O production" and " NO_3^- -derived N_2O production" processes, respectively, which were more precise.

Regarding your suggested explanation, we agree that the NO_3^- (or NO_2^-) produced by nitrifiers could be used by denitrifiers, which is exactly the NCD pathway of NH_4^+ -derived N_2O production process mentioned in our study. In the ^{15}N -tracing incubation of *semi-in-situ* sediment core, we added $^{15}\text{N-NH}_4^+$ and $^{15}\text{N-NO}_3^-$, respectively, to trace the N_2O produced during NH_4^+ -derived and NO_3^- -derived processes. Therefore, although this method cannot distinguish the three pathways of NH_4^+ -derived N_2O , it can accurately determine the rates of NH_4^+ -derived and NO_3^- -derived N_2O .

In addition, our study showed that sediment NH_4^+ , rather than NO_3^- (or NO_2^-), was the most important physicochemical factor, positively influencing N_2O production. Therefore, adding more NH_4^+ may increase the production rate of NH_4^+ -derived N_2O , but it is uncertain whether it will increase the nitrification rate. In this study, in order not to significantly affect the total NH_4^+ content, we controlled the addition of $^{15}\text{N-NH}_4^+$ (or NO_3^-) at 8% of the *in situ* concentration [NH_4^+ ($9.63 \pm 4.75 \text{ mg kg}^{-1}$ in the open water; $1.48 \pm 0.49 \text{ mg kg}^{-1}$ in the riparian zone) and NO_3^- ($0.66 \pm 0.06 \text{ mg kg}^{-1}$ in the open water; $0.76 \pm 0.12 \text{ mg kg}^{-1}$ in the riparian zone)]. The incubation was then conducted under conditions of *in situ* oxygen content, temperature and sediment state as close as possible to the *in situ* production situation.

101: these papers do not appear to be about the same sites as yours. Please clarify "in different river systems" as I don't expect all riparian zones to be the same in terms of permeability, organic carbon availability, reactive N, microbial community etc.

Response:

Revised. **(Line 106)**

101: wording it this way is potentially misleading as it's not necessarily the NH_4^+ oxidation itself that produces the N_2O unless it's coming from the NH_4^+ \rightarrow NO_3^- suite of enzymes. (I see new research indicating that N_2O

can come from hydroxylamine oxidation in AOA, here: <https://www.pnas.org/doi/10.1073/pnas.2220697120>. Previously, my understanding is that all N₂O from nitrifiers comes from them running nitrogen reductases (as a way to remove toxic NO₂⁻). Also it's hard to know what type of microbe is producing the N₂O with the data you've shown so far. Suggest something like, "N₂O was produced from NO₂⁻ or NO₃⁻ that originally came from labelled NH₄⁺"

Response:

As mentioned earlier (**Response letter Pages 11-12**), in this study, we used C₂H₂ (0.1% v/v) as the selective inhibitor of ammonia monooxygenase (AMO) enzyme (Taylor et al., 2013; Wang et al., 2021) to distinguish the N₂O source of NH₄⁺-derived and NO₃⁻-derived processes. In the meanwhile, we used ZnCl₂ (600 μl, 7M) as a biological inactivator, taking advantage of its ability to inactivate proteins (Zhu et al., 2013; Wang et al., 2020). Based on the experimental and groups showed below, the related N₂O production rate could be accurately distinguished and calculated:

a) The production rate of NH₄⁺-derived N₂O was calculated by the different value of the N₂O production rate between treatment i (Control) and ii (C₂H₂; 0.1% v/v). Here, the production rate of NH₄⁺-derived N₂O was the total rate of NN, ND, and NCD pathways.

b) The production rate of biotic N₂O was calculated by the different value of the N₂O production rate between treatment i (Control) and iii (ZnCl₂; 600 μl, 7M).

c) The production rate of NO₃⁻-derived N₂O was calculated by the different value of biotic and NH₄⁺-derived processes.

Now, the related sentence has been modified to "In riparian and open water zone, NH₄⁺-derived N₂O accounted for 89.2 ± 2.9% and 76.1 ± 2.6% of the total N₂O produced, respectively." (**Lines 106-108**) Meanwhile, the C₂H₂-inhibitor method has been detailed in the revised manuscript. (**Lines 333-338**)

References:

- Taylor, A. E., Vajjala, N., Giguere, A. T., Gitelman, A. I., Arp, D. J., Myrold, D. D., Sayavedra-Soto, L., & Bottomley, P. J. Use of aliphatic n-alkynes to discriminate soil nitrification activities of ammonia-oxidizing *Thaumarchaea* and bacteria. *Appl. Environ. Microbiol.* **79**, 6544-6551 (2013).
- Wang, S. Y., Wang, X. M., Jiang, Y. Y., Han, C., Jetten, M. S. M., Schwark, L., Li, F. B., & Zhu, G. B. Abundance and functional importance of complete ammonia oxidizers and other nitrifiers in a riparian ecosystem. *Environ. Sci. Technol.* **55**(8): 4573-4584 (2021).
- Zhu, G. B., Wang, S. Y., Wang, W. D., Wang, Y., Zhou, L. L., Jiang, B., Op den Camp, H. J. M., Hefting, M. M., Risgaard-Petersen, N., Peng, Y. Z., Schwark, L., Jetten, M. S. M., & Yin, C. Q. Hotspots of anaerobic ammonia oxidation at land/freshwater interfaces. *Nature Geo.* **6**, 103-107 (2013).
- Wang, S. Y., Zhu, G. B., Zhuang, L. J., Li, Y. X., Liu, L., Lavik, G., Berg, M., Liu, S. T., Long, X.-E., Guo, J. H., Jetten, M. S. M., Kuypers, M. M. M., Li, F. B., Schwark L., & Yin, C. Q. Anaerobic ammonium oxidation is a major N-sink in aquifer systems around the world. *ISME J.* **14**(1): 151-163 (2020).

104: I've always been of the opinion that (the way I do research), I can't differentiate NCD, ND or regular denitrification so they are lumped as "denitrification" in the environment, to me. I think you may see that reflected in the literature. I don't think too many people deny that NCD or ND are unimportant but they are hard to tease out.

Response:

As you mentioned above, the three denitrification pathways occur sharing same/similar genes under same conditions, so it seems difficult to be distinguished. Therefore, in this study, we used the complementary methods such as C₂H₂-inhibitor method, ¹⁵N tracing *semi-in-situ* sediment-core incubation, ¹⁵N-¹⁸O dual-tracing method, quantitative reverse transcription PCR (RT-qPCR), and metagenomic sequencing and assembly analysis. The first two methods can distinguish NH₄⁺-derived and NO₃⁻-derived processes. With ¹⁸O tracers added, the ¹⁵N-¹⁸O dual-tracing method can distinguish the NH₄⁺-derived N₂O production process into NN, ND and NCD pathways, and calculate the maximum and minimum contribution of NN, ND and NCD pathways. The ¹⁵N-¹⁸O dual-tracing method has been widely used and approved. In addition, in metagenomic sequencing and assembly analysis, we combined species annotation with functional gene annotation to distinguish the species and relative abundance of nitrifying and denitrifying bacteria with N₂O-related denitrifying genes, thereby revealing the metabolic mechanism of N₂O production. It has been detailed above.

In this study, we distinguished the N₂O production pathways according to the original N source deriving N₂O in global streams. In the revised manuscript, we clarified the "heterotrophic denitrification" to "NO₃⁻-derived process". (Line 109)

107: what are the nitrate and ammonia concentrations like in the riparian zone? If they are low (or dissolved O₂ is high), may not be much opportunity for denitrification (of any kind).

Response:

Thank you for your reminder. In the riparian zone, the sediment NH₄⁺ and NO₃⁻ was 1.48±0.49 mg kg⁻¹ and 0.76±0.12 mg kg⁻¹, respectively. The sediment NH₄⁺ in riparian zone was lower than that in open water (9.63±4.75 mg kg⁻¹), while NO₃⁻ was not significantly different between both zones (0.66±0.06 mg kg⁻¹ in open water).

109: the second part of this sentence was unclear to me. I think you are using "open water sediment" and "hyporheic zone" interchangeably but it keeps making me think you have 3 types of sites instead of 2.

Response:

The hyporheic zone is a highly dynamic region, roughly defined as the saturated interstitial sediment below the riverbed and adjacent riverbank where the exchange of channel water and groundwater occurs (Biddulph 2015). In our study, both site A and site B located in the hyporheic zone. Site A represents the nearshore hyporheic zone - riparian zone sediment, and site B represents the offshore hyporheic zone - open water sediment. There are significant differences in hydraulic and biogeochemical characteristics between the two zones. Based on our previous studies in the riparian zone (Zhu et al., *Nature Geo.* 2013; Wang et al., *Environ. Sci. Technol.* 2012, 2018, 2021; Wang et al., *Water Res.* 2019, 2020), this study sampled these two sites to better reveal the mechanism of N₂O production in the hyporheic zone.

In the revised manuscript, we have modified this unclear sentence. (Lines 119-122)

The approximate position of the hyporheic zone from a cross-sectional view of a river catchment (Biddulph 2015)

Diagram of Hyporheic Zone in this study

References:

Biddulph, M. Hyporheic Zone: In Situ Sampling. *Geomorphological Techniques. British Society for Geomorphology* (2015).

Zhu, G.B., Wang, S.Y., Wang, W.D., Wang, Y., Zhou, L.L., Jiang, B., Op den Camp, H.J.M., Hefting, M.M., Risgaard-Petersen, N., Peng, Y.Z., Schwark, L., Jetten, M.S.M., & Yin, C.Q. Hotspots of anaerobic ammonia oxidation at land/freshwater interfaces. *Nature Geo.* **6**, 103-107 (2013).

Wang, S.Y., Wang, X.M., Jiang, Y.Y., Han, C., Jetten, M.S.M., Schwark, L., Li, F.B., & Zhu, G.B. Abundance and functional importance of complete ammonia oxidizers and other nitrifiers in a riparian ecosystem. *Environ. Sci. Technol.* **55**(8): 4573-4584 (2021).

Wang, S.Y., Wang, W.D., Liu, L., Zhuang, L.J., Zhao, S.Y., Su, Y., Li, Y.X., Wang, M.Z., Wang, C., Xu, L.Y., & Zhu, G.B. Microbial nitrogen cycle hotspots in the plant-bed/ditch system of a constructed wetland with N₂O mitigation. *Environ. Sci. Technol.* **52**(11): 6226-6236 (2018).

Wang, S.Y., Zhu, G.B., Jetten, M. S.M., & Yin, C.Q. Anammox bacterial abundance, activity and contribution in riparian sediments of the Pearl River Estuary. *Environ. Sci. Technol.* **46**: 8834-8842 (2012).

Wang, S.Y., Pi, Y.X., Song, Y.P., Jiang, Y.Y, Zhou, L.G., Liu, W.Y., & Zhu, G.B. Hotspot of dissimilatory nitrate reduction to ammonium (DNRA) process in freshwater sediments of riparian zones. *Water Res.* **173**: 115539 (2020).

Wang, S.Y., Wang, W.D., Zhao, S.Y., Wang, X.M., Hefting, M.M., Schwark, L., & Zhu, G.B. Anammox and denitrification separately dominate microbial N-loss in water saturated and unsaturated soils horizons of riparian zones. *Water Res.* **162**(1): 139-150 (2019).

"higher potential of N₂O production" - does this mean that more copies of NOR genes were transcribed? Please clarify.

Response:

Yes. Reverse transcription PCR (RT-qPCR) results showed that the whole transcript abundance of N₂O production related genes were higher than that of N₂O reduction gene (*nosZ*) in both riparian zone and the open water. Hence, we said that the transcript abundance also showed that the hyporheic zone (both site A - riparian zone and site B - open water) had a higher potential of N₂O production. (Fig. 1-d)

To make it clearer, we clarified the related words in **the legend of Figure 1 (Lines 98-101)**.

Figure 1 Spatial-temporal N₂O emission fluxes and microbial production sources in the riparian zone and open water sediments along a transect of the riverine hyporheic zone at site scale. d) Transcript abundance of N₂O-production related (*amoA*, *norB*, *nirS*, and *nirK*) and N₂O-reduction related (*nosZ*) genes. Here, *amoA*, ammonia monooxygenase gene (NH₄⁺→NH₂OH) in NN pathway; *nirSK*, nitrite reductase genes (NO₂⁻→NO) in ND (*nirK* only), NCD and HD pathways; *norB*, nitric oxide reductase gene (NO→N₂O) in ND, NCD and HD pathways; *nosZ*, nitrous oxide reductase gene (N₂O→N₂) in NCD and HD pathways.

113: add references. I would assume this is true in pristine systems but perhaps not ones with a lot of NH₄⁺ pollution. Certainly NH₄⁺ is often much lower than NO₃⁻ in many oxic systems, suggest it's getting cycled quickly.

Response:

Due to your kind reminder, we modified these words to a more accurate statement - “ammonia oxidation is the rate-limiting step of nitrification (Kowalchuk et al., 2001; Wang et al., 2015 2019), and even the N-cycle (Wang et al., 2011)”. The references had been added in the newly manuscript. (Lines 123-124) Many thanks.

References:

Kowalchuk, G. A., & Stephen, J. R. Ammonia-oxidizing bacteria: a model for molecular microbial ecology. *Annu. Rev. Microbiol.* **55**, 485-529 (2001).

Wang, B., Zhao, J., Guo, Z., Ma, J., Xu, H., & Jia, Z. Differential contributions of ammonia oxidizers and nitrite oxidizers to nitrification in four paddy soils. *ISME J.* **9**(5), 1062 (2015).

Wang, B., Qin, W., Ren, Y. et al. Expansion of *Thaumarchaeota* habitat range is correlated with horizontal transfer of ATPase operons. *ISME J.* **13**, 3067-3079 (2019).

Wang, S., Wang, Y., Feng, X., Zhai, L., & Zhu, G. Quantitative analyses of ammonia-oxidizing archaea and bacteria in the sediments of four nitrogen-rich wetlands in China. *Appl. Microbiol. Biot.* **90**(2), 779-787 (2011).

115: specify which denitrification gene(s) as there are several as it's a stepwise reaction. If you're looking at NIR, both NIR_k and NIR_s?

Response:

Yes. Here the denitrification genes represent total *nir* (*nirS* plus *nirK*). We have clarified it to “denitrification gene *nir*”. (Line 126)

116: again, add ref that this is "traditionally" believed. It's important to clarify here that AOB could be using denitrification enzymes to produce N₂O. Probably needs the context that (at least in marine AOA) we now know that N₂O could also come from hydroxylamine oxidation. I see you mentioned this on line 120 - I suggest moving it earlier as it makes your results a lot easier to understand.

Response:

Good suggestion. We added references and move the “the qPCR assays cannot distinguish both *nirK* and *nirS* genes in NH₄⁺-derived and NO₃⁻-derived processes” earlier to make our results easier to understand.

(Lines 112-114)

References:

Beaulieu, J. J., Tank, J. L., Hamilton, S. K., Wollheim, W. M., Hall, R. O., Mulholland, P. J., & Thomas, S. M. Nitrous oxide emission from denitrification in stream and river networks. *Proc Natl. Acad. Sci. U.S.A.* **108**(1), 214-219 (2011).

Woodward, K. B., Fellows, C. S., Conway, C. L., & Hunter, H. M. Nitrate removal, denitrification and nitrous oxide production in the riparian zone of an ephemeral stream. *Soil Biol. Biochem.* **41**(4), 671-680 (2009).

117: Is this the ratio of transcriptions? It's a bit confusingly worded but I take this to mean if there are more N₂O reductase transcriptions than NO reductase transcriptions, the microbes must be reducing N₂O from elsewhere? I'm not an expert in what controls the transcription of N₂O reductase. is it N₂O conc in the cell? low dissolved oxygen? Something else? I know the word limit it tight here but I think you need to explain this more.

Response:

Good question.

Yes, this is the abundance ratio of transcriptions (gene *nirB* to *nirZ*). N₂O-producing related genes

involved in the microbial N-cycling processes mainly include the ammonia monooxygenase gene (*amoA*, $\text{NH}_4^+ \rightarrow \text{NH}_2\text{OH}$) and hydroxylamine oxidoreductase gene (*hao*, $\text{NH}_2\text{OH} \rightarrow \text{NO}_2^-$) in nitrifier nitrification (NN) pathway, the nitrite reductase gene (*nirK* and *nirS*, $\text{NO}_2^- \rightarrow \text{NO}$) and nitric-oxide reductase gene (*norB* and *norC*, $\text{NO} \rightarrow \text{N}_2\text{O}$) in nitrifier denitrification (ND), nitrification-coupled denitrification (NCD) and heterotrophic denitrification (HD) pathways. Additionally, nitrous-oxide reductase gene (*nosZ*, $\text{N}_2\text{O} \rightarrow \text{N}_2$) is involved in NCD and HD pathways. There are many genes related to produce N_2O , but the only known gene for N_2O reduction is *nosZ* gene in HD.

In this study, the transcriptional abundance of *nosZ* gene was higher than that of *norB*, indicating that there were other N_2O producing sources, such as NN and abiotic processes. Results by C_2H_2 -inhibitor method revealed that the contribution of abiotic process only accounted for less than 8% (Supplementary Fig. S1). The contribution of NN pathway was measured by ^{15}N - ^{18}O dual-isotope tracing method. This suggests that NH_4^+ -derived, rather than NO_3^- -derived process, might also be the main N_2O production pathway. We also refer to it in the next sentence. In the revised manuscript, we have clarified this confusing sentence to make it easy to understand. **(Lines 98-101)**

123-126 this really clearly explains some of the things I thought were confusing/misleading about your work above. Suggest putting this in the intro so the reader can follow.

Response:

Very good suggestion. Revised.

126: Are you counting the same nor genes in both pools (nitrifier pathway and lumped denitrifier pathways)? There's no way to tell the nor genes apart, right? Does a paired t test make sense if you're using the same data in both populations?

I think what's missing from this analysis is an estimate of how much N_2O you'd expect to get from HAO vs NOR for every molecule that goes through. Does anyone know this? What controls it? (So if you have really high HAO but only 1/10 000 molecules going through becomes N_2O , is that a significant pathway of N_2O production?)

Response:

We are grateful for the professional comment.

Based on your great suggestion, we have clarified ammonia oxidation and heterotrophic denitrification to NH_4^+ -derived and NO_3^- -derived N_2O production processes, respectively, which were more precise. The NH_4^+ -derived N_2O production process includes NN, ND and NCD pathways, while NO_3^- -derived process represents lumped HD pathway. HAO oxidised NH_2OH to NO_2^- , where some of the NH_2OH and/or NO_2^- are converted to the by-product - N_2O (NN pathway) (Wan et al., 2023; Qin et al., 2023). Conversely, NOR reduced NO to the direct product - N_2O in the ND, NCD and HD pathways. NCD and HD pathways share the similar *nor* gene and belong to the same denitrifiers. Therefore, we could not distinguish them. However, through combining species and functional gene annotation based on metagenomic sequencing, we were able to distinguish the *nor*-ND genes of nitrifiers from denitrifiers. In fact, we calculated the same *nor* gene in both NH_4^+ -derived NCD and NO_3^- -derived HD pathways to obtain the maximum N_2O potential of both pathways. However, the N_2O production efficiency of HAO is still unclear. Therefore, the abundance of the *hao* gene cannot be directly compared to *nor* gene, representing the maximum N_2O potential. Thanks for your critical comment. We have removed this part in the revised version.

We have considered your comment very carefully. Metagenomic sequencing alone cannot accurately determine the N_2O -producing efficiency per molecule. Thus, we measured the N_2O production rates of the NN, ND, NCD and HD pathways in the Tang River (Figure 2 in our manuscript). The average rates of N_2O

production in NN, ND, NCD and HD pathways are 13.16, 41.59, 18.32 and 26.93 ng N g⁻¹ d⁻¹, respectively. Results indicated that the NH₄⁺-derived pathways produced more N₂O than the NO₃⁻-derived pathway. Furthermore, we obtained the same results on a regional scale for riparian zone and open water sediments along transects of hyporheic zones (**Figure 2**) and in paddy soil (Qin et al., 2023).

Wan et al., *Proc. Natl. Acad. Sci. U S A* 2023

References:

- Qin, Y., Wang, S. Y., Wang, X. M., Liu, C. L., & Zhu, G. B. Contribution of ammonium-induced nitrifier denitrification to N₂O in paddy fields. *Environ. Sci. Technol.* **57**(7), 2970–2980 (2023).
- Wan, X. S., Hou, L., Kao, S.-J., Zhang, Y., Sheng, H.-X., Shen, H., Tong, S., Qin, W., Ward, B. B. Pathways of N₂O production by marine ammonia-oxidizing archaea determined from dual-isotope labeling. *Proc. Natl. Acad. Sci. U S A* **120**(11): e2220697120 (2023).

135: suggest same edits as above to clarify you can't really tell which pathway is being used, just that you added 15N-NH₄⁺ and it's ending up in N₂O. NCD is a type of "heterotrophic denitrification".

Response:

The misunderstanding problem due to the unclear terms again. Based on your kind suggestions and other representative literatures (Zhu et al., *Proc. Natl. Acad. Sci. U.S.A.* 2013; Kool et al., *Eur. J. Soil Sci.* 2010; Kool et al., *Soil Biol. Biochem.* 2011), we have clarified the ammonia oxidation and heterotrophic denitrification processes to NH₄⁺-derived and NO₃⁻-derived N₂O production processes, respectively. The NH₄⁺-derived N₂O included N₂O produced via NN, ND, and NCD pathways. The NO₃⁻-derived process meant N₂O produced via HD pathways. We think the new terms are more precise and make our manuscript more readability.

References:

- Zhu, X., Burger, M., Doane, T. A., & Horwath, W. R. Ammonia oxidation pathways and nitrifier denitrification are significant sources of N₂O and NO under low oxygen availability. *Proc. Natl. Acad. Sci. U.S.A.* **110**(16), 6328-6333 (2013).
- Kool, D. M., Wrage, N., Zechmeister-Boltenstern, S., et al. Nitrifier denitrification can be a source of N₂O from soil: a revised approach to the dual-isotope labelling method. *Eur. J. Soil Sci.* **61**(5), 759-772 (2010).
- Kool, D. M., Dolfing, J., Wrage, N., & Van Groenigen, J. W. Nitrifier denitrification as a distinct and significant source of nitrous oxide from soil. *Soil Biol. Biochem.* **43**(1), 174-178 (2011).

138: "at seasonal scale" sounds wrong to me. How about "on the seasonal scale"?

Response:

Agree. Revised. **(Line 140)**

140: Again, do you have dissolved O₂ measurements? I would expect these to be highly negatively correlated to NH₄⁺.

Response:

Described earlier. **(Response letter Page 10)**

142: I keep thinking about how to word this and maybe the best way is "N₂O production is NH₄⁺-limited" (??) It would be easier to think about what's going on if we saw NH₄⁺ and NO₃⁻ (and dissolved O₂) in the samples.

Response:

Agree. The unclear and confusing terms again. Based on your kind suggestions, we have clarified the ammonia oxidation and heterotrophic denitrification processes to NH₄⁺-derived and NO₃⁻-derived N₂O production processes, respectively, which also had been widely used in the previous articles (Kool et al., 2011; Zhu et al., 2013; Wrage et al., 2005; Duan et al., 2019; Qin et al., 2023; Yuan et al. 2023). The NH₄⁺-derived N₂O included N₂O produced via NN, ND, and NCD pathways. The NO₃⁻-derived process meant N₂O produced via HD pathways. In addition, both NH₄⁺-derived and NO₃⁻-derived N₂O are produced by multiple and various microorganisms. NH₄⁺ is the main factor but not the only factor affecting the hyporheic N₂O production, such as pH and temperature (Hu et al. 2022; Jiang et al. 2023).

We think the new terms are more precise and make our manuscript more readability.

References:

- Kool, D. M., Dolfing, J., Wrage, N., & Van Groenigen, J.W. Nitrifier denitrification as a distinct and significant source of nitrous oxide from soil. *Soil Biol. Biochem.* **43**(1), 174-178 (2011).
- Zhu, X., Burger, M., Doane, T. A., & Horwath, W. R. Ammonia oxidation pathways and nitrifier denitrification are significant sources of N₂O and no under low oxygen availability. *Proc. Natl. Acad. Sci. U.S.A.* **110**(16), 6328-6333 (2013).
- Wrage, N., van Groenigen, J. W., Oenema, O., & Baggs, E. M. A novel dual-isotope labelling method for distinguishing between soil sources of N₂O. *Rapid Commun. Mass Spectrom.* **19**(22), 3298-3306 (2005).
- Duan, P., Song, Y., Li, S., & Xiong, Z. Responses of N₂O production pathways and related functional microbes to temperature across greenhouse vegetable field soils. *Geoderma* **355**, 113904 (2019).
- Qin, Y., Wang, S. Y., Wang, X. M., Liu, C. L., & Zhu, G. B. Contribution of ammonium-induced nitrifier denitrification to N₂O in paddy fields. *Environ. Sci. Technol.* **57**(7), 2970-2980 (2023).
- Yuan, D. D., Zheng, L., Liu, Y. X., Cheng, H. G., Ding, A. Z., Wang, X. M., Tan, Q. Y., Wang, X., Xing, Y. Z., Xie, E., Wu, H. M., Wang, S. Y., & Zhu, G. B. Nitrifiers cooperate to produce nitrous oxide in Plateau wetland sediments. *Environ. Sci. Technol.* **57**(1): 810-821 (2023).
- Hu, L., Dong, Z.X., Wang, Z., Xiao, L.W., & Zhu, B. The contributions of ammonia oxidizing bacteria and archaea to nitrification-dependent N₂O emission in alkaline and neutral purple soils. *Sci. Rep.* **12**(1), 19928 (2022).
- Jiang, Z., Tang, S.Y., Liao, Y.H., Li, S.J., Wang, S., Zhu, X.F., & Ji, G.D. Effect of low temperature on contributions of ammonia oxidizing archaea and bacteria to nitrous oxide in constructed wetlands. *Chemosphere* **313**, 137585 (2023).

154: change to "dual-isotope tracing". I have to look up to see if you added labelled NH₄ or just assumed some isotopic fractionation factors...

Response:

Revised. **(Line 156)**

This paragraph is interesting! It seems strange to do it on a different river than the river with the more intensive sampling. I assume there was a logistical reason. But it makes me wonder if it's not better to do more intensive microbial genetics on samples from Tang R. Is the Tang R included in the regional results above? Might be worth bring it all the data from that river together so it's easier to follow.

Response:

Thank you very much for this comment.

Firstly, a regional-scale study has been conducted on five rivers across Baiyangdian riverine by using C₂H₂-inhibitor method. Of which, the Tang River is included in the regional scale study above, as shown in **Fig 2**.

Secondly, we used the ^{15}N - ^{18}O dual-isotope tracing technology to verify these results in one of the five streams - Tang River. Most importantly, we have achieved the consistent results that the NH_4^+ -derived N_2O production was significantly higher than NO_3^- -derived N_2O process. It indicated NH_4^+ -derived process, rather than NO_3^- -derived process, is the dominant hyporheic N_2O source. In the revised manuscript, all the raw data has been supplied in the supplementary tables.

160: why "should" be and how do you know they're not significantly different? My understanding is that isotopes will not distinguish NN and NCD but please clarify.

Response:

Because we have already provided a detailed description earlier, we will only provide a brief description here. In this study, we used the C_2H_2 -inhibitor method, ^{15}N tracing *semi-in-situ* sediment-core incubation, and developed ^{15}N - ^{18}O dual-tracing method to distinguish NH_4^+ -derived and NO_3^- -derived N_2O processes. With ^{18}O tracers added, the ^{15}N - ^{18}O dual-tracing method can further distinguish NH_4^+ -derived process to NN, ND and NCD pathways, and calculated the maximum and minimum contribution of the three pathways. This method has been widely used and approved. Here, we wanted to express that there was no significant difference of the average contributions between NN and NCD. To avoid misunderstandings, we have deleted these words.

163: add refs for this idea - what recent papers? Should also but put in intro. I see you don't ref anything from Bradley Eyre - he has a lot of dual isotope N_2O papers (I think mostly in estuaries). E.g. <https://www.sciencedirect.com/science/article/pii/S0016703721002994>

Response:

Many thanks for your comments on how to improve our manuscript.

Professor Bradley Eyre is an outstanding biogeochemist. He and his team studied the flow of carbon and nitrogen through coastal ecosystems, focusing on global change issues, such as eutrophication, climate change, and greenhouse gases. Wells & Eyre (2021) studied on the stream flow dynamics that regulates the balance between biological NO_3^- and N_2O production using *in-situ* stable isotopes ($\delta^{15}\text{N}$ - NO_3^- and $\delta^{18}\text{O}$ - NO_3^-) and Modelling. They emphasized the importance of nitrification, rather than denitrification, even in strongly heterotrophic stream through theoretical modelling. In our study, we used the isotope labeling method to analyze microbial composition and activity directly. We are pleased to see consistent results based on different research ideas, methods and study sites. In the revised manuscript, we have cited this wonderful work by Professor Eyre's team as well as other relevant publications.

References:

- Qin, Y., Wang, S. Y., Wang, X. M., Liu, C. L., & Zhu, G. B. Contribution of ammonium-induced nitrifier denitrification to N₂O in paddy fields. *Environ. Sci. Technol.* **57**(7), 2970–2980 (2023).
- Zhang, G. L., Liu, S. M., Casciotti, K. L., Forbes, M. S., Gu, X. J., Ren, Y. Y., & Zheng, W. J. Distribution of concentration and stable isotopic composition of N₂O in the shelf and slope of the Northern South China Sea: Implications for production and emission. *J. Geophys. Res. Oceans* **124**, 6218–6234 (2019).
- Yuan, D. D., Zheng, L., Liu, Y. X., Cheng, H. G., Ding, A. Z., Wang, X. M., Tan, Q. Y., Wang, X., Xing, Y. Z., Xie, E., Wu, H. M., Wang, S. Y., & Zhu, G. B. Nitrifiers cooperate to produce nitrous oxide in Plateau wetland sediments. *Environ. Sci. Technol.* **57**(1): 810-821 (2023).
- Cao, Y., Wang, X., Zhang, X., Misselbrook, T., & Ma, L. Nitrifier denitrification dominates nitrous oxide production in composting and can be inhibited by a bioelectrochemical nitrification inhibitor. *Bioresour. Technol.* **341**(24), 125851 (2021).
- Wells, N.S. & Eyre, B.D. Flow regulates biological NO₃⁻ and N₂O production in a turbid sub-tropical stream. *Geochim. Cosmochim. Acta* **306**, 124-142 (2021).
- Wells, N.S. & Eyre, B.D. δ¹⁵N patterns in three subtropical estuaries show switch from nitrogen “reactors” to “pipes” with increasing degradation. *Limnol. Oceanogr.* **64**, 860-876 (2019).
- Erlor, D.V., Santos, I.R., Zhang, Y., Tait, D.R., Befus, K.M., Hidden, A., Li, L., & Eyre B.D. Nitrogen transformations within a tropical subterranean estuary. *Mar. Chem.* **164**, 38-47 (2014).
- Murray, R.H., Erlor, D.V., & Eyre, B.D. Nitrous oxide fluxes in estuarine environments: Response to global change. *Glob. Chang. Biol.* **21**, 3219-3245 (2015).
- Murray, R., Erlor, D., Rosentreter, J., Maher, D., & Eyre, B.D. A seasonal source and sink of nitrous oxide in mangroves: Insights from concentration, isotope, and isotopomer measurements. *Geochim. Cosmochim. Acta* **238**, 169-192 (2018).

165: *I'm not a microbiologist but this seems more convincing to me than the gene transcriptions. I would suggest restructuring the paper so the data that are most conclusive (this and dual isotopes) are first and highlighted. The rest can always be put in supplementary material.*

Response:

This issue has already been explained above. Thanks for your constructive comments.

196: *is it possible to have a denitrifying bacteria with no NO reductase? Does it just stop at NO and that's how much energy it makes? I have never heard of this. I hope the other reviewer is a microbiologist who knows how to evaluate this claim but it strikes me as so odd.*

Response:

Yes. In fact, denitrifying bacteria without N₂O reductase are possible. The similar results also have been reported in studies on N₂O hot moment in peatlands by our team (Wang et al., ISME J. 2023), on anammox granules in a wastewater treatment system by distinguished microbiologist Professor Mike Jetten and his partners (Speth et al., Nat. Commun. 2016), and on truncated denitrifiers (lacking one or more denitrification genes) and N₂O cycling in tundra soils by another microbiology research team (Pessi et al., bioRxiv 2020). Meanwhile, metagenomics also reveals that NO₂⁻ reduction and NO reduction genes do not coexist in most denitrifying bacteria (Wang et al., ISME J. 2023; Speth et al., Nat. Commun. 2016; Pessi et al., bioRxiv 2020; This study), indicating the presence of denitrifying bacteria with NO as the end product.

Denitrification is a series of enzymatic steps in which NO₃⁻ is sequentially reduced to NO₂⁻, NO, N₂O, and N₂. Denitrification traits are found in a wide range of archaea, bacteria, and fungi, most of which are facultative anaerobes that convert N oxides as electron acceptors under anoxic conditions (Zumft et al., 1997). To date, only about one-third of the genomes of cultured denitrifiers sequenced encode the full set of enzymes required for complete denitrification (Graf et al., 2014; Pessi et al., bioRxiv 2020). Denitrification has been demonstrated

to be a modular process, with some microorganisms able to completely reduce NO_3^- to N_2 , while others can only perform one or a subset of steps (Zhang et al., 2023; Fuchsman et al., 2017; Graf et al., 2014; Wallenstein et al., 2006; Bertagnolli et al., 2020).

Although this type of denitrifying bacteria is very likely to exist, we have removed the relative abundance values in the revised manuscript to make it more readable. Thanks for your good and critical comment.

Fig. 4: Metabolic reconstruction and features of N_2O production- and reduction-related microorganisms identified in seasonally frozen peatland. (Wang et al., *ISME J.* 2023)

89% bins with no NO reductase, except bins 46 and 50

Figure 4: Schematic overview of N conversions in the Olburgen PNA reactor. (Speth et al., *Nat. Commun.* 2016)

④ CFX1 with no NO reductase

Fig. 2 Metabolic potential for denitrification in metagenome-assembled genomes 152 (MAGs) from tundra soils. (Pessi et al., *bioRxiv.* 2020)

few coexist of NO_2^- reductase and NO reductase

References:

- Wang, X.M., Wang, S.Y., Yang, Y.H., Tian, H.Q., Jetten, M.S.M., Song, C.Q., & Zhu, G.B. Hot moment of N_2O emissions in seasonally frozen peatlands. *ISME J.* **17**, 792–802 (2023).
- Speth, D., in't Zandt, M.H., Guerrero-Cruz, S., Dutilh, B.E., & Jetten, M.S.M. Genome-based microbial ecology of anammox granules in a full-scale wastewater treatment system. *Nat. Commun.* **7**, 11172 (2016).
- Pessi, I.S., Viitamäki, S., Rasimus, E.E., Delmont, T.O., Luoto, M., & Hultman, J. Truncated denitrifiers dominate the denitrification pathway in tundra soil metagenomes. *bioRxiv.* 419267 (2020).
- Zumft, W. G. Cell biology and molecular basis of denitrification. *Microbiol. Mol. Biol. Rev.* **61**, 533–616 (1997).
- Graf, D. R. H., Jones, C. M., & Hallin, S. Intergenomic comparisons highlight modularity of the denitrification pathway and underpin the importance of community structure for N_2O emissions. *PLoS one* **9**, e114118 (2014).
- Zhang, I.H., Sun, X., Jayakumar, A. et al. Partitioning of the denitrification pathway and other nitrite metabolisms within global oxygen deficient zones. *ISME Commun.* **3**, 76 (2023).
- Fuchsman, CA, Devol AH, Saunders JK, McKay C, Rocap G. Niche partitioning of the N cycling microbial community of an offshore oxygen deficient zone. *Front Microbiol.* **8**: 2384 (2017).
- Wallenstein, M. D., Myrold, D. D., Firestone, M., & Voytek, M. Environmental controls on denitrifying communities and denitrification rates: insights from molecular methods. *Ecol. Appl.* **16**, 2143–2152 (2006).
- Bertagnolli, A.D., Konstantinidis, K.T., & Stewart, F.J. Non-denitrifier nitrous oxide reductases dominate marine biomes. *Environ Microbiol Rep.* **12**: 681–92 (2020).

204: is this also true for microbes (your ref is about plants)?

Response:

Yes. It's true for microbes. The references have been modified to Moir et al., 2001 and Fukuda et al., 2015.

Many Thanks.

References:

Moir, J.W. & Wood, N.J. Nitrate and nitrite transport in bacteria. *Cell Mol. Life Sci.* **58**(2), 215-24 (2001).

Fukuda, M., Takeda, H., Kato, H.E., Doki, S., Ito, K., Maturana, A.D., Ishitani, R., & Nureki, O. Structural basis for dynamic mechanism of nitrate/nitrite antiport by NarK. *Nat. Commun.* **6**, 7097 (2015).

208: change consuming to consumption. My understanding was that AOB don't really do nitrite reduction for energy but rather to remove toxic NO₂⁻. I'm happy to be proven wrong but please provide some references.

Response:

The word “consuming” has been revised to “consumption”. (**Line 208**)

We apologize for this unclear sentence, and thank you for your wonderful comment, which gave us more inspiration. Compared with denitrifiers (NO₃⁻-derived process), the objective reasons for ammonia oxidizer (NH₄⁺-derived process) to produce more N₂O are: i) energy saving on substrate NO₂⁻/NO₃⁻ transport, ii) self-detoxification function for NO₂⁻ toxicity, iii) more sufficient energy supply, and iv) N₂O as end-product due to lack of N₂O reduction genes.

Firstly, N₂O production via nitrifier denitrification (ND) pathway is definitely an ATP consumption process, but not an energy synthesis process. Here, we would like to say that intracellular NO₂⁻ produced by ammonia oxidizers can be used directly as the substrate for the ND pathway (Kim et al., 2010), whereas denitrifying bacteria must consume ATP to transport ambient NO₃⁻ into the cell via the HD pathway (Moir & Wood, 2012). Thus, the ND pathway produced N₂O with less energy than the HD pathway, and it is more likely to occur than the HD pathway.

Secondly, we also agree with your explanation that AOB reduced nitrite (ND) to remove toxic NO₂⁻. Actually. In fact, we noticed the mechanism of NO₂⁻ toxicity and microbial self-detoxification as early as when the corresponding author Prof. Zhu was still a doctoral candidate. In his study, Prof. Zhu reviewed the toxicity of NO₂⁻ and the detoxification of AOB during the biological N removal with nitrification and denitrification via nitrite pathway (Peng & Zhu, 2006), which has been cited more than 700 times. We also cited references from other teams (Lu et al., 2023; Liu et al., 2018) and supplemented this explanation in the revised manuscript.

Thirdly, AOB can produce energy through aerobic respiration, converting the chemical energy of NH₄⁺ to NO₂⁻ (not nitrite reduction) into electron potential energy, driving the synthesis of ATP (Arp et al., 2002; Wright et al., 2023). The synthetic energy is then used for nitrite reduction and N₂O production (ND pathway, NO₂⁻ → NO → N₂O) (Walker et al., 2010). However, denitrifying bacteria produced energy from anaerobic fermentation, which is limited by organic matter and less efficient at energy production per molecular substrate than aerobic respiration. In addition, the synthetic energy of denitrifying bacteria is not only used for transmembrane transport of nitrate, but also for a series of nitrate reduction processes (NO₃⁻ → NO₂⁻ → NO → N₂O → N₂) (Feng et al., 2008). Therefore, we propose that the energy supply of the ND pathway is more sufficient than that of the HD pathway.

Fourthly, to date, there have been no reports of AOB containing N₂O reduction genes, which indicates that AOB can only produce and emit N₂O via NN and ND pathways without further reducing it to N₂ or others (Huang et al., 2014). Denitrifying bacteria have *nosZ* gene encoding N₂O reductase, and hence, N₂O produced via NCD and HD pathways can be further reduced to N₂ (Friedl et al., 2020).

We have supplemented the mechanism of NO₂⁻ toxicity and microbial self-detoxification in the revised manuscript. (**Lines 203-234**)

References:

- Kim, D.-J. & Kim, S.-H. Effect of nitrite concentration on the distribution and competition of nitrite-oxidizing bacteria in nitrification reactor systems and their kinetic characteristics. *Water Res.* **40**, 887-894 (2006).
- Moir, J.W. & Wood, N.J. Nitrate and nitrite transport in bacteria. *Cell Mol Life Sci.* **58(2)**, 215-224 (2001).
- Peng, Y.Z. & Zhu, G.B. Biological nitrogen removal with nitrification and denitrification via nitrite pathway. *Appl. Microbiol. Biotechnol.* **73(1)**:15-26 (2006).
- Lu, X., Wang, Z.Y., Duan, H.R., Wu, Z.P., Hu, S.H., Ye, L., Yuan, Z.G., & Zheng, M. Significant production of nitric oxide by aerobic nitrite reduction at acidic pH. *Water Res.* 230 (2023).
- Liu, B., Terashima, M., Quan, N. T., Ha, N. T., Van Chieu, L., Goel, R., & Yasui H. High nitrite concentration accelerates nitrite oxidizing organism's death. *Water Sci. Technol.* **77(12)**, 2812-2822 (2018).
- Arp, D.J., Sayavedra-Soto, L.A. & Hommes, N.G. Molecular biology and biochemistry of ammonia oxidation by *Nitrosomonas europaea*. *Arch. Microbiol.* **178**, 250–255 (2002).
- Wright, C.L. & Lehtovirta-Morley, L.E. Nitrification and beyond: metabolic versatility of ammonia oxidising archaea. *ISME J.* **17**, 1358–1368 (2023).
- Walker, C. B., de la Torre, J. R., Klotz, M. G., Urakawa, H., Pinel, N., Arp, D. J., & Stahl, D. A. Nitrosopumilus maritimus genome reveals unique mechanisms for nitrification and autotrophy in globally distributed marine crenarchaea. *Proc. Natl. Acad. Sci. U.S.A.* 107(19), 8818-8823 (2010).
- Feng, X. M., Cao, L. J., Adam, R. D., Zhang, X. C., & Lu, S. Q. The catalyzing role of PPK in *Giardia lamblia*. *Biochem. Biophys. Res. Commun.* 367(2), 394-398 (2008).
- Huang, T., Gao, B., Hu, X. K., Lu, X., Well, R., Christie, P., & Ju, X. T. Ammonia-oxidation as an engine to generate nitrous oxide in an intensively managed calcareous Fluvo-aquic soil. *Sci. Rep.* 4, 3950 (2014).
- Friedl, J., Scheer, C., Rowlings, D. W., Deltedesco, E., Gorfer, M., De Rosa, D., Keiblinger, K. M. Effect of the nitrification inhibitor 3,4-dimethylpyrazole phosphate (DMPP) on N-turnover, the N₂O reductase-gene *nosZ* and N₂O:N₂ partitioning from agricultural soils. *Sci. Rep.* 10(1) (2020).

212: My understanding was that AOB are generally chemolithoautotrophs. You cite a paper for one unusual marine species that seems to also be able to do aerobic respiration. Reword or clarify if this is common for AOB and provide appropriate references.

Response:

Agree. It is not appropriate that we use a special marine species as a sample to explain the finding. In the revised manuscript, we added the appropriate references about a common AOB - *Nitrosomonas europaea* (Arp et al., *Arch. Microbiol.* 2002) and a review of AOA (Wright et al., *ISME J.* 2023) to support it.

References:

- Arp, D.J., Sayavedra-Soto, L.A. & Hommes, N.G. Molecular biology and biochemistry of ammonia oxidation by *Nitrosomonas europaea*. *Arch. Microbiol.* **178**, 250–255 (2002).
- Wright, C.L., Lehtovirta-Morley, L.E. Nitrification and beyond: metabolic versatility of ammonia oxidising archaea. *ISME J.* **17**, 1358–1368 (2023).

214: You have not convinced me that AOB produce N₂O for energy. If they did, why not run denitrification to completion and make N₂? but they don't seem to have an N₂O reductase gene.

Response:

We apologize for not explaining our viewpoint clearly in our manuscript again.

AOB can produce energy through aerobic respiration, converting the chemical energy of NH₄⁺ to NO₂⁻ (not nitrite reduction) into electron potential energy, driving the synthesis of ATP (Arp et al., 2002; Wright et al., 2023). The synthetic energy is then used for nitrite reduction and N₂O production (ND pathway, NO₂⁻ → NO

→ N₂O) (Walker et al., 2010). N₂O production via nitrifier denitrification (ND) pathway is actually an ATP-consuming process, rather than an energy-synthesizing process. The NO₂⁻ produced by ammonia oxidizers can be used directly as substrate of ND pathway (Kim et al., 2010) without transmembrane transport, whereas the denitrifying bacteria have to consume at least two molecules of ATP (Marschner, 2012) to transport ambient NO₃⁻ in the HD pathway. Thus, ND pathway used less energy than HD to produce N₂O, and it is more likely to occur than HD pathway. In addition, to date, there have been no reports of AOB containing N₂O reduction genes, indicating AOB can only produce nitrous oxide without removing it. In summary, NH₄⁺-derived process may produce more N₂O than NO₃⁻-derived process.

References:

Kim, S. W., Miyahara, M., Fushinobu, S., Wakagi, T., & Shoun, H. Nitrous oxide emission from nitrifying activated sludge dependent on denitrification by ammonia-oxidizing bacteria. *Bioresour. Technol.* 101(11), 3958-3963 (2010).
Marschner, P. Marschner's Mineral Nutrition of Higher Plants. 3rd Edition, Academic Press, Cambridge. (2012).

215: "In addition, the genetic diversity of nitrifying and denitrifying bacteria is different." Very unclear, just remove this sentence and say, "additionally, nitrifiers have the potential to produce high amounts of N₂O because they do not have N₂O reductase" or whatever. [Again, is it true that NO nitrifiers have N₂O reductase? Most? a few?]

Response:

Up to now, there is no report that nitrifying bacteria have N₂O reductase gene. According to your good suggestions, we have revised. Many thanks. **(Lines 230-231)**

227: is temporal scale, seasonal? Does the seasonality matter here? Unclear which methods you're using on the global scale - this controls how strongly you can state your results!

Response:

Here we want to say that we sampled under different seasons and temperature conditions to verify the findings we have obtained above. As your suggestion, it is more appropriate to use "seasonal scale" here.

On the site-scale and regional-scale studies, both C₂H₂-inhibitor and two isotope-tracing methods were used together to determine the rate and pathway of N₂O production. Results showed there was no significant difference between different methods, i.e. i) C₂H₂-inhibitor method and ¹⁵N-tracing *semi-in-situ* sediment core incubation on the site-scale study, ii) C₂H₂-inhibitor method and ¹⁵N-¹⁸O dual-isotope tracing incubation in Tanghe River on Regional scale. The above results indicated that these methods are feasible for hyporheic sediments. Thus, we applied the C₂H₂-inhibitor method, a more economical and controllable method, to analyze the production of N₂O in global-scale streams. This has been added in the revised manuscript. **(Line 242)**

246: I am not convinced this is the current opinion! Suggest rewording to indicate the NCD and ND use the same genes as denitrification (or are denitrification). There is some evidence HAO can produce N₂O in nitrifiers but unclear how much. You might need to coin a term here like "nitrifier-supplied denitrification" or something to combine NCD and ND if you can't tell them apart.

Response:

Thank you for your reminder. We make a mistake of overgeneralization here.

This incorrect sentence has been modified to "NH₄⁺-derived process, rather than NO₃⁻-derived process, is the dominant hyporheic N₂O source". The term "NH₄⁺-derived process" could include NN, NCD and ND pathways clearly, thus it is more appropriate here. **(Lines 261-262)**

References:

Duan, P., Song, Y., Li, S., & Xiong, Z. Responses of N₂O production pathways and related functional microbes to temperature

across greenhouse vegetable field soils. *Geoderma* **355**, 113904 (2019).

Hu, L., Dong, Z.X., Wang, Z., Xiao, L.W., & Zhu, B. The contributions of ammonia oxidizing bacteria and archaea to nitrification-dependent N₂O emission in alkaline and neutral purple soils. *Sci. Rep.* **12**(1), 19928 (2022).

Jiang, Z., Tang, S.Y., Liao, Y.H., Li, S.J., Wang, S., Zhu, X.F., & Ji, G.D. Effect of low temperature on contributions of ammonia oxidizing archaea and bacteria to nitrous oxide in constructed wetlands. *Chemosphere* **313**, 137585 (2023).

Kool, D. M., Dolfing, J., Wrage, N., & Van Groenigen, J.W. Nitrifier denitrification as a distinct and significant source of nitrous oxide from soil. *Soil Biol. Biochem.* **43**(1), 174-178 (2011).

Qin, Y., Wang, S. Y., Wang, X. M., Liu, C. L., & Zhu, G. B. Contribution of ammonium-induced nitrifier denitrification to N₂O in paddy fields. *Environ. Sci. Technol.* **57**(7), 2970–2980 (2023).

Wrage, N., van Groenigen, J. W., Oenema, O., & Baggs, E. M. A novel dual-isotope labelling method for distinguishing between soil sources of N₂O. *Rapid Commun. Mass Spectrom.* **19**(22), 3298-3306 (2005).

Yuan, D. D., Zheng, L., Liu, Y. X., Cheng, H. G., Ding, A. Z., Wang, X. M., Tan, Q. Y., Wang, X., Xing, Y. Z., Xie, E., Wu, H. M., Wang, S. Y., & Zhu, G. B. Nitrifiers cooperate to produce nitrous oxide in Plateau wetland sediments. *Environ. Sci. Technol.* **57**(1): 810-821 (2023).

Zhu, X., Burger, M., Doane, T. A., & Horwath, W. R. Ammonia oxidation pathways and nitrifier denitrification are significant sources of N₂O and no under low oxygen availability. *Proc. Natl. Acad. Sci. U.S.A.* **110**(16), 6328-6333 (2013).

Suggest comparing these results to IPCC estimates, e.g. all reactive N is nitrified, half is denitrified. Using their EF5s how does this compare to your results? Nothign wrong with the Nevison paper but it is 23 years old and has been revisited, e.g. by Maavara et al. (2018).

Response:

Thanks for your kind comments. Frankly, we think it's better to focus the core of this article on "ammonia oxidation pathways are the dominant hyporheic N₂O sources" because there are too many analyses in this paper, just as the reviewer said above. As for the academic value and environmental significance of this finding, such as comparing these results to IPCC estimates, we think it would be more appropriate for other studies to answer.

Of course, the too old reference has been replaced with the one suggested.

Reference:

Maavara, T., Lauerwald, R., Laruelle, G. G., Akbarzadeh, Z., Bouskill, N. J., Van Cappellen, P., & Regnier, P. Nitrous oxide emissions from inland waters: Are IPCC estimates too high? *Global Change Biology* **25**, 473–488 (2019).

315: is in-situ O₂ atmospheric or what was measured in the field when the sediment was collected? O₂ will have a big role to play in whether nitrification or denitrification dominates.

Response:

We did not fully measure dissolved oxygen in the water column, because all of our samples were sediments, and the primary goal was to study the rate and pathway of N₂O production in stream sediments. Based on our limited data on dissolved oxygen of the water column (7.73 ± 0.39 mg L⁻¹), we were unable to obtain a reliable relationship between oxygen and N₂O concentration. However, we measured the *in-situ* O₂ content in sediments (0.34 ± 0.07 mg L⁻¹) by using a Pocket Oxygen Meter (FireStingGO2, PyroScience GmbH, Germany). Pearson correlation analysis also showed that the most important physicochemical factor affecting N₂O production was ammonium rather than oxygen content in sediments. In addition, Zhu Xia (Zhu et al., 2013) also found that as long as oxygen was present, even at a very low level (0.5% O₂), the NH₄⁺-derived N₂O production process contributed a large amount of N₂O, even exceeding the process of NO₃⁻-derived N₂O production. The NO₃⁻-derived process was responsible for all N₂O production at 0% O₂. It also indicated that sediment oxygen is probably not a significant limiting factor for NH₄⁺-derived or NO₃⁻-derived N₂O production dominates in the natural rivers.

Reference:

Zhu, X., Burger, M., Doane, T. A., & Horwath, W. R. Ammonia oxidation pathways and nitrifier denitrification are significant sources of N₂O and no under low oxygen availability. *Proc. Natl. Acad. Sci. U.S.A.* **110**(16), 6328-6333 (2013).

317: Not sure what "completely designed at random vials" means. Please clarify.

Response:

This sentence is intended to express the incubation design of ¹⁵N-¹⁸O dual-isotope tracing methods.

5 g of homogeneous fresh sediment with 25 ml overlying water were placed into **at least 12 parallel 60-mL glass serum vials** (Ochs Laborbedarf, Germany). Four treatments enriched in ¹⁸O and ¹⁵N were completely applied **at random vials** in triplicate. This approach is designed to minimize incubation differences.

In the revised manuscript, we have already deleted this unclear sentence.

Reviewer #2

Noteworthy results: 1) evidence that "nitrifying bacteria contain higher abundances of N₂O production-related genes than denitrifying bacteria"; 2) "ammonia oxidation was the most important hyporheic N₂O source"; 3) "These findings differ from the prevailing opinion that heterotrophic denitrification is the main contributor to N₂O in the riverine hyporheic zone."

Significance: high, paradigm shifting.

Evidence: the evidence provided supports the claims and conclusions well.

Methods/Analysis: sound and repeatable.

Comments:

There are a few, minor typos throughout. Please revise with the copy editor.

L117: 2.2:3.1 is a unique way to display a ratio. Why not 0.7:1?

Response:

Thanks for your nice comments. It has been revised to 0.7:1. Please see the line xx in the revised manuscript.

Conclusion: overall, very good work. Worthy of publication.

Response:

Thank you for your approval of our manuscript.

Reviewer #3

Summary: This study aims to investigate the microbial sources and mechanisms responsible for N₂O production in riparian sediments. Specifically, investigators used various methods including omics, tracer experiments, and geochemistry to parse N metabolism at site-level (Xiaoqinghe River), regional-level (Baiyangdian river network), and global scales. Key findings highlight ammonia as a more important predictor of N₂O production over nitrate, in contrast to a widely held modeling assumption.

Major comments:

1. Data deposition- The manuscript does not contain a data availability statement and should not be considered for publication until this is included. Specifically, this manuscript should not be accepted for publication until raw metatranscriptomic and metagenomic reads, along with the derived genome bins are deposited into a repository such as NCBI or equivalent. Additionally, raw numbers for tracer or physio-chemical characteristics should be reported as an additional data file rather than as averages in flat pdf tables.

Response:

Thank you for pointing out the need for data availability, which can effectively improve the scientific, readable and reliable nature of our manuscript.

We have added a statement of the availability of all numerical data to the manuscript (**Lines 396-401**), including:

1) The raw metagenomic reads and the derived genome bins deposited into NCBI Sequence Read Archive (SRA) under the accession number SAMN33620661-SAMN33620760 (PRJNA943572) and SAMN37879210-SAMN38023784 (PRJNA1031250), respectively, detailed in **Supplementary Tables S8 and S9**;

2) The Source data of physio-chemical characteristics, ¹⁵N isotopic tracing and ¹⁵N-¹⁸O dual-isotope tracing experiments were detailed in the **Supplementary Tables S2, S5 and S7**, respectively.

Thank you again for your kind help and necessary comments.

2. Methodological concerns

a. Details on numbers of samples seem to be left out throughout the entire manuscript and should be added at least in the methods. Specific areas of improvement include:

- *Line 263- how many sediment cores?*

Response:

Agree. Thank you for the nice comment.

The site-scale study was conducted along a transect of the riverine hyporheic zone at Xiaoqinghe stream. At least nine parallel sediment cores (0–20 cm depth) were collected from the riparian zone and open water, respectively. Six sediment cores were *semi-in-situ* incubated for ¹⁵N-tracer assay and the other three for C₂H₂ inhibitor assay and molecular analysis.

The number of sediment cores have been added in **lines 278-282**.

- *Line 266 five sites along each river for the regional study- how many rivers? Were these also collected in triplicate?*

Response:

At the regional-scale study, five rivers in the Baiyangdian riverine networks, namely the Bai River, the Cao River, the Juma River, the Zhulong River, and the Tang River, were collected during dry and rainy seasons. For each river, there are five sampling sites (Figure 2a). Similarity to the site-scale study, all the 25 regional-scale sites (5 rivers * 5 sampling sites) were sampled in triplicate. The related information has been added and clarified

in lines 282-285 of the revised manuscript.

Figure 2a. Overview of the Baiyangdian riverine network and the sampling sites

• In general, each study (site, regional, and global) should have a sentence that states exactly how samples were collected (e.g., in triplicate, where in the river they were collected from, sites per river), the total number of samples generated, and the number of rivers sampled. It may also be nice to have a table of this information per site, regional, and global studies along with the analyses (e.g., was metagenomic performed at every site?) performed on each scale.

Response:

Thank you for your kind comments.

At the site scale, we collected three parallel sediment cores at the riparian zone and open water zone, respectively (Figure 1). At the regional scale, 25 sampling sites (5 rivers * 5 sampling sites) in the Baiyangdian riverine network were sampled in riparian zone and open water sediments, respectively, in both rainy and dry seasons. Therefore, metagenomic analysis was performed at 100 sample sites (5 rivers * 5 sampling sites * 2 zones * 2 seasons) (Figure 2). Based on site and regional scale results, the global-scale study covered 28 low-order agricultural streams, regardless of seasons and zones (Figure 4). All the above sampling sites were collected in triplicate.

Based on your nice suggestion, we have added the main information in lines 278-296 and details in Supplementary Table S1.

• Which analyses were performed on which samples- the N-tracer analyses state it is just the site-specific samples, the N-O tracer studies do not state which samples these were done on, and for the global studies Supplementary table has cited N₂O fluxes. Were the global samples not analyzed but taken from literature values?

Response:

Our research extends from site scale to regional scale and then to global scale. At the site scale, a novel ¹⁵N tracing semi-in-situ sediment-core incubation and an C₂H₂-inhibitor method were used to investigate the N₂O production rate and source at the riparian zone and open water sediments in the Xiaoqinghe River, respectively (Figure 1).

On the regional scale, the C₂H₂-inhibitor method was performed to investigate the N₂O production rate and source in Baiyangdian riverine. Then, the ¹⁵N-¹⁸O tracer studies were applied to the Tanghe River samples, one of the five streams studied in Baiyangdian riverine, aiming to further distinguish the pathways of N₂O production (Figure 2).

On the global scale, we conducted C₂H₂-inhibitor experiments on 28 sampling sites around the world to

investigate the N₂O production rate and source, further validating the main conclusions of site and regional scale studies (**Figure 4**). The N₂O flux in Supplementary Table S4 were collected from literatures.

In the revised version, we have added a detailed **supplementary Table S1** to display the information of experimental samples more clearly.

- *Line 326- how many samples were RNA extracted from?*

Response:

At the site-scale study, RNAs were extracted from three parallel sediment cores of both riparian zone and open water, respectively (2 sediments *3 parallel). We have clarified in the method section in **lines 349-350**.

- *How many metagenomes are included in this study? What is the depth of sequencing?*

Response:

Metagenomic analysis was conducted on all the 100 sediments from the regional scale study with 100 metagenomes obtained in total. The sequencing depth was 10G in this study. The related information was supplemented in the method section in **line 364-369**.

b. For the sampling and analysis of global rivers, more information should be provided on the methods. Were all these global rivers sampled by the same team? Where were these samples analyzed? On line 272 it states these samples were immediately transferred back to the lab- which lab, were they on ice? Were these shipped to a specific lab where all analyses were done? What time of year were the global river sites sampled? Were any analyses done on the global sediments, as no data or methods specific to global rivers is reported in the supplementary tables- it appears to just be a citation table? It is very unclear what was done in this study for the global aspect.

Response:

Thank you for your professional comment.

We collected stream/river sediments around the world through our team and the assistance of collaborators between 2013 to 2020. All the sediments were placed in individual sterile plastic bags, stored on ice immediately, and then transported to our laboratory at a 4 °C cooler as soon as possible, for subsequent analyses. In our laboratory, the N₂O production rate/source were detected by C₂H₂-inhibitor method. In addition, all the physical and chemical properties were exclusively conducted in our laboratory.

The detailed information on global streams sediments were added in **Supplementary Table S1**.

c. How were samples collected- the manuscript just states that sediment cores (line 263) were taken? E.g., what type of corer? Scratch this- I see this information is on lines 278. Consider moving up and stating this information for all studies (site, regional, global), as this coring information seems to only be stated for the site-specific study.

Response:

Thank you for your good comment.

In this study, almost all the sediment cores from all scale studies were collected using an auger (Beijing New Landmark Soil Equipment, Beijing, China) with a plexiglass tube (5.5 cm diameter, 20 cm height). Only the site-scale sediment cores used for ¹⁵N-tracer *semi-in-situ* incubation was stored in the individual plexiglass tube, with both nozzles of each plexiglass tube closed with sealing caps. Besides, the rest sediment cores were placed in individual sterile plastic bags on ice immediately. All the sediment cores were transported to our laboratory in 4°C coolers for subsequent analyses as soon as possible.

The related information has been added to the methods section in **lines 291-296**.

d. No details were provided on DNA extraction or metagenomic sequencing, the manuscript states “filtering was carried out on the original metagenomic data...” with no citation. What study was the original metagenomic data from? Why are analyses (e.g., binning, assembly) being redone here? Either cite the original data publication with DNA extraction, library prep, and sequencing methods or include details in this manuscript.

Response:

Thanks for your constructive comment. Metagenomic analysis on all the regional-scale sediments from Baiyangdian riverine was conducted on the Illumina PE150 (150-bp paired-end). Since qPCR assays cannot distinguish N₂O-production functional genes *nir* and *nor* in ammonia oxidation (ND, NCD) and denitrification (HD) pathways, we tried to analyze the microbial mechanism of N₂O production by metagenomics analysis. Nevertheless, the metagenomics analysis, while analyzing annotated combinations of species and functional genes, is also unable to distinguish effects.

Compared with metagenomics analysis, metagenomic binning technique helps to obtain the whole genome sequence of uncultured microorganisms, retrieve the genome sequence and function of new species, and predict the culture method of unknown species (Mattock et al. 2023; Albertsen et al. 2013; Altshuler et al. 2022; Johansen et al. 2022). Thus, metagenomic binning analyses could provide more accurate annotation of microorganisms and more complete prediction of metabolic pathways.

Previously, metagenomic binning analysis was used to explore the potential metabolic pathways of denitrification associated with N₂O-production hot moment in seasonally frozen soil, which has been published in The ISME Journal (Wang et al. 2023). In this study, we employed this method to distinguish accurately the microbial mechanism of N₂O production by ammonia oxidation and heterotrophic denitrification at the perspective of energy metabolism. Metagenomic binning results indicated that the abundances of N₂O-producing genes in nitrifying bacteria were significantly higher than those in denitrifying bacteria, regardless of the total or individual abundance, further proving our finding that ammonia oxidation is the main pathway for N₂O production.

We have supplemented the detail information on DNA extraction, library preparation and metagenomic sequencing in the method section in **lines 364-369 and Supplementary Table S8**.

References:

- Mattock, J., & Watson, M. A comparison of single-coverage and multi-coverage metagenomic binning reveals extensive hidden contamination. *Nat. Methods* **20**, 1170–1173 (2023).
- Albertsen, M., Hugenholtz, P., Skarshewski, A. Nielsen, K.L., Tyson, G.W., & Nielsen, P.H. Genome sequences of rare, uncultured bacteria obtained by differential coverage binning of multiple metagenomes. *Nat. Biotechnol.* **31**, 533–538 (2013).
- Altshuler, I., Raymond-Bouchard, I., Magnuson, E., Tremblay, J., Greer, C.W., & Whyte L.G. Unique high Arctic methane metabolizing community revealed through *in situ* ¹³CH₄-DNA-SIP enrichment in concert with genome binning. *Sci. Rep.* **12**, 1160 (2022).
- Johansen, J., Plichta, D.R., Nissen, J.N. et al. Genome binning of viral entities from bulk metagenomics data. *Nat. Commun.* **13**, 965 (2022).
- Wang, X.M., Wang, S.Y., Yang, Y.H., Tian, H.Q., Jetten, M.S.M., Song, C.Q., & Zhu, G.B. Hot moment of N₂O emissions in seasonally frozen peatlands. *ISME J.* **17**, 792–802 (2023).

e. Annotation of key nitrogen genes- Functional nitrogen genes have high similarity to functional genes of other processes, thus without stringency in annotation, these genes could easily be misclassified if additional analyses were not performed. For example, ammonia monooxygenase (*amo*) is a homolog with methane monooxygenase (*pmo*), while *nxr* and *nar* are homologs. Please provide additional annotation information on how these annotations were done. What bitscore/e-value cutoffs were used (line 354)? Were additional phylogenetic

analyses/trees done on key functional genes? Was taxonomy of bin taken into consideration? Which subunits or other genes were considered?

Response:

We really appreciate your good and professional comment.

In our study, the functional genes with all subunits were first annotated against GO databases. To distinguish functional genes with homology, we further applied a specialized N-cycling gene database - NCycDB database (<https://github.com/qichao1984/NCyc>) (Tu et al. 2019), which has been used to distinguish N-cycling homologous genes successfully (Wang et al. 2023; Chen et al. 2023; Zhou et al. 2023). Additionally, we also used species annotation to distinguish genes *nor* and *nir* from ND, NCD, and HD pathways. In this study, we have classified nitrifiers containing genes *nor* and/or *nir* as ND-pathway microorganisms, and denitrifiers containing genes *nor* and/or *nir* as NCD/HD-pathway microorganisms. At the same time, the taxonomy of MAGs was also considered. The e-value cutoffs were set up less than $1e^{-5}$ in the blast of MAGs (Wang et al. 2023). Besides, we conducted a phylogenetic tree of the 16S rRNA genes of N₂O-producing MAGs, including 8 Nitrifiers and 7 Denitrifiers, which also assessed the homology between MAGs and known species. We agree that further phylogenetic analysis may make gene annotation more precise. Nevertheless, the above multivariate annotations could eliminate the influence of homologous genes and obtain accurate gene annotations (Wang et al. 2023; Chen et al. 2023; Zhou et al. 2023). As a result, we did not conduct phylogenetic analyses on key functional genes. The detail description has been mentioned in **line 382**.

References:

- Tu, Q., Lin, L., Cheng, L., Deng, Y., & He, Z. NCycDB: a curated integrative database for fast and accurate metagenomic profiling of nitrogen cycling genes. *Bioinformatics* **35**(6), 1040-1048 (2019).
- Wang, X.M., Wang, S.Y., Yang, Y.H., Tian, H.Q., Jetten, M.S.M., Song, C.Q., & Zhu, G.B. Hot moment of N₂O emissions in seasonally frozen peatlands. *ISME J.* **17**, 792–802 (2023).
- Chen, L., Hong, T., Wu, Z., Song, W., Chen, S.X., Liu, Y., & Shen L. Genomic analyses reveal a low-temperature adapted clade in Halorubrum, a widespread haloarchaeon across global hypersaline environments. *BMC Genomics* **24**(1), 508 (2023).
- Zhou, X., Lennon, J.T., Lu, X., & Ruan, A. Anthropogenic activities mediate stratification and stability of microbial communities in freshwater sediments. *Microbiome* **11**(1):191 (2023).

f. Figure 3a has a genome phylogenetic tree. Please include methods for how this tree was generated, and the accession numbers for the isolate genomes.

Response:

Both the 100 isolate genomes and 198 MAGs were deposited into NCBI Sequence Read Archive (SRA) under the accession numbers SAMN33620661-SAMN33620760 (PRJNA943572) and SAMN37879210-SAMN38023784 (PRJNA1031250), respectively, which has been list in the Data Availability (**lines 396-401**) and **Supplementary Tables S8 and S9**.

The methods of constructing phylogenetic trees were well known to us and have been proficiently applied in relevant research (Wang et al., 2012 2020; Zhu et al., 2011 2013), so as to reveal the mechanism of biogeochemical N-cycle at the species level. We completed sequence alignment using Clustalw. Then the genome phylogenetic tree was generated by Maximum Likelihood statistical method (**lines 385-387 and Supplementary information**) using software Mega 11. According to your good comment, the construction method was briefly stated in the revised manuscript and meanwhile detailed as a new section of supplementary information.

References:

- Wang, S.Y., Zhu, G.B., Jetten, M. S.M., & Yin, C.Q. Anammox bacterial abundance, activity and contribution in riparian sediments of the Pearl River Estuary. *Environ. Sci. Technol.* **46**: 8834-8842 (2012).

Wang, S.Y., Zhu, G.B., Zhuang, L.J., Li, Y.X., Liu, L., Lavik, G., Berg, M., Liu, S.T., Long, X.-E., Guo, J.H., Jetten, M. S.M., Kuypers, M.M.M., Li, F.B., Schwark L., & Yin, C.Q. Anaerobic ammonium oxidation is a major N-sink in aquifer systems around the world. *ISME J.* **14**(1): 151-163 (2020).

Zhu, G.B., Wang, S.Y., Wang, W.D., Wang, Y., Zhou, L.L., Jiang, B., Op den Camp, H.J.M., Hefting, M.M., Risgaard-Petersen, N., Peng, Y.Z., Schwark, L., Jetten, M.S.M., & Yin, C.Q. Hotspots of anaerobic ammonia oxidation at land/freshwater interfaces. *Nature Geo.* **6**, 103-107 (2013).

Zhu, G.B., Wang, S.Y., Wang, Y., Wang, C.X., Risgaard-Petersen, N., Jetten, M.S.M., & Yin, C.Q. Anaerobic ammonia oxidation in a fertilized paddy soil. *ISME J.* **5**(12): 1905-12 (2011).

g. Line 351 states there were 198 dereplicated genome bins used for subsequent analysis, however the MAG table includes less than 30 MAGs (Tables S5-S6). Where is the information for the rest of the MAGs? Are the ones in the tables only N cycling? I suggest including a supplementary data file for all MAGs (n=198) with all the relevant reporting information for MIMAGs outlined in Bowers, et al. For example, information on rRNAs and tRNAs should also be included to be in compliance with genome reporting standards.

Response:

Professional comment. In this study, we have used dRep software to de redundant MAGs. The default redundancy parameters for this software are completeness $\geq 75\%$ and coherence $\leq 25\%$. Then, the values of completeness $\geq 75\%$ and contamination $\leq 15\%$ were considered good MAGs. Here, we obtained a total of 198 MAGs. In the initial submission, we only provided 15 MAGs information containing N₂O-production gene, including 7 denitrifiers and 8 nitrifiers. Based on your good suggestion, we have added the information of all the 198 MAGs, including quality, completeness, annotation, relative abundance, according to Bowers et al. (2017), and NCBI sequence accession number (**Line 398**), etc in the new **Supplementary Table S9**.

Reference:

Bowers, R. M., et al. Minimum information about a single amplified genome (MISAG) and a metagenome-assembled genome (MIMAG) of bacteria and archaea. *Nat Biotechnol.* **35**(8), 725–731 (2017).

h. How were the 28 global streams determined to be agricultural? What metrics needed to be met to be included in this study as a global agricultural stream?

Response:

Thank you for your comment. We define rivers with land use type of cropland as agricultural rivers based on the local land use type of the sampling sites. The global scale map of cropland land is as follows (Lu et al. 2020). Our global-scale study sites were located within the range of cropland land. We have supplemented this description in the main text in **lines 285-286**.

● Global synergy cropland map from Lu et al. 2020

● Global-scale sampling streams in our study

Reference:

Lu, M., Wu, W., You, L., See, L., Fritz, S., Yu, Q., Wei, Y., Chen, D., Yang, P., & Xue, B. A cultivated planet in 2010 – Part 1: The global synergy cropland map. *Earth Syst. Sci. Data* **12**, 1913–1928 (2020).

3. In general, the supplementary tables seem to be incorrectly referenced throughout the manuscript, but especially in the methods, making the manuscript hard to review in its current form. Please revise. Specific examples are listed below but these table references should be revised throughout the manuscript.

a. Line 272, supplementary table 4 should be about the global study but is instead spearman correlation table at the regional scale.

Response:

Sorry for this mistake. According to your nice and precise comments, we have added a new **Supplementary Table S6**, including information of regional- and global- scale sampling sites.

b. Line 267- should be something on the regional scale study but is table on site scale investigation.

Response:

The incorrect table references have been revised to **Supplementary Table S1**.

c. Line 238- states information on sampling sites can be found in Supplementary Table S5, but S5 is a table of bins.

Response:

The incorrect table references have been revised to **Supplementary Table S1**.

d. Line 172- Supplementary Tables S5-6 only have information for less than 30 MAGs, not 111 or 198 as stated.

Response:

Yes. In the initial submission, we only provided 15 MAGs information containing N₂O-production gene, including 7 denitrifiers and 8 nitrifiers. Based on your suggestion, we have added the information of all the 198 MAGs, including quality, completeness, annotation, relative abundance, and NCBI sequence accession number, etc. Please see **Supplementary Table S9**.

4. Global claims throughout manuscript are not warranted based on data and should be toned down to fit the results presented in manuscript. See specific examples below, but statements should be revised throughout the manuscript.

a. Line 233- “Ammonia oxidation pathways are the dominant hyporheic N₂O source in streams” – the streams surveyed in the global study are all agricultural, low-order streams which should be clarified in this sentence.

Response:

Agree. The streams in this sentence have been clarified to “low-order agricultural streams” (**Line 248**). Meanwhile, the similar statements also have been revised throughout the whole manuscript.

Thanks for your professional comment!

b. Lines 35-36- while ammonia loading might be more important for low order agricultural streams, nitrate may be important in others.

Response:

The sentence has been tuned down to “These results indicate that ammonia loading, rather than nitrate, plays a more important role in controlling N₂O production within low-order agricultural streams”. (**Lines 36-37**)

Minor comments:

- Remove grey highlight text on lines 258-259- “Its uppr and mid regions”

Response:

Precise comment. The grey highlight has been deleted.

- *Line 259- one should be first if talking about stream order*

Response:

Thank you for your nice comment. We have modified it. **(Line 275)**

- *Line 342- citation for Kneaddata should be included*

Response:

OK! We have attached the source information of Kneaddata (github.com/biobakery/kneaddata) in the revised manuscript. **(Lines 370-371)**

- *Line 342- what is citation 39 citing? It appears to be for a tool called PRINSEQ for quality control of sequencing data- was that used or Kneaddata?*

Response:

Sorry for this incorrect citation. In this study, Kneaddata was used to quality control and host filtering. We have added its source information in the revised manuscript. **(Lines 370-371)**

- *Line 343- add the word “while” before Bowtie2*

Response:

Thank you for your comment. We have revised this. **(Line 373)**

- *Line 346- “15,00” should be “1,500”*

Response:

Thank you. We have revised this. **(Line 375)**

- *Line 348- “Checkm” should be “CheckM”*

Response:

Revised. **(Line 377)**

- *Line 349- “de-redundant” should “dereplicated”*

Response:

Revised. Thanks. **(Lines 378)**

- *Line 353- what database version was used for GTDB?*

Response:

The GTDB database version was v2.3.0. **(Line 381)**

*****END*****

REVIEWER COMMENTS

Reviewer #1 (Remarks to the Author):

Much improved - nice job! Some specific comments:

Line 26: specify agriculture-dominated streams

Line 29: not sure what "N₂O production rate was not limited in riparian or open water zones but mainly depended on the ammonia content." - limited by what? Could consider changing "content" to "concentration"

Line 50: clarify what IPCC methods are (ie for estimating and reporting GHG emissions to the UN)

Line 63: remove "However,"

line 67: Did you remove the idea that the global streams are agriculture-dominated? If so can you be more specific about what kind of streams they are (size, variety of land uses or specific land uses?)

Line 74: I don't think I mentioned this last time (maybe I did) but in some of my research, we see NH₄⁺ and N₂O high at the same time, which is also when dissolved O₂ is quite low. Similarly, in Harrison et al. 2005 (DOI 10.1007/s00027-005-0776-3), they see low O₂ at night and high NH₄⁺ but the NO₃⁻ has already gone down (been denitrified) and N₂O is also low. All this to say, a correlation between NH₄⁺ and N₂O does not necessarily imply the N₂O is coming from the NH₄⁺.

Line 77: change "negative influence" to "is negatively correlated to"

Line 104: change "consistence" with "agreement" or "accordance"

Line 203 - add citation or explain how you conclude this

Line 211: remove " realizes the biological detoxification function and"

Line 213: not sure what you mean here. When nitrite is used up, the NN pathway is activated to produce more nitrite from ammonia (even though it's toxic)?

Line 219: shorten this to "consumption of NO₂⁻ for detoxification"

Line 225: Respiration is the wrong word here (because they're not oxidizing organic carbon). Similarly, denitrification is not the same as fermentation

Line 229: I don't follow the logic here - there should be more N₂O produced from nitrifiers because fewer enzymes are involved? Do more enzymes necessarily make it less energy-efficient? I don't think the number of enzymes matters if they are all going to completion. You could just remove this sentence.

Line 231: add ref for fact that nitrifiers do not (that we've found yet) have N₂O reductase.

Line 237: change "Chinese" to your river basin name

Line 241: change "national" to "regional"

Line 262: add "in lower-order agricultural streams"

Line 266: need ref for China's ammonia consumption

Line 279: "at least 9" - provide a specific number

Line 280 and earlier: "open water" keeps making me think you sampled the water column. What about "riverbed sediments" as opposed to riparian zone?

Line 278: specify what percentage of catchment is ag for stream to be considered agricultural

last thoughts: I'm thinking about your conclusion- we all agree we can be more judicious with NH₄⁺ fertilizers but does it really matter which pathways are being used in the river? Either you dump NH₄⁺

into a river and the nitrifiers make N₂O. Or you dump NH₄⁺ into a river and the nitrifiers produce NO₃⁻. Then the denitrifiers make N₂O. Either way, if you reduce NH₄⁺ to the river you'll cut down on N₂O emissions. Think about how to word this "why we should care" portion.

Recommendation: Accept with minor revisions

Reviewer #2 (Remarks to the Author):

L25: rather than nitrate-derived pathway; should be "the" nitrate-derived...

L43: mole fractions have; should be mole fraction has...

L44: "recent" is a poor word choice since people will be reading this paper for years (hopefully). Plus, the year of the article cited in 2020 - not that recent really. Just say, "A 2020 study"

L45: verb should be past tense "showed"

L52: why is there a slash after "and"

L73: authors state "no difference" between the two zones (i.e., riparian and open water), yet in L89-90 these zones are compared with the riparian zone being called "lower". Clearly, given the std deviation reported, these values are likely the "same".

L94: "zone" misspelled as "zon".

This manuscript is so difficult to read. I cannot focus on the science due to the extremely poor writing.

I am concerned the main conclusion of the research is flawed by the use of the acetylene blockage technique which was largely debunked decades ago. See just one example:

<https://www.sciencedirect.com/science/article/abs/pii/S0038071797000072>

The acetylene blockage technique leads to an underestimation of what the authors call nitrate-derived N₂O. This would cause the appearance of comparatively more ammonium-derived N₂O due to this experimental artifact. The authors describe C₂H₂ as a selective inhibitor which is simply not true. Since it appears the entire study hinges on the false assumption that C₂H₂ is selective, I cannot recommend publication of this work.

Reviewer #3 (Remarks to the Author):

I am satisfied with the response to comments. One note, PRJNA943572 does not seem to be available. The authors should edit this number before acceptance. The other accession numbers I spot checked are available.

Reviewer #4 (Remarks to the Author):

This article explores the pathways of nitrous oxide (N₂O) emissions from river sediments using isotope tracing and acetylene inhibitor methods. The results showed that the ammonium reduction process (ammonia derived pathway) released more N₂O emissions than nitrate reduction. The manuscript illustrates some attractive and innovative conclusions, and the author has made a lot of revisions according to the reviewer's opinion, which has greatly improved the quality of the manuscript. So this manuscript is recommended for acceptance.

Responses to Reviewer Comments

Words labeled *in italic* are the *reviewers' comments*, and labeled with **blue color** is the **response**.

Reviewer #1

Much improved - nice job! Some specific comments:

Line 26: specify agriculture-dominated streams

Response:

Agree. We have clarified it to "agricultural streams" in the revised manuscript. **Line 26**

Line 29: not sure what "N₂O production rate was not limited in riparian or open water zones but mainly depended on the ammonia content." - limited by what? Could consider changing "content" to "concentration"

Response:

This confusing sentence has been re-written, as follows:

"The N₂O production rate showed a low level of heterogeneity between the riparian and riverbed zones but was mainly positive correlation with the ammonia concentration." **Lines 29-31**

Thanks for your good comment.

Line 50: clarify what IPCC methods are (ie for estimating and reporting GHG emissions to the UN)

Response:

According to your nice comments, the IPCC methods have been clarified, as follows:

"Although the IPCC Guidelines for national greenhouse gas inventories include N₂O emission from nitrification⁴, previous studies have been focused on quantifying denitrification and N₂O emissions resulting from nitrate (NO₃⁻)^{5,6}." **Lines 49-51**

References:

4. IPCC: 2019 *Refinement to the 2006 IPCC Guidelines for National Greenhouse Gas Inventories*, in: Vol. 4. Agriculture, forestry and other land uses; Intergovernmental Panel on Climate Change, Geneva, Switzerland, <https://www.ipcc.ch/report/2019-refinement-to-the-2006-ipcc-guidelines-for-national-greenhouse-gas-inventories/>
5. Maavara, T., Lauerwald, R., Laruelle, G. G., Akbarzadeh, Z., Bouskill, N. J., Van Cappellen, P., & Regnier, P. Nitrous oxide emissions from inland waters: Are IPCC estimates too high? *Global Change Biol.* **25**, 473-488 (2019). <https://dx.doi.org/10.1111/gcb.14504>
6. Beaulieu, J. J., Tank, J. L., Hamilton, S. K., Wollheim, W. M., Hall, R. O., Mulholland, P. J., Thomas, S. M. Nitrous oxide emission from denitrification in stream and river networks. *Proc Natl. Acad. Sci. U.S.A.* **108**(1), 214-219 (2011). <https://dx.doi.org/10.1073/pnas.1011464108>

Line 63: remove "However,"

Response:

Deleted.

line 67: Did you remove the idea that the global streams are agriculture-dominated? If so can you be more specific about what kind of streams they are (size, variety of land uses or specific land uses?)

Response:

No. We haven't removed this idea. We have clarified to "temperate and tropical small agricultural streams (lower than fourth-order streams)". Line 67

Line 74: I don't think I mentioned this last time (maybe I did) but in some of my research, we see NH_4^+ and N_2O high at the same time, which is also when dissolved O_2 is quite low. Similarly, in Harrison et al. 2005 (DOI 10.1007/s00027-005-0776-3), they see low O_2 at night and high NH_4^+ but the NO_3^- has already gone down (been denitrified) and N_2O is also low. All this to say, a correlation between NH_4^+ and N_2O does not necessarily imply the N_2O is coming from the NH_4^+ .

Response:

We agree with your professional viewpoint. In the revised manuscript, we have tune down this statement, as follows:

"Ammonium (NH_4^+) showed the most positive correlation with N_2O flux, irrespective of the sampling time and zones." Lines 73-74

Line 77: change "negative influence" to "is negatively correlated to"

Response:

Revised.

Line 104: change "consistence" with "agreement" or "accordance"

Response:

Revised. Line 104

Line 203 - add citation or explain how you conclude this

Response:

We have deleted this confusing sentence.

Line 211: remove " realizes the biological detoxification function and"

Response:

Deleted.

Line 213: not sure what you mean here. When nitrite is used up, the NN pathway is activated to produce more nitrite from ammonia (even though it's toxic)?

Response:

We have deleted this confusing sentence.

Line 219: shorten this to "consumption of NO_2^- for detoxification"

Response:

Revised. Line 210

Line 225: Respiration is the wrong word here (because they're not oxidizing organic carbon). Similarly, denitrification is not the same as fermentation

Response:

We have deleted this wrong sentence. Thanks for your professional comments.

Line 229: I don't follow the logic here - there should be more N₂O produced from nitrifiers because fewer enzymes are involved? Do more enzymes necessarily make it less energy-efficient? I don't think the number of enzymes matters if they are all going to completion. You could just remove this sentence.

Response:

We have deleted this confusing sentence. Thanks.

Line 231: add ref for fact that nitrifiers do not (that we've found yet) have N₂O reductase.

Response:

Added. Line 213

Reference:

42. Stein, L. Y., Arp, D. J., Berube, P. M., Chain, P. S. G., Hauser, L., Jetten, M. S. M., Klotz, M. G., Larimer, F. W., Norton, J. M., Op den Camp, H. J. M., Shin, M., Wei, X. Whole-genome analysis of the ammonia-oxidizing bacterium, *Nitrosomonas eutropha* C91: implications for niche adaptation. 9(12), 2993-3007 (2007). <https://doi.org/10.1111/j.1462-2920.2007.01409.x>

Line 237: change "Chinese" to your river basin name

Response:

Revised. Lines 221-222

Line 241: change "national" to "regional"

Response:

Revised. Line 226

Line 262: add "in lower-order agricultural streams"

Response:

Added. Line 247

Line 266: need ref for China's ammonia consumption

Response:

Added. Line 252

Reference:

12. Tian, H. Q., Lu, C. Q., Melillo, J., Ren, W., Huang, Y., Xu, X. F., Reilly, J. Food benefit and climate warming potential of nitrogen fertilizer uses in China. Environ. Res. Lett. 7, 044020 (2012). <https://dx.doi.org/10.1088/1748-9326/7/4/044020>

Line 279: "at least 9" - provide a specific number

Response:

This number has been clarified to Nine. Line 266

Line 280 and earlier: "open water" keeps making me think you sampled the water column. What about "riverbed sediments" as opposed to riparian zone?

Response:

Agree. We have revised "open water" to "riverbed sediments" throughout the whole manuscript. Lines 30, 59, 70, and another 15 lines throughout the whole manuscript

Line 278: specify what percentage of catchment is ag for stream to be considered agricultural

Response:

According to the reference, the percentage of catchment for agricultural stream is specify to greater than 5.5 %. Line 264

Reference:

Li, S., Shen, Z. F., Liu, K. J., Xu, Z. Y., Wang, H. Y., Jiao, S. H., Liu, X. C., Lei, Y. T. Analysis of terrain gradient effects of land use change in Daqing River Basin. *Journal of Agricultural Engineering* 5, 275-284 (2021). (in Chinese)

last thoughts: I'm thinking about your conclusion- we all agree we can be more judicious with NH₄⁺ fertilizers but does it really matter which pathways are being used in the river? Either you dump NH₄⁺ into a river and the nitrifiers make N₂O. Or you dump NH₄⁺ into a river and the nitrifiers produce NO₃⁻. Then the denitrifiers make N₂O. Either way, if you reduce NH₄⁺ to the river you'll cut down on N₂O emissions. Think about how to word this "why we should care" portion.

Response:

We fully agree with and appreciate your approval about our conclusion regarding the management of agricultural ammonia fertilizers in controlling N₂O emissions. This is one of the main significances of this study.

In addition, as we know, hyporheic N₂O production has been shown to be quite heterogeneous, but this heterogeneity can hardly be interpreted in models and is therefore frequently excluded. Currently, the microbial mechanisms underlying N₂O production in hyporheic exchange zones remain largely unknown. The ammonia-derived (nitrification, nitrifier denitrification, and nitrification coupled with denitrification pathways) and nitrate-derived processes may significantly influence global N₂O budgets and be potentially severely underestimated. Consequently, our study, which focuses on distinguishing these pathways of ammonia-derived N₂O production, would provide new guidance for the global N₂O models of riverine ecosystems. This is another significance of this study.

Thank you for your professional and nice comments to improve the scientific quality of the manuscript.

Recommendation: Accept with minor revisions

Response:

Thank you for your nice approval of our revised manuscript.

Reviewer #2 (Remarks to the Author):

L25: rather than nitrate-derived pathway; should be "the" nitrate-derived...

Response:

Revised. Line 25

L43: mole fractions have; should be mole fraction has...

Response:

Agree. Revised. Line 42

L44: "recent" is a poor word choice since people will be reading this paper for years (hopefully). Plus, the year of the article cited in 2020 - not that recent really. Just say, "A 2020 study"

Response:

Agree. Revised. Line 43

L45: verb should be past tense "showed"

Response:

Revised. Line 44

L52: why is there a slash after "and"

Response:

In fact, it was used to express the different pathways of N₂O production by "nitrifier, nitrifier coupled with denitrifier, and denitrifier", but was ambiguous.

In the revised manuscript, it has been clarified to "nitrifier denitrification, nitrification-coupled denitrification, and heterotrophic denitrification N₂O production pathways". Lines 51-52

L73: authors state "no difference" between the two zones (i.e., riparian and open water), yet in L89-90 these zones are compared with the riparian zone being called "lower". Clearly, given the std deviation reported, these values are likely the "same".

Response:

Thanks for your professional comments. We have deleted this confusing sentence.

L94: "zone" misspelled as "zon".

Response:

We are sorry for this mistake. It has been revised in the revised manuscript. Line 93

This manuscript is so difficult to read. I cannot focus on the science due to the extremely poor writing.

Response:

We are sorry that the science of this manuscript maybe not clear enough due to the non-native writing. According to your critical comments, this manuscript has been polished by native language professional editors from Springer Nature, and the proof of polishing has been provided as follows. Meanwhile, we also provide a short abstract to make this study more clearly.

After the second round of revision, we are confident that the work presented in this manuscript is an original and significant finding, and we would be very grateful if you would be satisfied with our revised manuscript and give us a chance for a further communication.

This document certifies that the manuscript

Ammonium-derived nitrous oxide is a global source in streams

prepared by the authors

Shanyun Wang, Bangrui Lan, Longbin Yu, Manyi Xiao, Liping Jiang, Yu Qin, Yucheng Jin, Yuting Zhou, Gawhar Armanbek, Jingchen Ma, Manting Wang, Mike S. M. Jetten, Hangin Tian, Guibing Zhu, Yong-guan Zhu

was edited for proper English language, grammar, punctuation, spelling, and overall style by one or more of the highly qualified native English speaking editors at SNAS.

This certificate was issued on **March 28, 2024** and may be verified on the SNAS website using the verification code **EB94-21B9-5607-0028-A17F**.

Neither the research content nor the authors' intentions were altered in any way during the editing process. Documents receiving this certification should be English-ready for publication; however, the author has the ability to accept or reject our suggestions and changes. To verify the final

SNAS edited version, please visit our verification page at secure.authorservices.springernature.com/certificate/verify.

If you have any questions or concerns about this edited document, please contact SNAS at support@as.springernature.com.

SNAS provides a range of editing, translation, and manuscript services for researchers and publishers around the world. For more information about our company, services, and partner discounts, please visit authorservices.springernature.com.

Short abstract:

“An increasing number of studies have indicated that the global riverine N₂O emissions have increased by over 4-fold over the last century. It appears that the hyporheic zones in small streams may contribute approximately 85% to these N₂O emissions. However, the mechanisms and pathways controlling hyporheic N₂O production in stream ecosystems remain unknown.

Here, we first report that NH₄⁺-derived pathways (nitrifier nitrification, nitrifier denitrification, and nitrification-coupled denitrification), rather than NO₃⁻-derived pathway (heterotrophic denitrification), are the dominant hyporheic N₂O sources (69.6 ± 2.1%) in small agricultural streams around the world. These observations are made based on complementary methods involving ¹⁵N-¹⁸O dual-isotope tracing, ¹⁵N tracing of *semi-in situ* sediment cores, quantitative reverse transcription PCR (RT-qPCR), and metagenomic assembly and binning analysis for a wide range of sample types and temperature zones globally.

This study highlights the importance of mitigating the agriculturally-derived ammonium in low-order agricultural streams for controlling N₂O emissions. The global models of riverine ecosystems should better represent ammonia-derived pathways, so as to accurately estimate and predict riverine N₂O emissions.”

I am concerned the main conclusion of the research is flawed by the use of the acetylene blockage technique which was largely debunked decades ago. See just one example: <https://www.sciencedirect.com/science/article/abs/pii/S0038071797000072>

The acetylene blockage technique leads to an underestimation of what the authors call nitrate-derived N₂O. This would cause the appearance of comparatively more ammonium-derived N₂O due to this experimental artifact. The authors describe C₂H₂ as a selective inhibitor which is simply

not true. Since it appears the entire study hinges on the false assumption that C₂H₂ is selective, I cannot recommend publication of this work.

Response:

We would like to express our sincere gratitude for your critical comments, although you rejected our manuscript in your first round of review. There are two main reasons to prove that our results are credible:

(1) You questioned that the main conclusion of the research is flawed by the use of the acetylene blockage technique, which may be due to a misunderstanding that we did not make ourselves understood. The C₂H₂ inhibitor method can be used with either low (0.1-10 Pa, 0.01-0.1% v/v) or high (1-20 kPa, 1-20% v/v) C₂H₂ concentration (Bollmann et al., 1997). In our study, we used the low acetylene inhibitor method, rather than the high acetylene inhibitor method, which is the flawed method that you point out.

a) Low concentrations of acetylene (0.1-10 Pa, 0.01-0.1% v/v) irreversibly inhibit the ammonium monooxygenase of nitrifiers (Hynes and Knowles, 1978; Berg et al., 1982), so that the N₂O released is only due to denitrification. In our study, we used the 0.01% C₂H₂-inhibitor technology to inhibit nitrification, and the N₂O release rate represents **the potential rate of denitrification (NO₃-derived) N₂O production**. In the meanwhile, we used ZnCl₂ (600 μl, 7M) as a biological inactivator, taking advantage of its ability to inactivate proteins (Zhu et al., 2013; Wang et al., 2020). Based on the experimental and groups shown below, the related N₂O production rate in agricultural streams worldwide could be accurately distinguished and calculated:

- The production rate of NH₄⁺-derived N₂O was calculated by the different value of the N₂O production rate between treatment i (Control) and ii (C₂H₂; 0.1% v/v). Here, the production rate of NH₄⁺-derived N₂O was the total rate of NN, ND, and NCD pathways.
- The production rate of biotic N₂O was calculated by the different value of the N₂O production rate between treatment i (Control) and iii (ZnCl₂; 600 μl, 7M).
- The production rate of NO₃⁻-derived N₂O was calculated by the different value of biotic and NH₄⁺-derived processes.

Notably, the 0.01% C₂H₂ inhibitor method is commonly used to study the potential rates of N₂O production via NH₄⁺-derived (nitrification) and NO₃⁻-derived (denitrification) processes, and it has been widely adopted in numerous studies, i.e. Zhu et al., *Proc. Natl. Acad. Sci. U.S.A.* 2013, Wang et al *ISME J* 2023, Hink et al., *ISME J* 2018, Garrido et al., *Soil Biol Biochem* 2000, Hu et al. *Sci Rep* 2022, and so on.

b) High concentrations of acetylene (1-20 kPa) inhibit the N_2O reductase of the denitrifiers (Yoshinari et al., 1977), as a result, N_2O , instead of N_2 will accumulate, thus giving a measure of the denitrification rate (N_2O+N_2). However, the acetylene probably results in the scavenging of part of the NO produced as the intermediate in the denitrification process to NO_2 , thus, it can not be further reduced to N_2O . Consequently, the denitrification rates are underestimated. It is indeed a practical problem about the determination of denitrification rates by the acetylene blockage technique. **However, this is not our research object and purpose. Instead, the N_2O sources and rates are the key and sole topics in our study.** In addition, despite the serious limitations, this method is one among the very few methodological options to estimate total denitrification at the high temporal resolution and the small spatial scale, with limited workload and costs involved. The high acetylene inhibition method has been used in studies (Ishii et al., ISME J 2011; Felber et al., Biogeosciences Discuss 2012).

(2) Besides the 0.01% C_2H_2 -inhibitor technology, our study also used complementary methods such as the ^{15}N - ^{18}O double-isotope tracing, ^{15}N tracing of the semi-in-situ sediment core, quantitative reverse transcription PCR (RT-qPCR), and metagenomic sequencing, annotation, and assembly genome analyses, to investigate the source and microbial mechanism of hyporheic N_2O source production. Your critical comments have made us more certain of the necessity of studying samples through multiple techniques concurrently.

The main microbial sources of NH_4^+ -derived and NO_3^- -derived N_2O production and their associated enzymes (Unpublished)

Microorganisms carry enzymes that perform 8 redox reactions involving 7 key inorganic nitrogen species (see color coded circles for: NO_3^- , NO_2^- , NO , N_2O , NH_2OH , NH_4^+ , N_2). The main microbial sources of N_2O production involve: 1) NH_4^+ -derived N_2O : with ambient NH_4^+ as substrate, the NH_4^+ -derived N_2O production process includes nitrifier nitrification (NN, red), nitrifier denitrification (ND, purple), and nitrification-coupled denitrification (NCD, blue) pathways. 2) NO_3^- -derived N_2O : With ambient NO_3^- as substrate, the NO_3^- -derived process represents the lumped heterotrophic denitrification (HD, green) pathway (Zhu et al., Proc. Natl. Acad. Sci. U.S.A. 2013; Kool et al., Eur. J. Soil Sci. 2010; Kool et al., Soil Biol. Biochem. 2011). Genes encoding enzymes that conduct the important transformations include those for ammonium monooxygenase (Amo), hydroxylamine oxidoreductase (Hao); nitrite oxidoreductase (Nxr); membrane-bound (Nar) and periplasmic (Nap) dissimilatory nitrate reductases; nitrite reductase (Nir); nitric oxide reductase (Nor); nitrous oxide reductase (NosZ).

a) **^{15}N - ^{18}O double-isotope tracing** The ^{15}N - ^{18}O dual-isotope tracing method was improved by our team compared with that published by Wrage et al., 2005; Kool et al., 2010 2011; and Zhu et al., 2013, which ensured incubation under *in situ* oxygen and temperature conditions. Therefore, the results detected by the improved method were closer to the actual production conditions than the original method. The improved ^{15}N - ^{18}O dual-isotope tracing method has recently been reported in the studies by our team (Wang et al., ISME J. 2023; Jiang et al., Global Change Biol. 2023; Qin et al., Environ. Sci. Technol. 2023).

b) **^{15}N tracing of the *semi-in-situ* sediment core** In fact, the ^{15}N tracing incubation of *semi-in-situ* sediment core was designed by the first author Dr. Wang in this study. ^{15}N - NH_4^+ and ^{15}N - NO_3^- were added to trace NH_4^+ -derived and NO_3^- -derived N_2O production, respectively. Meanwhile, the sediment core was incubated under the intact sediment structure and the *in-situ* oxygen and temperature conditions, which were closer to the actual production conditions than those of slurry incubation (Trimmer et al., Mar. Ecol. Prog. Ser. 2006; Zhu et al., Nature Geo. 2013; Wang et al., ISME J. 2020). In contrast, slurry incubation completely disrupts the natural gradients of substrates and redox reactions in soils (Trimmer et al., Mar. Ecol. Prog. Ser. 2006), thereby accelerating the conduction of the medium in the pore water. Without the addition of ^{18}O tracers, it is unable to further distinguish the NH_4^+ -derived process among NN, ND and NCD pathways. However, this method could accurately measure the *semi-in-situ* N_2O production rate and distinguish the contributions of NH_4^+ -derived and NO_3^- -derived processes.

c) **Quantitative reverse transcription PCR (RT-qPCR)** To further investigate the microbial mechanism and activity, the transcript abundances of microbial N₂O-related genes were quantified via RT-qPCR assay. The transcript abundances showed that the N₂O-production related genes had higher transcript abundances (*amoA*, *norB*, *nirS*, and *nirK*) than the reduction gene (*nosZ*) in both the riparian and open water zones, providing evidence that the hyporheic zone has a higher potential of N₂O production. In addition, due to the common genes *nirK* and *norB* in ND, NCD, and HD pathways, the transcript abundances of the NH₄⁺-derived N₂O production were higher than those of the NO₃⁻-derived N₂O production, indicating that denitrification is not the main process of N₂O production.

d) **Metagenomic sequencing, annotation, and assembly genome analyses** In this study, we combined species annotation with functional gene annotation to distinguish the species and relative abundances of the nitrifying and denitrifying bacteria with N₂O-related genes. Later, we distinguished the N₂O-production functional genes *nir* and *nor* in NH₄⁺-derived N₂O (ND, NCD) and NO₃⁻-derived N₂O (HD) pathways, thereby revealing the metabolic mechanism of N₂O production. Compared with metagenomics analysis, the metagenomic binning technique helps to obtain the whole genome sequence of uncultured microorganisms, retrieve the genome sequence and function of new species, and predict the culture method of unknown species (Mattock et al. 2023; Albertsen et al. 2013; Altshuler et al. 2022; Johansen et al. 2022). Thus, metagenomic binning analyses can provide more accurate annotations of microorganisms and more complete prediction of metabolic pathways. Previously, metagenomic binning analysis was used to explore the potential metabolic pathways of denitrification associated with N₂O-production hot moment in seasonally frozen soil, which has been published in The ISME Journal (Wang et al. 2023). In this study, we employed this method to accurately distinguish the microbial mechanism of N₂O production by ammonia oxidation and heterotrophic denitrification from the perspective of energy metabolism. Our metagenomic binning results indicated that the abundances of N₂O-production genes in nitrifying bacteria were significantly higher than those in denitrifying bacteria, regardless of the total or individual abundances, further proving our finding that ammonia oxidation is the main pathway of N₂O production.

Due to the limitations of *in situ* field of widely spatiotemporal scales, we selected different but suitable research methods at different scales. These above complementary technologies of isotopic, inhibitor, and molecular biology methods came to the consistent conclusion that **NH₄⁺-derived N₂O production (NN, ND, and NCD), rather than NO₃⁻-derived N₂O production (HD), is the dominant hyporheic N₂O source in global streams.** Thus, the global models of riverine ecosystems need to better represent NH₄⁺-derived N₂O production, so as to accurately estimate and predict riverine N₂O emissions. Taken together, these factors highlight the importance of N management, especially NH₄⁺ fertilizer, in the agricultural field.

Your criticisms and suggestions have led us to highlight the new technologies and research results, and in the second round of revision, we have toned down “the N₂O production rate observed by the 0.01% C₂H₂ inhibitor method” to “the potential rate of N₂O production”. These revisions have indeed improved the scientific quality of our manuscript.

References:

Albertsen, M., Hugenholtz, P., Skarshewski, A. Nielsen, K.L., Tyson, G.W., & Nielsen, P.H. Genome sequences of rare, uncultured bacteria obtained by differential coverage binning of multiple metagenomes. *Nat. Biotechnol.* **31**, 533–538 (2013).

- Altshuler, I., Raymond-Bouchard, I., Magnuson, E., Tremblay, J., Greer, C.W., & Whyte L.G. Unique high Arctic methane metabolizing community revealed through *in situ* $^{13}\text{CH}_4$ -DNA-SIP enrichment in concert with genome binning. *Sci. Rep.* **12**, 1160 (2022).
- Bollmann, A. & Ralf, C. Acetylene blockage technique leads to underestimation of denitrification rates in oxic soils due to scavenging of intermediate nitric oxide. *Soil Biol. Biochem.* **29**, 1067-1077 (1997).
- Felber, R., Conen, F., Flechard, C. R., Neftel, A. The acetylene inhibition technique to determine total denitrification ($\text{N}_2+\text{N}_2\text{O}$) losses from soil samples: potentials and limitations. *Biogeosciences Discuss.* **9**, 2851–2882 (2012).
- Garrido, F., Hénault, C., Gaillard, H., Germon, J. C. Inhibitory capacities of acetylene on nitrification in two agricultural soils. *Soil Biol. Biochem.* **32**(11-12), 1799-1802 (2000).
- Hink, L., Gubry-Rangin, C., Nicol, G. W., et al. The consequences of niche and physiological differentiation of archaeal and bacterial ammonia oxidisers for nitrous oxide emissions. *ISME J.* **12**, 1084–1093 (2018).
- Hu, L., Dong, Z., Wang, Z., et al. The contributions of ammonia oxidizing bacteria and archaea to nitrification-dependent N_2O emission in alkaline and neutral purple soils. *Sci. Rep.* **12**, 19928 (2022).
- Ishii, S., Ohno, H., Tsuboi, M., et al. Identification and isolation of active N_2O reducers in rice paddy soil. *ISME J.* **5**, 1936–1945 (2011).
- Jiang, L. P., et al. Complete ammonia oxidization in agricultural soils: High ammonia fertilizer loss but low N_2O production. *Global Change Biol.* **29**(7), 1984-1997 (2023).
- Johansen, J., Plichta, D. R., Nissen, J. N., et al. Genome binning of viral entities from bulk metagenomics data. *Nat. Commun.* **13**, 965 (2022).
- Kool, D. M., Wrage, N., Zechmeister-Boltenstern, S., et al. Nitrifier denitrification can be a source of N_2O from soil: a revised approach to the dual-isotope labelling method. *Eur. J. Soil Sci.* **61**(5), 759-772 (2010).
- Kool, D. M., Dolfing, J., Wrage, N., Van Groenigen, J. W. Nitrifier denitrification as a distinct and significant source of nitrous oxide from soil. *Soil Biol. Biochem.* **43**(1), 174-178 (2011).
- Mattock, J., & Watson, M. A comparison of single-coverage and multi-coverage metagenomic binning reveals extensive hidden contamination. *Nat. Methods* **20**, 1170–1173 (2023).
- Yu, Q., et al. Contribution of ammonium-induced nitrifier denitrification to N_2O in paddy fields. *Environ. Sci. Technol.* **57**(7), 2970–2980 (2023).
- Trimmer, M., Risgaard-Petersen, N., Nicholls, J. C., & Engstroem, P. Direct measurement of anaerobic ammonium oxidation (anammox) and denitrification in intact sediment cores. *Mar. Ecol. Prog. Ser.* **326**, 37-47 (2006).
- Wang, S. Y., Zhu, G., Zhuang, L., et al. Anaerobic ammonium oxidation is a major N-sink in aquifer systems around the world. *ISME J.* **14**, 151-163 (2020).
- Wang, X. M., Wang, S. Y., Yang, Y. H., Tian, H. Q., Jetten, M. S. M., Song, C. Q., Zhu, G. B. Hot moment of N_2O emissions in seasonally frozen peatlands. *ISME J.* **17**, 792–802 (2023).
- Wrage, N., van Groenigen, J. W., Oenema, O., Baggs, E. M. A novel dual-isotope labelling method for distinguishing between soil sources of N_2O . *Rapid Commun. Mass Spectrom.* **19**, 3298–3306 (2005).
- Yoshinari, T., Hynes, R., Knowles, R. Acetylene inhibition of nitrous oxide reduction and measurement of denitrification and nitrogen fixation in soil. *Soil Biol. Biochem.* **9**, 177–183 (1977).
- Zhu, G. B., Wang S. Y., Wang W., Wang Y., Zhou L., Jiang B., et al. Hotspots of anaerobic ammonium oxidation at land–freshwater interfaces. *Nature Geo.* **6**(2), 103-107 (2013).
- Zhu, X., Burger, M., Doane, T. A., Horwath, W. R. Ammonia oxidation pathways and nitrifier denitrification are significant sources of N_2O and no under low oxygen availability. *Proc. Natl. Acad. Sci. U.S.A.* **110**(16), 6328-6333 (2013).

Reviewer #3

I am satisfied with the response to comments. One note, PRJNA943572 does not seem to be available. The authors should edit this number before acceptance. The other accession numbers I spot checked are available.

Response:

Thanks for your nice approval for our revision.

In our original submission, this BioProject submission (PRJNA943572) would be released on 2024-04-30 or upon publication, whichever is first. Due to your kind comments, this project (PRJNA943572) has been updated to be available now.

Reviewer #4

This article explores the pathways of nitrous oxide (N₂O) emissions from river sediments using isotope tracing and acetylene inhibitor methods. The results showed that the ammonium reduction process (ammonia derived pathway) released more N₂O emissions than nitrate reduction. The manuscript illustrates some attractive and innovative conclusions, and the author has made a lot of revisions according to the reviewer's opinion, which has greatly improved the quality of the manuscript. So this manuscript is recommended for acceptance.

Response:

Thank you for your nice approval of our revised manuscript.

We thank all reviewers very much for their both positive and negative comments. These professional and insightful comments are helpful for the dissemination of our results to a wide scientific community. Meanwhile, we want to provide special acknowledgement to the editor for your nice work of our manuscript. It is our sincere hope that the improvements of our manuscript have been satisfactory and that our contribution is now ready for discussion in the public domain.

*****END*****